# Method to characterize directional changes in arctic sea ice drift and associated deformation due to synoptic atmospheric variations using Lagrangian dispersion statistics

J.V. Lukovich[1], C.A. Geiger[2], D.G. Barber[1]

[1]Centre for Earth Observation Science, University of Manitoba, Winnipeg, R3T 2N2, CANADA
[2]College of Earth, Ocean, and Environment: Geography, University of Delaware, Delaware, 19716, USA

*Correspondence to*: J.V. Lukovich (Jennifer.Lukovich@umanitoba.ca)

**Abstract.** A framework is developed to assess the directional changes in sea ice drift paths and associated deformation processes in response to atmospheric forcing. The framework is based on Lagrangian statistical analyses leveraging particle dispersion theory which tells us whether ice drift is in a subdiffusive, diffusive, ballistic, or superdiffusive dynamical regime using single-particle (absolute) dispersion statistics. In terms of sea ice deformation, the framework uses two- and three-particle dispersion to characterize along and across-shear transport as well as differential kinematic parameters. The approach is tested with GPS beacons deployed in triplets on sea ice in the southern Beaufort Sea at varying distances from the coastline in fall of 2009 with eight individual events characterized. One transition in particular follows the SLP high on 8 October in 2009 while the sea ice drift was in a superdiffusive dynamic regime. In this case, the dispersion scaling exponent (which is a slope between single-particle absolute dispersion of sea ice drift and elapsed time) changed from superdiffusive ($\alpha \sim 3$) to ballistic ($\alpha \sim 2$) as the SLP was rounding its maximum pressure value. Following this shift between regimes, there was a loss in synchronicity between sea ice drift and atmospheric motion patterns. While this is only one case study, the outcomes suggest similar studies be conducted on more buoy arrays to test momentum transfer linkages between storms and sea ice responses as a function of dispersion regime states using scaling exponents. The tools and framework developed in this study provide a unique characterization technique to evaluate these states with respect to sea ice processes in general. Application of these techniques can aid ice hazard assessments and weather forecasting in support of marine transportation and indigenous use of near-shore Arctic areas.

## 1 Introduction

Quantifying a transitioning global climate with respect to associated anthropogenic forcing, impacts, and resiliency requires in depth understanding of sea ice drift and deformation; specifically the connections to critical air-sea dynamic exchanges. Current levels of connectivity begin with an accelerated ice drift speed over the past several decades as a result of a weaker ice cover. The weaker ice cover is due to the loss of

thicker – and therefore stronger – multiyear ice which has been exported from the Arctic Ocean through the Fram Strait as a result of changing polar atmospheric circulation patterns (Hakkinen et al. 2008; Barber et al., 2009; Rampal et al. 2009b; Spreen et al., 2011; Kwok et al., 2013). The changing drift patterns have subsequently altered ice-ice interactions which are associated with the opening of leads and closings into

pressure ridges and rubble fields. Such features change in location and shape during sea ice deformation processes; specifically divergence and convergence events. These changing deformation patterns in turn impact moisture and heat exchange between the ocean and atmosphere by way of the sea ice thickness and redistribution (Hutchings et al., 2011, Bouillon et al., 2015). All of these changing patterns impact international shipping, indigenous use of near-shore Arctic areas, and pollutant/contaminant transport.

One area of particular importance to a number of stakeholders is the Beaufort Sea region located just north of Canada and the United States (IPCC, 2015). Here, the large-scale ice-covered oceanic Beaufort Gyre circulation pattern is predominantly anticyclonic (clockwise) with reversals to cyclonic (counter clockwise) circulation throughout the annual cycle in response to changing surface wind patterns and oceanic responses to coastal boundaries (LeDrew et al. 1991; Preller and Posey, 1989; Lukovich et al.,

2011; Proshutinsky et al., 2015). When circulation patterns change, the ice cover changes abruptly which impacts navigation channels as a result of ice-ice and ice-coastline momentum and energy flux exchanges (e.g., The Polar Group, 1980; Hwang, 2005; McPhee, 2012), air-sea heat exchanges increase (e.g., Carmack et al., 2015), with newly opened leads venting large amounts of moisture into the atmosphere as a strong mass exchange process (Bourassa et al., 2013). Understanding how these changes develop and relate

to the orientation of a coastline is essential when diagnosing response patterns. For clarity, directional change in this study refers to changes in the ice drift path relative to storm tracks, which have typical duration on the order of days and recurrence rates on the order of several days to weeks.

Arctic air-ice-sea interactions on synoptic timescales (several days to weeks) are governed by three interactive components: i) sea ice motion, ii) a confining coastline, and iii) atmospheric forcing. Previous

studies have examined sea ice drift and deformation response to atmospheric forcing and coastline geometry on varying timescales (Overland et al., 1995; Richter-Menge et al., 2002; Geiger and Perovich, 2008; Hutchings et al., 2011). In an assessment of springtime sea ice drift in a region to the west of the Antarctic Peninsula, Geiger and Perovich (2008) identified low-frequency motion in response to atmospheric forcing and coastal geometry associated with regional-scale transport, and higher-frequency

near-inertial oscillatory motion associated with mixing. On regional and synoptic scales, Richter-Menge et al. (2002) also distinguish between translational and differential motion associated with shear zones and discontinuities in the ice drift characteristics in the southern Beaufort Sea.

The role of forcing (wind stress) and coastline geometry in establishing coherence in lead patterns/fractures in the ice cover captured by sea ice deformation has also been explored in past studies (Overland et al., 1995; Hutchings et al., 2005, 2011). Overland et al. (1995) demonstrated that in the Beaufort Sea for spatial scales: i) exceeding 100 km the sea ice cover moves as an aggregate; ii) less than 100 km the ice cover moves as an aggregate or discrete entity; and iii) on the order of 1 km the ice cover is characterized by floe (ice-ice) interactions. Through analysis of a nested beacon configuration and array with spatial scales ranging from 10 km to 140 km as part of the late winter (April) 2007 Sea Ice Experiment: Dynamic Nature of the Arctic (SEDNA) campaign in the Beaufort Sea, Hutchings et al. (2011) demonstrated coherence between 140 km and 20km divergence arrays for time periods of up to 16 days in March. Over shorter (sub-synoptic) timescales from May 2007 onward, nested buoy arrays captured the loss of connectivity in the sea ice cover associated with the winter-to-summer transition during a substantial ice-loss year (Stroeve et al., 2008). Building upon these earlier findings, an effective next step is to characterize the time-evolving relationships of an ice flow field by leveraging the physics principles in Lagrangian dispersion statistics, which is the focus of this work beginning with a summary of the fundamental principles below.

## 1.1 Lagrangian dispersion statistics

Traditionally used to characterize patterns and structure in atmospheric and oceanic dynamical phenomena, Lagrangian dispersion statistics identify topological and dynamical features within a flow field (LaCasce, 2008 and references therein). A number of previous studies already exist using Lagrangian dispersion and ice beacon trajectories to quantify sea ice drift and deformation in the Arctic (Colony and Thorndike, 1984, 1985; Rampal, 2008, 2009a,b, 2016; Lukovich et al., 2011, 2014, 2015). And so, as a review, single-particle (absolute) dispersion provides a signature of circulation and organized structure in the fluid (or fluid-like) system, and captures linear time-dependence in fluctuating velocity variance characteristic of turbulent diffusion theory (Taylor, 1921; Rampal et al., 2009a). Departures in ice fluctuating velocity statistics from turbulent diffusion are attributed to intermittency associated with sea ice deformation and internal ice stress (Rampal et al., 2009).

A two-particle (relative) dispersion analysis monitors sea ice deformation by evaluating time changes in the distances between two ice floe, with each floe typically marked with a GPS beacon. Through evaluation of buoy pair separations as a proxy of strain-rate (divergence, convergence, and shear) components combined, two-particle dispersion provides, by the normal flow rule, an approximation for internal ice stress (Rampal et al., 2009b). Two-particle dispersion also explains heterogeneity and intermittency in the sea ice deformation field associated with space/time coupling inherent in fracturing of

the sea ice cover as described by sea ice mechanics (Rampal et al., 2008; Weiss, 2013; Weiss and Dansereau, 2017). Rampal et al. (2008) noted that a triplet or multiple-particle analysis explains deformation and related small-scale kinematic features in sea ice.

Three-particle dispersion, and triplet areas in particular, are explored in this study to distinguish
between strain-rate components of divergence/convergence and shear. Specifically, sea ice divergence depicts open water formation and accompanying processes such as new ice growth, brine rejection to the ocean, and heat and moisture exchange. Conversely, ice convergence depicts ridge and rubble formation which contribute to ice thickness redistribution (Stern and Lindsay, 2009; Kwok and Cunningham, 2012) and are further involved in the triplet area phenomenon referred to by Thorndike (1986) as "nondivergent
diffusive mixing" due to compressibility in the ice cover. Finally, shear depicts deformation imposed by coastal and ice pack boundaries, such as is found in the Beaufort Sea due to anticyclonic Beaufort Gyre circulation.

Early studies of oceanic circulation have used multiple particles to monitor small-scale deformation and mixing. Ice beacon triplet arrays have also been used to monitor sea ice deformation off the Canadian
east coast and in Antarctica (Prinsenberg et al., 1997; Heil et al., 2002; 2008; 2009; 2011). Studies of correspondence between ice stress, convergence and atmospheric forcing off the southern coast of Labrador in March, 1996 showed little change in convergence within an already compact ice cover, in addition to an increase in stress with winds and decrease in stress with temperature as the icepack loses its ability to transmit pressure (Prinsenberg et al., 1997). These results are consistent with studies of derived ice motion
fields using synthetic aperture radar data showing sea ice deformation and production 1.5 times higher in the seasonal than in the perennial ice zone throughout the Arctic in late fall and winter due to differences in ice strength and thickness (Kwok, 2006).

Application of Lagrangian dispersion (single- and two-particle) in the seasonal ice zone in the Beaufort Sea region in past studies showed that single-particle dispersion captures the existence of two distinct
dynamical regimes characterized by distinctive scaling laws; $t^2$ scaling in the zonal direction characteristic of advection, and $t^{5/4}$ scaling in the meridional direction characteristic of quasi-geostrophic 2D turbulence (Lukovich et al., 2011). Two-particle dispersion studies in this region, based on an assessment of loop and meander reversal events, demonstrated enhanced meridional separation indicative of ice-ice and ice-coast interactions and increased connectivity in the ice cover in winter relative to spring (Lukovich et al., 2014).
In this study we build upon previous analyses to quantify spatiotemporal synoptic changes in sea ice drift and deformation using a novel observational and analytical approach based on one-, two-, and three-particle dispersion statistics. To address the characterization of sea ice drift and deformation using Lagrangian dispersion statistics, we pose the following research questions:

i)   How can directional changes in sea ice drift trajectories be characterized? What insight is provided by single-particle dispersion statistics?

ii) How can associated/corresponding sea ice deformation processes for varying distances relative to the coastline be characterized? What insight is provided by two- and three-particle dispersion statistics?

We address these questions through the development of a framework for understanding sea ice drift and deformation in the Beaufort Sea on daily timescales based on single-, two-, and three-particle dispersion. Diagnostic information resulting from this framework can be used by modelers, satellite image analysts and in field observations to quantify relative contributions (atmospheric, oceanic, internal ice stress) to ice drift. Furthermore, these methods are relevant both from an observational and modelling

perspective, with the potential for application to forthcoming model-data comparisons, and an assessment of dynamical regimes in other regions of the Arctic.

The paper proceeds as follows. Data used to identify directional changes in sea ice drift are described in Section 2. In Section 3, methods based on Lagrangian dispersion and the triplet area approach, are presented. Results associated with each of the two objectives are provided in Section 4, followed by the

discussion in Section 5 and conclusions in Section 6 in addition to a short description of future work.

**2 Data**

Sea ice position data were obtained from an array of ten ice beacons and one ice mass balance buoy launched from the CCGS *Amundsen* in the marginal ice zone of the southern Beaufort Sea in September, 2009 (Figure 1). From this array, four triangular configurations were selected, hereinafter referred to as

triplets A to D, to monitor divergence and convergence of sea ice, with initial inter-beacon distances of approximately 11, 11, 11.5, and 7 km for the shortest leg, and 15, 37, 11.5, and 12.5 km for the longest leg, respectively. Triplets A to D were deployed on multi-year ice (MYI) and labeled according to their proximity to the continental coastline: triplet A was located closest to the coastline, while triplet D was located furthest from the coastline. Position coordinates were available for all beacons in: triplet A until

October 6[th]; triplet B until November 4[th]; triplet C until November 25[th], and triplet D until November 3[rd], yielding time intervals with durations of 28, 56, 77, and 59 days, respectively. As reported in Lukovich et al. (2011), positional accuracy of the ice beacons ranged from $\delta x = 2.5$ to 5 m based on circular and spherical error probability associated with the GPS module, while temporal accuracy was on the order of nanoseconds and thus negligible. Position accuracy for the ice mass balance buoys was less than 3m

according to Garmin GPS16X-HVS product Standard GPS accuracy. The temporal resolution of the beacon data is two hours, and daily averages were calculated for the analysis and time series. Since the anticipated lifetime of the beacon batteries is at least one year, the beacon longevity may be attributed either to

alternative mechanical failure or ice deformation and ridging. The data are archived long term through the Canadian Polar Data Catalogue (Buoy triplet centroid 2009 data, 2016).

Sea ice extent and ice type are examined using Environment Canada Canadian Ice Service (CIS) weekly ice charts, in addition to 12.5 km resolution Advanced Microwave Scanning Radiometer – EOS (AMSR-E) daily sea ice concentration (SIC) data. Daily and weekly maps illustrate spatial variability of sea ice concentrations in the Beaufort Sea, while also enabling an assessment of ice conditions in the vicinity of the triplet centroids during their evolution from September to November, 2009.

Atmospheric forcing in the form of sea level pressure (SLP), wind speed and direction, and surface air temperature (SAT) was obtained from North American Regional Reanalysis (NARR) data (Mesinger et al., 2006). Daily atmospheric forcing is derived by averaging 3-hourly NARR data in the vicinity of triplet centroids. Time series of daily-averaged sea level pressure (SLP) are then characterized into relative high (maxima) and low (minima) pressure tendencies. Time series of daily-averaged 10m NARR winds are used to characterize on-shore and off-shore winds.

## 3 Methods

In this study, single-, two-, and three-particle dispersion statistics are used to quantify dynamical changes in the sea ice cover. Specifically, directional changes in ice drift trajectories are quantified through single-particle dispersion. Associated deformation processes and differential kinematic parameters (DKPs) are identified through two- and three-particle dispersion. This methodology provides a diagnostic product that quantifies sea ice response to atmospheric forcing (through detection of distinct dynamical regimes in single-particle dispersion), and ice interactions (through identification of deformation and DKPs in two- and three-particle dispersion). Therefore this approach can contribute to topics not specifically addressed since they are beyond the scope of this study including an understanding of dominant terms in the force balance of sea ice and yielding mechanics associated with DKP component ratios.

In this section we describe Lagrangian dispersion used to quantify i) directional changes in ice drift trajectories and ii) corresponding sea ice deformation that develops in response to atmospheric forcing for varying distances from the coastline. Presented also are diagnostics (base-to-height and perimeter-to-area ratios, Okubo-Weiss criterion, and shear-to-divergence ratios) used to characterize sea ice drift and deformation during directional changes in ice drift paths.

### 3.1 Single-particle dispersion, triplet centroids, and directional changes in ice drift trajectories

Single-particle dispersion monitors organized structure in the flow field and is defined as (Taylor, 1921)

$$A^2 = \langle |x_i(t) - x_i(0) - \langle x_i(t) - x_i(0) \rangle|^2 \rangle$$

for $x_i$ the zonal and meridional location of the *i*th particle/beacon in the ensemble as a function of elapsed time, *t*, and where angle brackets denote ensemble averaging. Flow dynamics are characterized by the scaling exponent $\alpha$ according to the relation

$$A^2 \sim t^\alpha,$$

where, $\alpha > 1$ corresponds to a superdiffusive dynamical regime, $\alpha = 1$ to a diffusive regime, and $\alpha < 1$ to a subdiffusive or "trapping" regime. Within the superdiffusive category, $\alpha = 2$ corresponds to a ballistic regime indicative of advection, $\alpha = 5/3$ to an elliptic regime, $\alpha = 5/4$ to a hyperbolic regime. As noted in previous studies, in the context of sea ice dynamics, a ballistic dispersion regime depicts advection
associated with organized structure in the ice drift field. An elliptic regime indicates a strong rotational component in the ice drift field, whereas a hyperbolic regime indicates strain (shear and stretching)-dominated flow associated with along-shear transport such as, in this study region, anticyclonic Beaufort Gyre circulation. A diffusive regime captures the behaviour of particles/beacons/ice floes that follow independent random walks (Provenzale, 1999). A subdiffusive regime characterizes trapping such as would
occur with dominant contributions from ice-ice-interactions. Single-particle dispersion also provides a signature of Lévy dynamics (Schlesinger et al., 1993), or processes associated with intermittent "flight" (anomalous or ballistic transport) and "trapping" events, such as are observed in directional changes in the ice cover explored in the present investigation. Previous studies have illustrated the use of Lagrangian dispersion statistics to quantify dynamical regimes in the ice cover (Rampal et al., 2009a,b; Lukovich et al.,
2011, 2015; Rampal et al., 2016). In this study, we build upon previous analyses by identifying inflection points in total absolute dispersion results to identify transitions in distinct dynamical regimes.

Ice and atmospheric conditions are investigated according to the spatial and temporal evolution in ice beacon triplet centroids. In this study, a regional characterization of ice drift (provided by $\alpha_R$) is computed using triplet A to D centroids in the ensemble calculation, while a local characterization of ice
drift (provided by $\alpha_L$) is computed using beacons comprising each triplet in the ensemble calculation for single-particle dispersion. Ice drift velocities for each triplet centroid, computed as outlined in Appendix A, further highlight acceleration/deceleration in the triplets during fall, 2009. Turning angles are calculated as the difference between 10 m NARR surface winds and beacon-derived ice drift.

**3.2 Two- and three-particle dispersion and sea ice deformation**

Two-particle dispersion monitors the separation in a pair of particles/beacons/ice floes and is defined as

$$R^2 = \langle (x_i(t) - x_{i+1}(t) - \langle x_i - x_{i+1} \rangle)^2 \rangle$$

for adjacent particle pairs $x_i$ and $x_{i+1}$, and where angle brackets again denote ensemble averaging. In contrast to single-particle dispersion, two-particle dispersion reflects the behaviour of spatial gradients in the ice drift field rather than the ice drift field itself. Dynamical regimes associated with velocity gradients are defined according to the relation $R^2 \sim t^\beta$. In short- and long-time limits, particles experience linear

displacements, and approach behaviour comparable to single particles as the pairs lose memory of their origin, respectively. For intermediate times, $R^2 \sim t^3$, in what is referred to as Richardson's (1926) law, resulting from the assumption that eddy diffusivities are dependent on inter-particle separation. In the context of sea ice dynamics, two-particle dispersion characterizes intermittency and heterogeneity in the ice drift field. Exponential $R^2 \sim \exp(ct)$ versus algebraic $R^2 \sim t^\beta$ growth further distinguishes hyperbolic, or

non-local, from elliptic, or local, behaviour (LaCasce, 2008). Compressible flow fields, as is the case for sea ice, can suppress separation between particle pairs, resulting in clustering, or trapping of particle trajectories in "potential wells", while time dependence can result in enhanced transport and escape from these potential wells (Falkovich et al., 2001).

In consideration of three-particle dispersion, triplet areas were computed from recorded beacon

latitude/longitude coordinates using Heron's formula $A = \sqrt{s(s-a)(s-b)(s-c)}$, where a, b, and c denote the length of the sides for each triplet, and $s = \frac{1}{2}(a + b + c)$. Error propagation analysis for the triangle area and triplet evolution according to Heron's formula yields initial error estimates on the order of

$\delta_A = \frac{\delta x}{\sqrt{8A}} \sqrt{(b^2 + c^2 - a^2)^2 a^2 + (a^2 + c^2 - b^2)^2 b^2 + (a^2 + b^2 - c^2)^2 c^2} \sim 0.05,\ 0.12,\ 0.04,$ and $0.04$

km$^2$ for triplets A to D, respectively.

An assessment of the time rate of change in triplet area provides insight about sea ice deformation, namely the differential kinematic parameters (DKPs) of divergence (D), vorticity (V), shearing (S) and stretching (N) deformation rates. In particular, the change in area of a triangular configuration or triplet of drifters to estimate the divergence and local change in flow can be expressed as

$$D = \frac{1}{A}\frac{dA}{dt} = \frac{\partial u}{\partial x} + \frac{\partial v}{\partial y},$$

where $A$ denotes the triangle area, and $u$ and $v$ depict the zonal and meridional components of ocean

circulation or ice drift (following Molinari and Kirwan, 1975; LaCasce, 2008; Wadhams, 1989) with negative values corresponding to convergence. Similarly gradients in sea ice motion or deformation characteristics such as vorticity, shearing and stretching deformation rates can be computed from changes in the triplet area through rotation of the velocity vectors (Saucier, 1955). Comparable expressions and their relations are provided both from an oceanic perspective (Saucier, 1955; Molinari and Kirwan, 1975), and in

an assessment of sea ice deformation in the Weddell Sea in Wadhams (1989) such that:

$$V = \frac{\partial v}{\partial x} - \frac{\partial u}{\partial y} = \frac{1}{A'}\frac{dA'}{dt}; u' = v \ and \ v' = -u$$

$$S = \frac{\partial u}{\partial y} + \frac{\partial v}{\partial x} = \frac{1}{A''}\frac{dA''}{dt}; u'' = v \ and \ v'' = u$$

$$N = \frac{\partial u}{\partial x} - \frac{\partial v}{\partial y} = \frac{1}{A'''}\frac{dA'''}{dt}; u''' = u \ and \ v''' = -v$$

where primes indicate 90° clockwise rotation of velocity vectors. Divergence is associated with a change in area, vorticity with a change in orientation, and shear and stretching with a change in triangle shape due to distortion (Table 2). According to the error estimates for triplet area, a threshold value for significant DKPs relative to uncertainties to ensure a sufficiently large signal to noise ratio is on the order of $10^{-6}$ s$^{-1}$

for the daily timescales considered.

From the perspective of physical changes in sea ice, divergence (convergence) captures opening (closing) in the ice cover related to ice-ocean interactions and flux exchange (ridging). In the Arctic, negative (positive) vorticity depicts anticyclonic (cyclonic) circulation associated with surface winds and inertial oscillations. Negative (positive) shear captures a shape change whereby the northern triangle

beacons travel west (east) relative to the southern pair without changing the triangle area. A negative stretching deformation rate (hereinafter referred to as stretching) indicates stretching along the y-axis (north-south), and shrinking along the x-axis (east-west), without changing the triangle orientation (i.e. stretching parallel or perpendicular to the coast).

In non-divergent flow, the triplet area is conserved so that expansion in one direction is accompanied

by contraction in another direction and the triangle becomes an elongated filament (Prinsenberg et al., 1998). Changes in the aspect ratio (defined as the longest leg, or base, divided by the height) also describe changes in the triplet area; i.e., increasing values indicate elongation of the triplet and filamentation or stretching of the triangular configuration, while decreasing values indicate an equilateral configuration. An equilateral triangle is depicted by a base-to-height ratio of $\frac{2}{\sqrt{3}} \sim 1.155$. Furthermore, the perimeter-to-area

ratio provides a signature of 'folding' in the sea ice drift cover in a manner similar to mechanical annealing whereby compression reduces the dislocation density of materials (Shan et al., 2007; Lawrence Berkeley National Laboratory, 2008). Perimeter-to-area ratios may also provide insight about floe shape and size distributions (Gherardi and Lagomarsino, 2015). Elongated triangles are captured by vanishing perimeter-to-area ratios, while an equilateral configuration is depicted by a perimeter-to-area ratio of $\sim 4\sqrt{3}/a$, where $a$

is the length of the equilateral triangle side.

Previous studies of DKPs using the triangle area approach have shown that the role of triplet areas in describing DKPs resides in the evolution in the time rate of change in the triangle area (Saucier, 1955; Molinari and Kirwan, 1975). If the lengths of the base (defined as the longest triangle side) and height

(defined as the perpendicular distance and 2A/b) differ by an order of magnitude so that the triangle is significantly distorted, a decrease in area will occur. If in addition the change in area exceeds its uncertainty, the DKP associated with the relevant rotation of coordinates will increase, providing a signature of strong deformation. If, however, little change in triplet area is observed (less than the area

uncertainty ~ 0.12 km$^2$), the DKP in question will essentially vanish. In the present study, as is noted below, minimum values for the triplet area amongst all triplets are on the order of 1 km. Previous studies have further quantified conditions for which a triangular configuration does not provide an accurate estimate of DKPs (Righi and Strub, 2001; Ohlmann et al., 2017). In these studies scatter plots for divergence residuals and the base-to-height ratios were used to identify a threshold value for b/h beyond

which DKPs cannot be accurately assessed. A similar criterion is applied in the present analysis for divergence for all four triplets. Specifically, an evaluation of the conditions for which three-particle statistics provide a reasonable estimate of DKPs (Figure 9) following Ohlmann et al. (2017) demonstrates that intervals 14 – 22 September, 2009 for all triplets, 11 – 17 October, 2009 and 19 – 22 October, 2009 for triplet B, and following early November for triplet C be excluded from DKP assessments due to elongation

and distortion in the triangle area.

Relative contributions of the DKPs are monitored using total deformation $D^2 + S^2 + N^2$ to assess distortion in the ice cover due to divergence, and the shearing and stretching deformation rates, as well as the vorticity squared $V^2$ to assess the rotational component (capturing influence from winds and/or inertial oscillations (Gimbert et al, 2012, albeit on shorter timescales)). The Okubo-Weiss (OW) criterion, defined

as (Okubo, 1970; Weiss, 1991)

$$OW = Re(\frac{1}{4}\left(D^2 + S^2 + N^2 - V^2 + |D|\sqrt{S^2 + N^2 - V^2}\right),$$

highlights relative contributions from deformation and the rotational component. Values with OW < 0 (OW > 0) indicate flow dominated by vorticity (deformation). In order to further distinguish relative contributions from divergence, shearing and stretching deformation rates to the total deformation, the shear to divergence ratio is evaluated such that

$$\theta = arctan\left(\frac{\sqrt{S^2 + N^2}}{D}\right),$$

as a signature of sea ice strength (Erlingsson, 1988; Timco and Weeks, 2010). The shear-to-divergence ratio demonstrates spatial and temporal variability in DKPs. Values of 0, 45, 90, 135, and 180 degrees depict divergence, uniaxial extension, shear, uniaxial contraction, and convergence, respectively (Wilchinsky and Feltham, 2006; Feltham, 2008; Fossen, 2016). This ratio further characterizes ice mechanical properties by indicating which (tensile, flexural, shear, or compressive) strength regimes are

encountered by the triplet array (Timco and Weeks, 2010).

## 4 Results

In this section, we identify directional changes in sea ice drift and corresponding sea ice deformation characteristics using Lagrangian dispersion statistics. Two results in particular are emphasized in conjunction with triplet diagnostics relative to the distance from a coastline. First, single-particle dispersion is investigated with respect to ice and external forces contributing to changes in sea ice drift direction. Second, sea ice deformation is examined using two- and three-particle dispersion.

### 4.1 Single-particle dispersion identifies directional changes in sea ice drift

From September to November, 2009, eight regional-scale drift-direction changes are visually seen in the drift trajectory of centroid positions (Figure 1 and Table 1) from each of the four triplet arrays (A through D). These changes correspond to minima and maxima of the running variance from 3-day running means of centroid positions (Figure 2). For all triplets, these same time periods of directional change also match zonal and meridional inflection points. As a result, we identify these inflection points in zonal and meridional dispersion statistics near 12, 19, 25, and 30 September, and 5, 12, 18, and 29 October, which correspond to drift trajectory changes *e1* to *e8*, respectively (Figure 3).

With respect to external forcing connections, we focus first on one case study to understand the connections and then generalize afterward. We select the most dramatic change which occurs on 8 October by looking at the centroid drift minima variances (Figure 2), inflection points in regional absolute dispersion (Figure 3), and minima in SLP (Figure 5) all of which correspond plus or minus one day (within a 3-day running mean) from the beginning of the time series (9 September) until 8 October. At this time there is an Ekman convergence caused by an SLP high with strong off-shore ice drift and a deterioration of the ice cover (reduced ice area − Figure 7). From 8 October onward the SLP and drift centroid variance maxima are out of phase by approximately two to three days which indicates a loss of synchronicity in ice-atmosphere interactions. Furthermore, total (i.e., zonal and meridional components combined in quadrature) single-particle dispersion statistics for an ensemble including triplets A to D, to provide a regional characterization of sea ice drift (Figure 3), is governed by zonal dispersion prior to, with increased meridional dispersion contributions following, early October. Recalling that the scaling exponent $\alpha$ describes sea ice dynamical regimes, total dispersion also shows that $\alpha \sim 3$ before 8 October, which indicates a superdiffusive regime. After 8 October, $\alpha \sim 2$, which indicates a ballistic regime characteristic of linear root mean square displacements in particle paths (Figure 3). Turning angles (Figure 5) further highlight loss of synchronicity in ice-atmosphere interactions and the ice cover following the 8 October SLP high event, evident in phase differences of opposite sign between triplets B, C, and D, and more

distinctive differences between all triplet turning angles. Hence, with this case study we see both a loss of synchronicity in ice-atmosphere interactions evident in turning angles, and a transition in sea ice dynamical regimes captured by single-particle dispersion for the ensemble of triplets.

With the single case study above now clarifying regional dispersion regimes from super-diffusive ($\alpha_R \sim$ 3) to ballistic ($\alpha_R \sim 2$), we examine absolute dispersion for an ensemble of three beacons comprising each triplet to provide a local characterization of sea ice drift (Figure 4). Triplets A and B are characterized by sub-diffusive ($\alpha_L < 1$) behaviour from 19 September, 2009 to 6 October, 2009, and from $17 - 21$, October, 2009, and super-diffusive ($\alpha_L > 2$) scaling from $7 - 12$ October, 2009, and following $22^{nd}$ October (Figure 4b). Triplet C, located further from the coastline, experiences similar sub-diffusive behaviour prior to 6 October, 2009, and predominantly super-diffusive behaviour following 12 October, 2009. Triplet D, located furthest from the continental coastline experiences the smallest displacements, with sub-diffusive scaling to 5 October, 2009, and diffusive scaling following 19 October, 2009, with no instances of ballistic scaling. As is explored further below, these dynamical regimes correspond to changes in atmospheric and sea ice conditions.

In terms of external forcing connections, regional sea ice dispersion regimes appear to transition during a SLP maximum/high, accompanied by an Ekman convergence, and subsequently an offshore ice drift (Figures 3, 5, and 6). Furthermore, ice concentration plays an important role as follows. When the ice concentration is high (>80%), meridional drift speeds away from the coast are suppressed when high SLP sets up offshore Ekman drift (Figures 5, 6, and 7). However when ice concentration is low (<80%) offshore SLP-induced Ekman drift increases due to a reduction in ice-ice frictional interaction (the so-called free-drift state). Enhanced ice drift is in addition observed following the 8 October SLP event in a manner consistent with a regional shift in dynamical regimes captured by the transition from $\alpha_R \sim 3$ to $\alpha_R \sim 2$ in regional absolute dispersion results (Figures 3 and 6). Correspondence between local sea ice drift and concentration is further reflected in local total absolute dispersion (Figures 4 and 7). In particular, subdiffusive scaling ($\alpha_L < 1$) captures limited beacon displacements in high (> 95%) ice concentration regimes, while superdiffusive scaling ($\alpha_L > 1$) captures enhanced transport for lower ice concentrations (< 95%), with the exception of triplet D following the 8 October event. In this case, triplet D experiences subdiffusive behaviour indicative of trapping or an obstruction even in lower ice concentrations, during which time local differences in ice drift and winds compared to triplets B and C are also observed. Comparison of time series between ice drift and surface winds also show that *e1, e3, e4, e6,* and *e8* are accompanied by meridional reversals in surface winds (i.e. southerly to northerly for *e1, e4,* and *e8*, and northerly to southerly for *e3* and *e6*), whereas *e2, e5,* and *e7* are accompanied by persistent northerly winds (Figure 6). Results from this analysis show that meridional changes in surface winds associated with

directional changes in ice drift are captured by inflection points in regional total absolute dispersion; the transition in slopes from $\alpha_R \sim 3$ to $\alpha_R \sim 2$ documents enhanced meridional transport and deterioration in the ice cover following the 8 October SLP high. Local connections between sea ice drift and concentrations are captured in slopes for local total absolute dispersion, with $\alpha_L < 1$ subdiffusive (($\alpha_L > 1$ superdiffusive) regimes indicating reduced (enhanced) local ice dispersion and transport.

## 4.2 Two- and three-particle dispersion and sea ice deformation

In this section we explore interactions beyond the sea ice drift by looking at the differential drift or deformation between beacons in each array. In the context of two- and three-particle dispersion, we want to understand the relationships between the sea ice cover and the atmosphere relative to distance from the coastline.

With respect to sea ice response to external forcing as manifested in sea ice deformation, we examine the 8 October case study for two-particle (relative) and three-particle dispersion. Relative (two-particle) dispersion shows that total dispersion is initially (19 September to 5 October) governed by zonal separation for triplets A, B, and C, and by both zonal and meridional separation for triplet D (Figure 9). This distinction may be attributed to predominantly meridional motion (and along-shear transport) of triplet D along the eastern segment of the anticyclonic Beaufort Gyre. A significant decrease in zonal separation is observed closest to the coastline in triplet B due to convergence in the ice cover in response to the 8 October SLP high. Triplet B subsequently experiences comparable zonal and meridional separations (right panel in Figure 9) until 26 October, following which enhanced zonal separations are observed as beacons encounter lower ice concentrations (Figures 7 and 9). Noteworthy also is a transition in two-particle dispersion scaling law values from $\beta < 1$ to $\beta > 1$ prior to and following 8 October, respectively.

We now examine relative dispersion during directional change events (Figure 9). Whereas events *e1, e3, e4, e6, and e8* are associated with reversals in surface winds, changes in along-shear (zonal for triplets A, B, and C, meridional for triplet D) and cross-shear (meridional for triplets A, B, and C, zonal for triplet D) separation accompany northerly winds during events *e2, e5, e7*, with delayed responses following the SLP high on 8 October. During *e2*, an increase in along-shear transport is observed for all triplets, evident in an increase in zonal separations for triplets A, B, and C in the direction of transport along the southern portion of the anticyclonic Beaufort Gyre, and in meridional separations for triplet D in the direction of motion along the eastern segment of the BG. During *e5*, an onset in an increase in across-shear transport, namely meridional separation, accompanied by a decrease in zonal separation is observed for triplet B, which is sustained during the SLP high. Following *e7*, a delayed increase in along-shear transport

and zonal separation for all triplets is observed. These results highlight contributions from along- and across shear transport to sea ice drift in response to atmospheric forcing.

Evaluation of triangle areas and differential kinematic parameters (DKPs) using three-particle dispersion further highlights changes in the ice cover prior to and following the 8 October SLP high (Figure 10). As is noted in the Methods section, DKPs are evaluated for time intervals excluding 14 – 22 September (all triplets), 11 – 17 October and 19 – 22 October (triplet B) based on the analysis presented in Figure 9 monitoring triangle elongation and conditions for which triangles provide a reasonable estimate of DKPs. Regional differences in triplet area representative of sea ice deformation are observed in the evolution of all triplet triangles (Figure 10). Triplet area evolution demonstrates enhanced variability in triplet B relative to triplets A, C, and D, particularly following 8 October. An increase in triangle base (defined as the longest triangle side) is observed with decreasing distance from the coastline, in a manner consistent with local absolute dispersion (Figure 4). Base-to-height and perimeter to area ratios illustrate an equilateral configuration near the ice edge and coastline in the early stages of evolution for triplet A, and late stages of evolution for triplet B, captured in base-to-height ratio values approaching 1.155. Stretching is observed closest to the coastline following 8 October within a consolidated ice regime (Triplet B), captured in increasing perimeter-to-area and base-to-height ratios.

As for two-particle dispersion, changes in area, height, and P/A and b/h ratios capture deformation associated with events *e2, e5*, and *e7*, with delayed responses to northerly winds following the 8 October SLP high (Figure 10). Following *e2*, the base for triplets A and B increases and height decreases during consolidation, resulting in enhanced P/A and b/h. By contrast, P/A and b/h are approximately constant for triplets C and D, indicating that the triangle maintains its shape further from the continental coastline. Following *e5* and the SLP high, the greatest stretching (high base and ratio values) is observed in triplet B located closest to the coastline. Several days following *e7* a decrease in area for triplet B is associated with a decrease in height so that b/h and P/A increase, while once again the area and ratios for triplet C and D remain constant.

Local differences in sea ice deformation are further reflected in differential kinematic parameters (DKPs) defined as the weighted time rate of change in triplet area, as described in the methods section (Figure 11). Results show that sea ice deformation is approximately an order of magnitude smaller for triplets furthest from the coastline ($\sim 10^{-5}$ for triplet A versus $\sim 10^{-4}$ for triplet B). A strong shear event is in addition observed for triplet B during the SLP high on 8 October, and for triplet C on 10 October. In consideration of directional changes due to northerly winds e5 and e7, during *e5*, weak DKPs are observed for triplet C, with triplets B and D governed by vorticity. During *e7* convergence and divergence dominate for triplet B.

Relative DKPs highlight relative contributions from external forcing associated with winds and distance from the coastline as manifested in vorticity and total deformation. Total deformation is shown to decrease with increasing distance from the coastline (top panel, Figure 12), particularly following the SLP high on 8 October. The Okubo-Weiss criterion results show enhanced distortion in the (bounded) ice cover closest to the coastline, especially following the 8 October SLP maximum. Deformation dominates for all triplets for two weeks following the 8 October maximum (OW > 0 from 8 to 22 October). Theta values illustrate changes in relative contributions from divergence and the total strain rate $(S^2 + N^2)$. Noteworthy are the high $(\theta \sim 180)$ values during *e2, e7,* and following *e5*, indicating convergence associated with offshore ice drift due to Ekman convergence and sea ice deformation associated with along- and across-shear transport. Results from three-particle dispersion, namely triangle areas, DKPs, and relative DKPs thus show that sea ice is governed by deformation following the 8 October SLP event associated with deterioration and loss of connectivity in the ice cover that increases with decreasing distance from the coastline, with dominant contributions from sea ice convergence associated with Ekman convergence and persistent northerly local winds and subsequent failure in the ice cover for triplets B and to a lesser extent triplet C.

## 5 Discussion

A framework based on scaling laws and metrics/diagnostics derived from single-, two-, and three-particle dispersion has been presented in this study to quantify sea ice drift and deformation in response to atmospheric forcing for varying distances from the coastline. In single-particle dispersion the scaling exponent $\alpha$ provides a signature of flow topology and organized structure in the ice drift field (Table 1). Single-particle dispersion analyses in the present investigation show that $\alpha_R$ (for single-particle dispersion defined according to an ensemble comprised of triplets A to D) in a regional characterization provides a signature of regional changes in ice dynamical regimes. This is evident in a transition from $\alpha_R \sim 3$ to $\alpha_R \sim 2$ that indicates a suppression in transport associated with deterioration in the ice cover, enhanced meridional (cross-shear) ice drift closest to the coastline that suppresses along-shear transport, and increased ice-ice interactions following the 8 October SLP high. The scaling exponent $\alpha_L$ (for single-particle dispersion defined according to an ensemble comprised of beacons within each triplet) in a local characterization provides a signature of local changes in ice drift and state, namely "trapping" events in higher ice-concentrations and "flight" events in lower ice concentrations.

Previous studies have shown, as noted in Section 3.1, that single-particle dispersion provides a signature of super-diffusive Lévy dynamics describing "trapping" and "flight" events (Shlesinger et al., 1993; Solomon et al., 1994). Results from this analysis suggest that a Lévy walk (or fractal space random

walk) interpretation may be an appropriate description for directional changes in sea ice drift according to local and to a lesser extent regional absolute dispersion statistics. In particular, super-diffusive behaviour can provide a signature of long-range correlations associated with along-shear transport (in this case zonal in the southern and meridional in the eastern branch of the anticyclonic Beaufort Gyre). Sub-diffusive behaviour can capture interruptions/disruptions in organized flow due to ice-ice interactions (captured by local dispersion), cross-shear transport, or trapping (elliptic) dynamical regimes (captured by regional dispersion). Inflection points indicate transitions between these dynamical regimes.

Relative (two-particle) dispersion provides a signature of changes in along- and cross-shear transport in response to atmospheric and ice conditions (Table 2). Specifically, two-particle dispersion characterizes sea ice deformation, and local changes in the ice cover. Enhanced meridional separation (cross-shear transport) closest to the coastline and near the ice edge (triplet B) following the 8 October SLP high is captured by a transition in meridional, zonal, and total two-particle dispersion. Furthermore, $t^3$ scaling associated with Richardson's scaling law and attributed to atmospheric dispersion as described in Rampal et al. (2009), is evident ($\beta \sim 3.4$) for triplet C from 10 – 17 October; higher (exponential and algebraic) scaling exponents exist for triplets B and D following the SLP high on 8 October. Exponential growth for triplet B indicates the existence of a single-timescale associated with non-local behaviour and unimpeded westward advection due to anticyclonic circulation in lower ice concentrations (reduced frictional effects in lower sea ice concentrations).

As noted in the Introduction, previous studies of sea ice dynamics in the Beaufort Sea have demonstrated that the sea ice cover moves as an: aggregate for spatial scales exceeding 100 km; aggregate for an elliptic (homogeneous) regime or discrete entity for a hyperbolic (discrete) regime for scales ranging from several to 100 km; and is governed by ice-ice interactions for scales on the order of 1 km (Overland et al., 1995). Two-particle results from the present investigation show that meridional separations range from ~10 to ~30 km, and zonal separations from ~10 km to spatial scales exceeding 100 km (triplet B) (Figure 8). The total (meridional and zonal combined in quadrature) two-particle scaling exponent β indicates that the ice cover in this study experiences both regimes (i.e. moves as an aggregate and discrete entity), evident in a transition in the scaling exponent from β < 1 to β > 1 following the 8 October SLP high, thus providing a measure of connectivity in the sea ice cover. Furthermore, β ≥ 3 for triplets C and D in high ice-concentrations, and β ~ exp($t$) for triplet B in low ice concentrations in late October, which further distinguishes elliptic (local) from hyperbolic (non-local) behaviour.

Deformation components, or differential kinematic parameters (DKPs) are computed using three-particle dispersion and changes in triangle area (Table 2). Results from an evaluation of the Okubo-Weiss criterion, which monitors relative contributions from total deformation (divergence and shear combined in

quadrature) and vorticity, show enhanced deformation (OW > 0) associated with loss of connectivity (captured by a transition in two-particle dispersion scaling exponent from $\beta < 1$ to $\beta > 1$ following the 8 October SLP high). Deformation was also found to be 100 times larger near the coast, with vanishing values for triplets C and D indicating weak deformation further from the coastline. Evaluation of individual directional change events in the context of three particle dispersion also showed that θ, in enabling additional distinction between divergence, convergence, and shear to total deformation strain states in particular, namely uniaxial extension (θ ~ 45), shear (θ ~ 90), and uniaxial compression (θ ~ 135), can potentially provide a signature of failure in the ice cover, as is described further below.

Previous studies have demonstrated correspondence between ice deformation and stress measurements in the Beaufort Sea (Richter-Menge et al., 2002). Sea ice strength is dependent on ice concentration and thickness. The ice cover fails when the internal ice stress is comparable to ice strength. Previous studies have also shown that uniaxial extension can give rise to ridging for a particular alignment of flows and anisotropic ice cover (Wilchinsky and Feltham, 2006). Failure in the ice cover is in addition attributed to compression within a region of confinement or what is referred to as "trapping" in the present study. Results from this analysis suggest that the three-particle dispersion approach, and shear-to-divergence ratio in particular can be used in combination with ice concentrations and (in future studies) thickness to assess conditions for which failure in the ice cover occurs (Figures 7 and 12). Convergence (θ ~ 180) experienced by triplet C on 26 September within a high ice concentration regime, and subsequent reduction in SIC encountered by triplet beacons suggests local failure in the ice cover. In addition, compression experienced by triplet B during the 8 October SLP high and Ekman convergence coincides with reduced ice concentrations and the strong ice shear event (Figures 7, 11, and 12), also indicating failure in the ice cover for triplets C and D. This deterioration further continues near the ice edge for triplet B to 20 October as shear-to-divergence values on the order of 180 are accompanied by significant reductions in ice concentrations as the triplet encounters the ice edge and free drift conditions. These results also suggest that directional change events *e2, e5* and *e7*, associated with along-shear transport, northerly surface winds and ice deformation anticipate failure in the ice cover closest to the coastline (evident in lower θ values following each event for triplet B), although additional study is required to confirm this hypothesis.

Table 3 synthesizes and describes sea ice dynamical regimes characterized by single- and two-particle scaling exponents α and β, respectively. In the context of sea-ice drift, the regional single-particle dispersion scaling exponent $\alpha_R$ describes flow topology and organized structure in the flow (ice drift) field; the local single-particle dispersion scaling exponent $\alpha_L$ describes local changes in the sea ice state. In the context of sea ice deformation, β provides a signature of spatial gradients in the ice drift field. The sub-

diffusive regime ($\alpha_L$ <1) for separations defined by fixed spatial scales over time ($\beta$ < 1) characterizes trapping associated with ice-ice interactions in high ice concentrations and an ice cover with high connectivity that moves as an aggregate. This behaviour is observed for directional change events *e3* to *e5* with characteristic length scales of 30, 40, and 10 km for triplets A and B, C, and D, respectively.

Following the 8 October SLP high event, triplet C experiences a super-diffusive regime ($\alpha_L$ >1) for increasing separations ($\beta \sim 3$) indicating loss of connectivity in the ice cover, with increases in scale separation from $\sim 40$ to 70 km in ice concentrations ranging from 90-95% also associated with deterioration in the ice cover. By contrast, triplet D located further from the coastline experiences a subdiffusive regime ($\alpha_L < 1$) for increasing separations ($\beta \sim 3.8$), with increases in scale separation from $\sim$10 to 15 km,

indicating limited displacements and sustained trapping for enhanced separations. Triplet B, located closest to the coastline experiences the most rapid increase in separations (from $\sim$ 15 km to 70 km) and limited displacements ($\alpha_L$ <1) within a high ice concentration regime, providing a signature of reduced connectivity, enhanced deformation, and compression associated with ice convergence due to the 8 October SLP high. An exponential scaling law for triplet B indicates sea ice dynamics governed by large-scale

processes and advection along the southern branch of the anticyclonic Beaufort Gyre. In general terms, the upper right corner of Table 3 depicts sea ice dynamics governed by atmospheric processes ($\alpha$, $\beta$ > 1), and the lower left corner depicts sea ice dynamics governed by ice interactions and coastline ($\alpha$, $\beta$ < 1). Sea ice deformation monitored by the Okubo-Weiss criterion and shear-to-divergence ratio illustrates regional sea ice response to external forcing based on distance from the coastline, with higher deformation values closer

to the coastline, and transitions from convergence and shear to divergence providing a local signature of failure in the ice cover. This framework thus provides a prescription for understanding sea ice drift and deformation based on single-, two-, and three-particle dispersion and associated diagnostics.

**6 Conclusions**

In this study we developed a framework to characterize, using single- and two-particle scaling laws and three particle dispersion diagnostics, directional changes in sea ice drift and associated deformation processes in response to atmospheric forcing based on Lagrangian dispersion statistics. We tested this approach using single-, two-, and three-particle dispersion applied to beacon arrays deployed in a triangular configuration as triplets at varying distances from the coastline in the southern Beaufort Sea.

In consideration of our first research question, single-particle dispersion characterizes directional changes in sea ice drift trajectories and dynamical changes in the ice cover. Specifically, single-particle dispersion captures i) a shift in ice dynamical regimes following the 8 October SLP high, ii) inflection

points and directional changes in the meridional ice drift component associated with interactions with the coastline, and iii) loss of synchronicity in ice-atmosphere interactions to provide a regional characterization of sea ice drift. Local differences in scaling, ice drift and state at varying distances from the coastline are captured by single-particle dispersion for beacons associated with each triplet. Single-particle displacements are also shown to decrease with increasing distance from the coastline.

In consideration of our second research question, two- and three-particle dispersion characterize associated deformation processes for varying distances relative to the coastline. In particular, two- and three-particle dispersion capture i) sea ice deformation induced by northerly winds and accompanying along- and cross-shear transport ii) relative DKPs and enhanced deformation with decreasing distance from the coastline following the 8 October event, and iii) local and regional variations in sea ice deformation in the ice cover based on an assessment of the shear-to-divergence ratios.

An assessment of sea ice drift and deformation based on synthesis of single- and two-particle dispersion results, and in particular scaling exponents $\alpha$ and $\beta$, respectively, demonstrates that the single-particle scaling exponent $\alpha$ provides a signature of sea ice drift response to atmospheric forcing ($\alpha_R$), and sea ice state ($\alpha_L$), characterized by diffusive, sub-, and super-diffusive regimes, while the two-particle scaling exponent provides a signature of loss of connectivity in the ice cover associated with enhanced separations, again in response to atmospheric forcing, and based on distance from the coastline. Three-particle dispersion diagnostics including the Okubo-Weiss criterion and shear-to-divergence ratio further characterize DKPs and sea ice deformation. The Okubo-Weiss criterion shows enhanced (two orders of magnitude larger) deformation near the coastline. The shear-to-divergence ratio further distinguishes contributions from convergence, divergence and shear to total deformation and through the identification of strain states (extension, shear, and contraction) provides insight on failure in the ice cover in response to persistent northerly winds.

In terms of impacts and insight addressed in the discussion, Lagrangian dispersion thus provides a consistent framework and prescription for understanding directional changes in sea ice drift and accompanying deformation in response to atmospheric forcing. The diagnostics and tools developed in this study provide a unique characterization of sea ice drift and deformation processes in the southern Beaufort Sea, with implications for ice hazard assessments and forecasting applications. Results from this analysis can be applied to develop an integrated observational-modeling framework for Lagrangian dispersion designed specifically to understand ice-atmosphere interactions in the context of drift and deformation at regional and local spatial scales. Building on diagnostics and insights from this study, new techniques to examine higher-frequency fluctuations associated with inertial oscillations are explored in a companion

paper (Geiger and Lukovich, in preparation). Implications for EVP and rheological characterizations of sea ice using observations, results and diagnostics from this investigation could further be explored.

Proposed work using results from this study includes efforts to address the question: What are the implications of changing ice and atmospheric patterns and dynamics for ice –atmosphere interactions, including heat and momentum exchange in particular, and local and global-scale processes more generally? Answering this question using diagnostics that evaluate underlying structure in atmospheric, sea ice and oceanographic flow fields will be essential in/contribute to understanding, predicting, and addressing climate change impacts within an increasingly unpredictable environmental regime.

## Acknowledgements

Buoy data were funded by the Natural Sciences and Engineering Research Council of Canada, the Canada Foundation for Innovation and partner organizations. Funding for this study was also provided by the Canadian Networks of Centres of Excellence (NCE) program, and Canada Research Chairs (CRC) programme (D.G. Barber). The authors would like to thank D. Babb and R. Galley for triplet deployment and for contributions to an earlier version of this manuscript. This work was completed as a contribution to the ArcticNet and Arctic Science Partnership networks.

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

a)

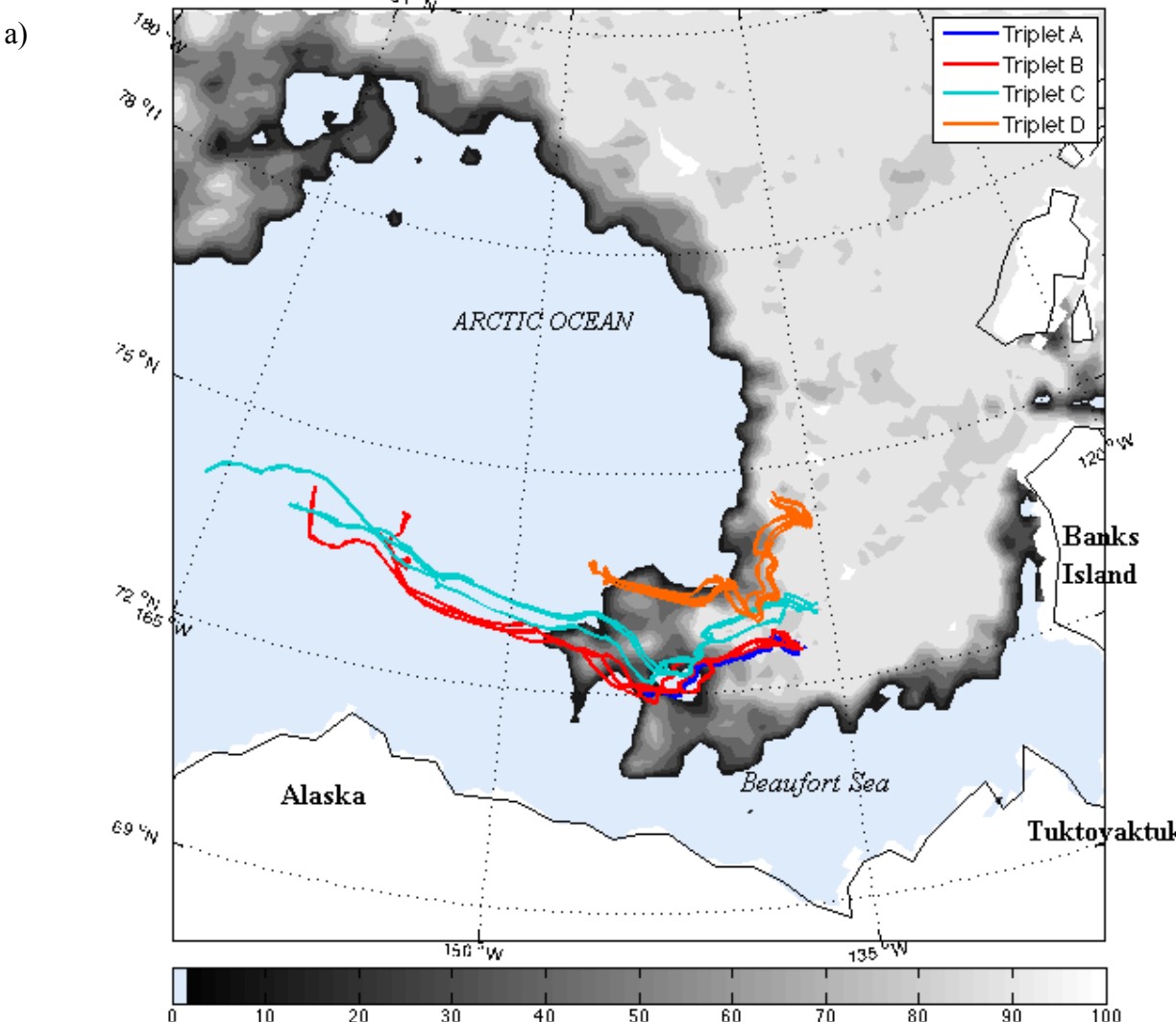

b)

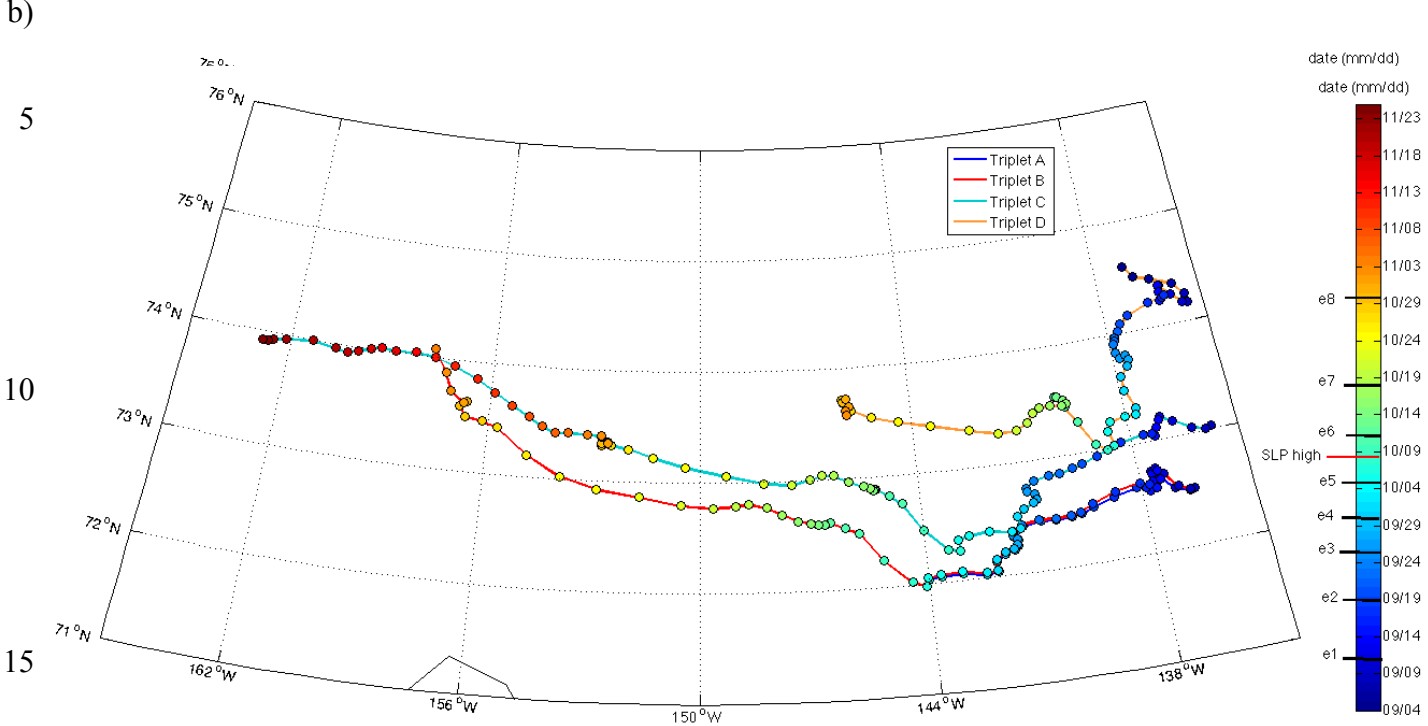

**Figure 1: GPS buoy-array-triplet trajectories relative to sea ice and coastline. Trajectories for beacons deployed near 135 °W between 72 °N and 75 °N capture spatiotemporal evolution in ice beacon triplet centroids beginning in September, 2009, with triplet A located closest to, and triplet D located furthest from, the continental coastline. Triplets A and B, deployed near 72 °N, share two of the three beacons and are advected westwards to approximately 144 °W and 158 °W, surviving until October 7th and November 5th, respectively. Triplet C is deployed near 73 °N and is also advected westward to 162 °W, surviving until November 26th. Triplet D is deployed near 74.5 °N and traverses a shorter path southwards and westwards to 145 °W, surviving until November 4th. a) Evolution in triplet A to D centroid trajectories, superimposed on sea ice concentration map in early September, 2009 on triplet deployment. Triplet D is also initially located nearest the tongue of multiyear ice edge. b) Temporal evolution of triplet centroids A to D, colour-coded by date with each directional change event enumerated in the colourbar timeline. The SLP high resulting in strong offshore ice drift, and decoupling in ice-atmosphere interactions is also depicted.**

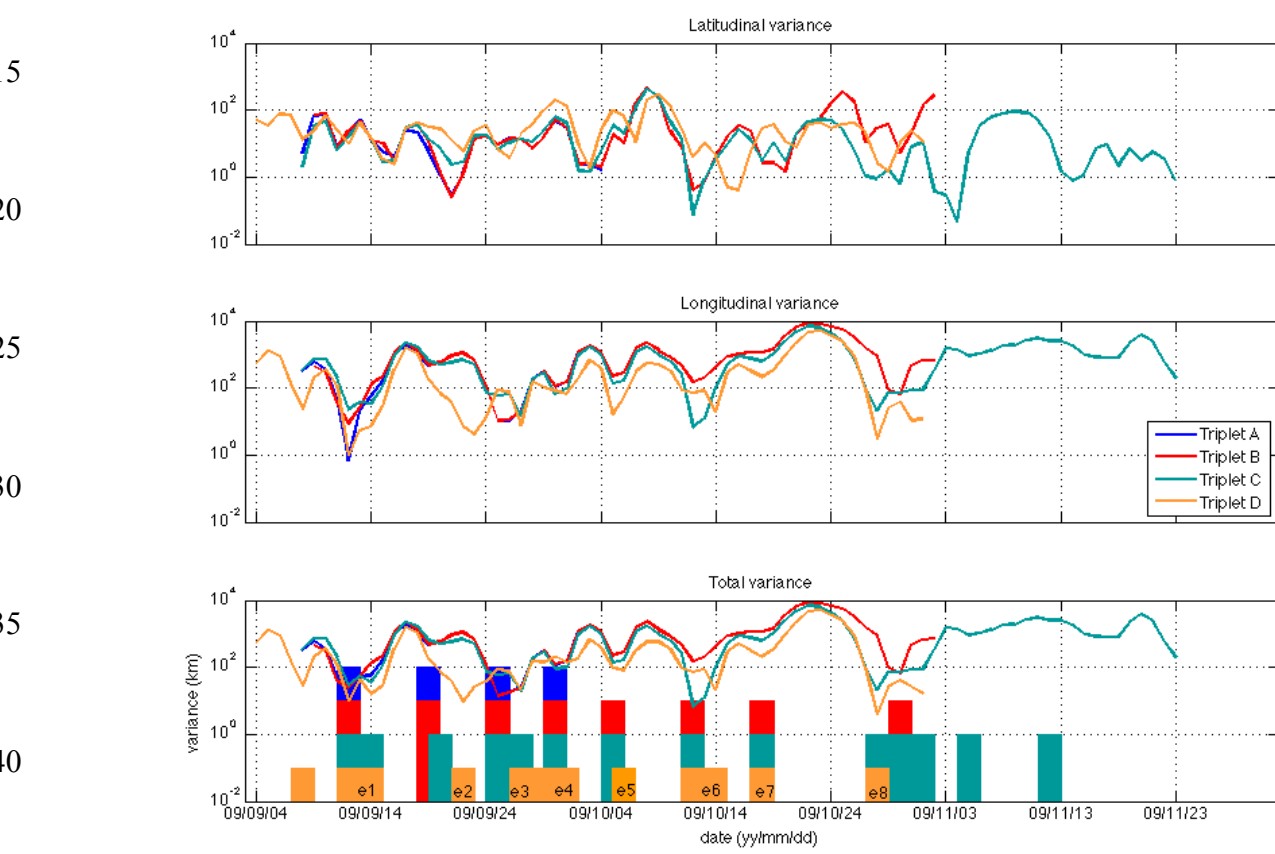

**Figure 2: Directional changes in triplet A to D centroid trajectories, defined as minima in the time series for the total variances of the 3-day running mean triplet centroid positions. The upper, middle, and lower panels depict latitudinal, longitudinal and total variance, respectively. The bars in the bottom panel depict the triplet events associated with minima in the total variance. Enhanced latitudinal variance is observed for Triplet D relative to Triplet A in late September/early October. By contrast longitudinal variance decreases with increasing distance from the coastline, evidenced in lower values for triplet D relative to triplet A. Comparison with mean SLP in the vicinity of the triplets shows correspondence between SLP and drift variance maxima prior to 8 October; minima occur between SLP and variance maxima.**

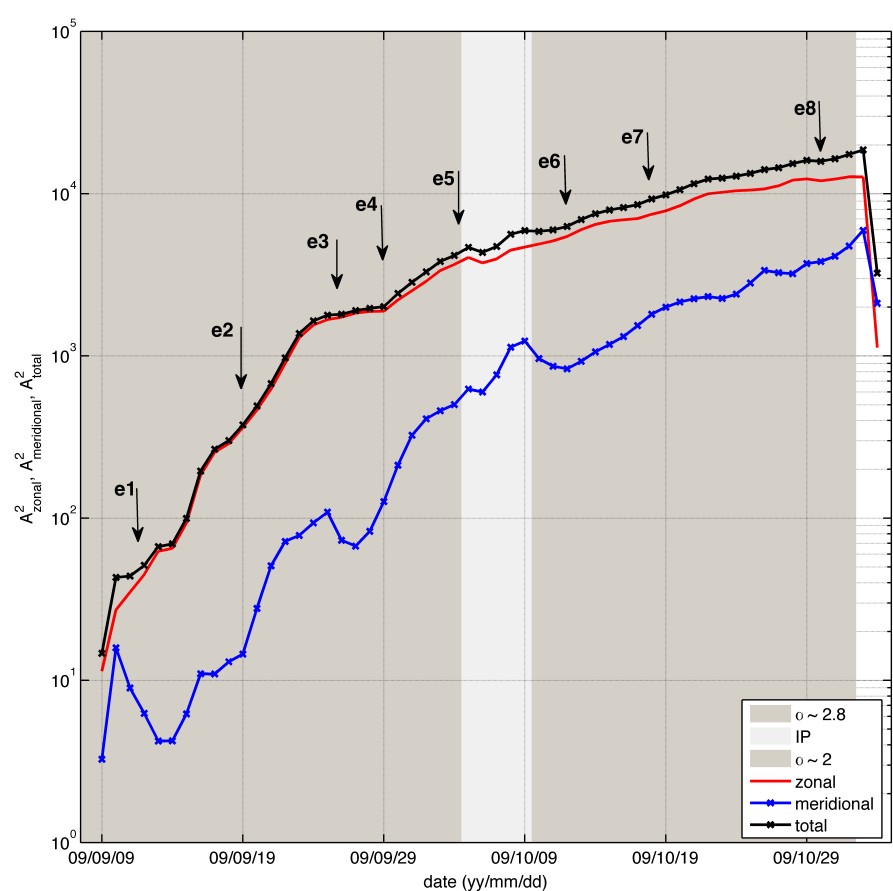

**Figure 3: Absolute (single-particle) dispersion statistics for triplets A to D, depicting zonal (red), meridional (blue), and total (black) dispersion, providing a regional characterization of sea ice drift in the SBS. Arrows and labels depict triplet events *e1 – e8*. Directional change events captured by the (Eulerian) variance interpretation (Figure 2) are also captured by inflection points in the (Lagrangian) single-particle zonal and meridional dispersion analyses near 12, 19, 25, and 30 September, and 5, 12, 18, and 29 October. A plateau in the zonal and meridional single-particle dispersion curves depicts minimal displacements in the direction of motion relative to the mean as the triplet centroids and paths change course.**

**Total (zonal and meridional components combined) single-particle dispersion for triplets A to D provides a regional characterization of changes in sea ice drift. Specifically, total absolute dispersion is governed by zonal dispersion prior to, and meridional dispersion following, early October, with scaling exponents of $\alpha \sim 3$ and 2, respectively, highlighting a transition in ice dynamical regimes. Dark shaded areas depict dynamical regimes associated with $\alpha \sim 3$ and $\alpha \sim 2$ prior to and following the (light-shaded) inflection point (IP) range.**

a)

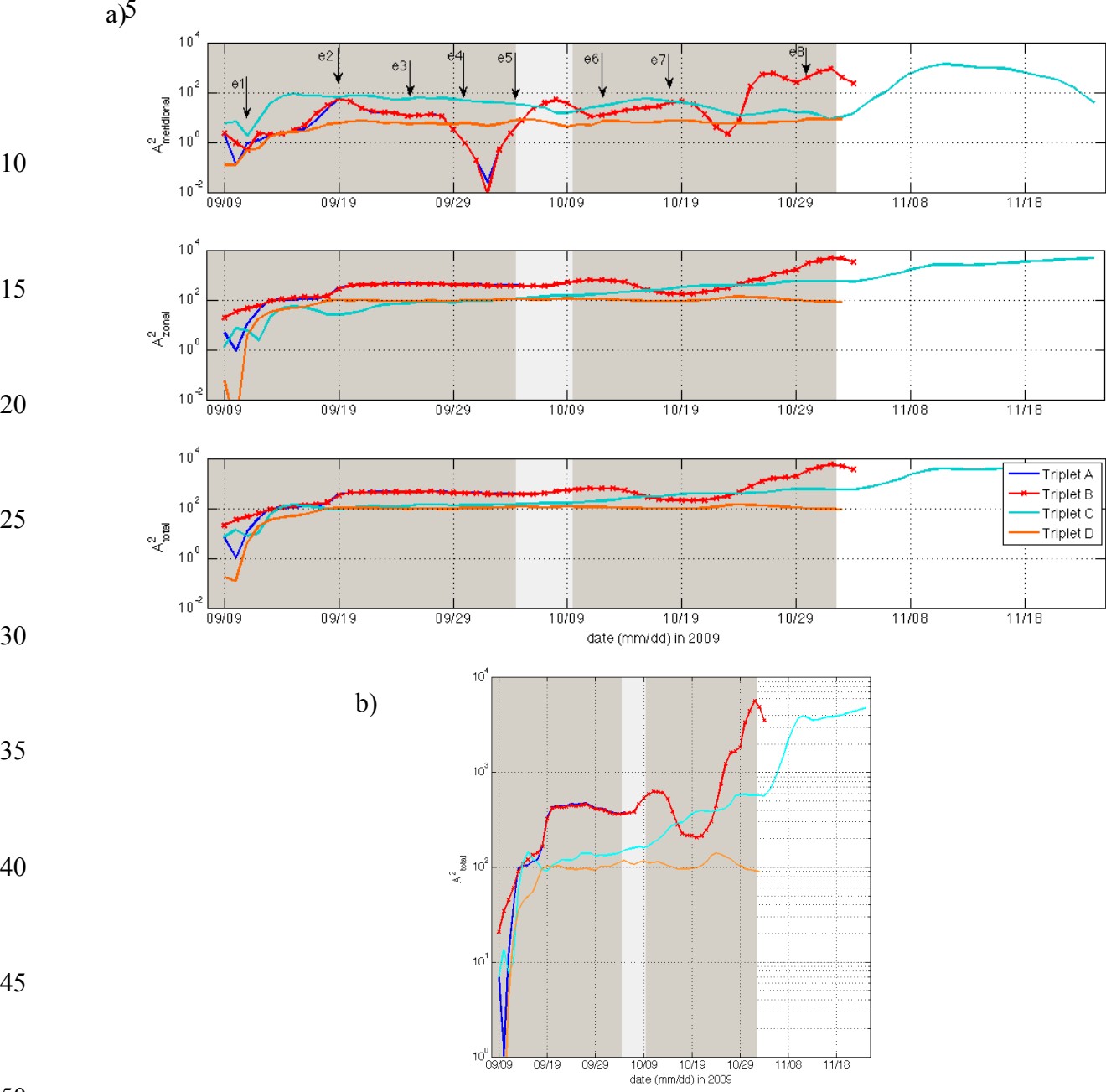

**Figure 4: Absolute (single-particle) dispersion statistics for beacons comprising triplets A (blue), B (red), C (cyan) and D (amber), depicting meridional (top), zonal (middle), and total (lower) dispersion, providing a local characterization of ice drift at varying distances from the coastline. a) Triplets A and B, located closest to the coastline, exhibit considerable meridional variability relative to triplets C and D (upper panel in Figure 4a). Single-particle displacements are also shown to decrease with increasing distance from the coastline. b) Total single-particle dispersion for triplets A (blue), B (red), C (cyan), and D (amber) as an enlargement of lower panel in Figure 4a, capturing temporal evolution in and differences between triplet slopes based on local beacon behaviour associated with each triplet. Total absolute dispersion is governed by zonal dispersion. Furthermore, inflection points in absolute dispersion associated with the local characterization capture transitions in dynamical regimes due to changes in local ice conditions. Shading and arrows are as in Figure 3 providing a regional characterization.**

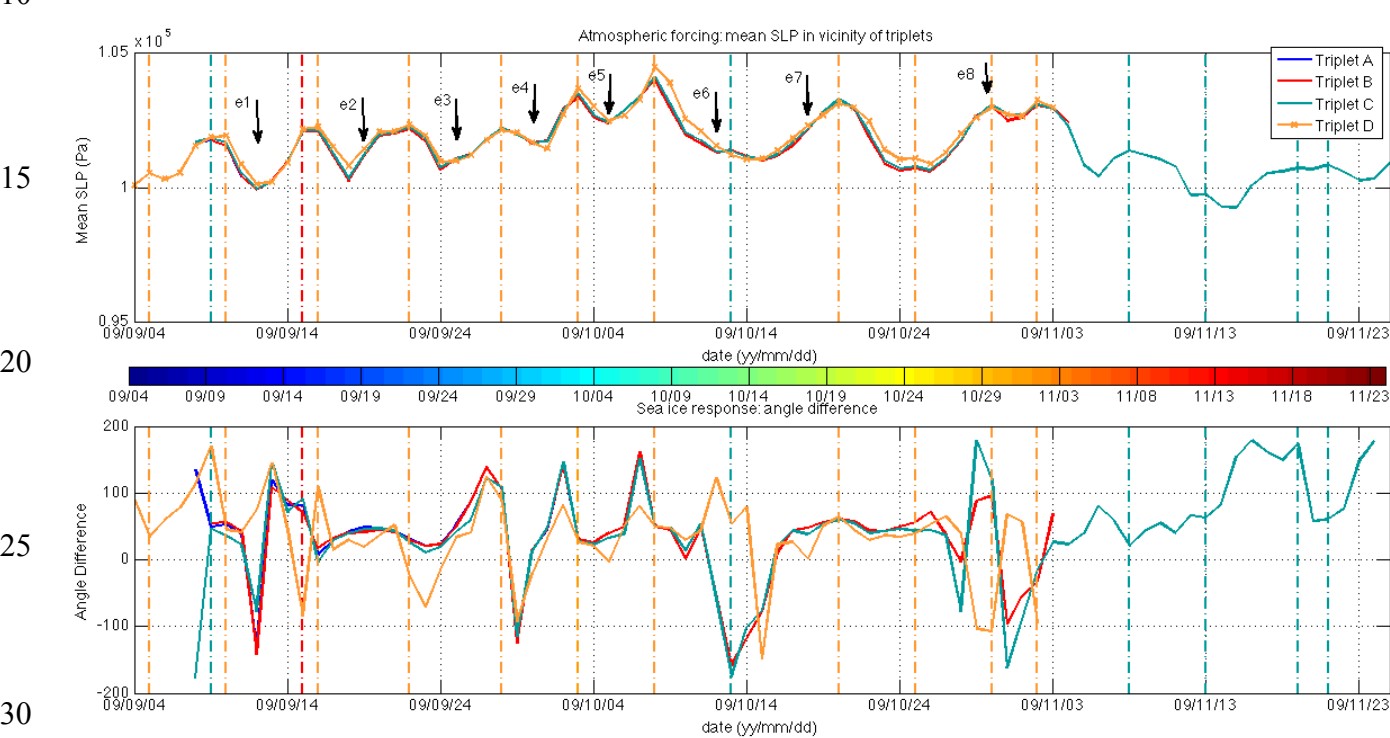

**Figure 5: Atmospheric forcing and sea ice response. Mean SLP in the vicinity of triplets A to D highlighting atmospheric forcing (upper panel). Turning angle between surface winds and triplet centroid drift depicting sea ice response (lower panel). Colour bar indicates colours associated with centroid dates in Figure 1b. Mean SLP in the vicinity of triplets A to D is uniform, with some differences in the vicinity of triplet D. By contrast, turning angles highlight spatial variability in ice drift for intervals between high SLP regimes, and provide an initial indication of sea ice mechanics and deformation. Specifically, differences in turning angles highlight spatial (relative to distance from the coastline) differences in sea ice response to external forcing as the SLP high enters the region in the vicinity of the beacon triplets on 8 October, 2009, and subsequent loss of coherence in the ice cover.**

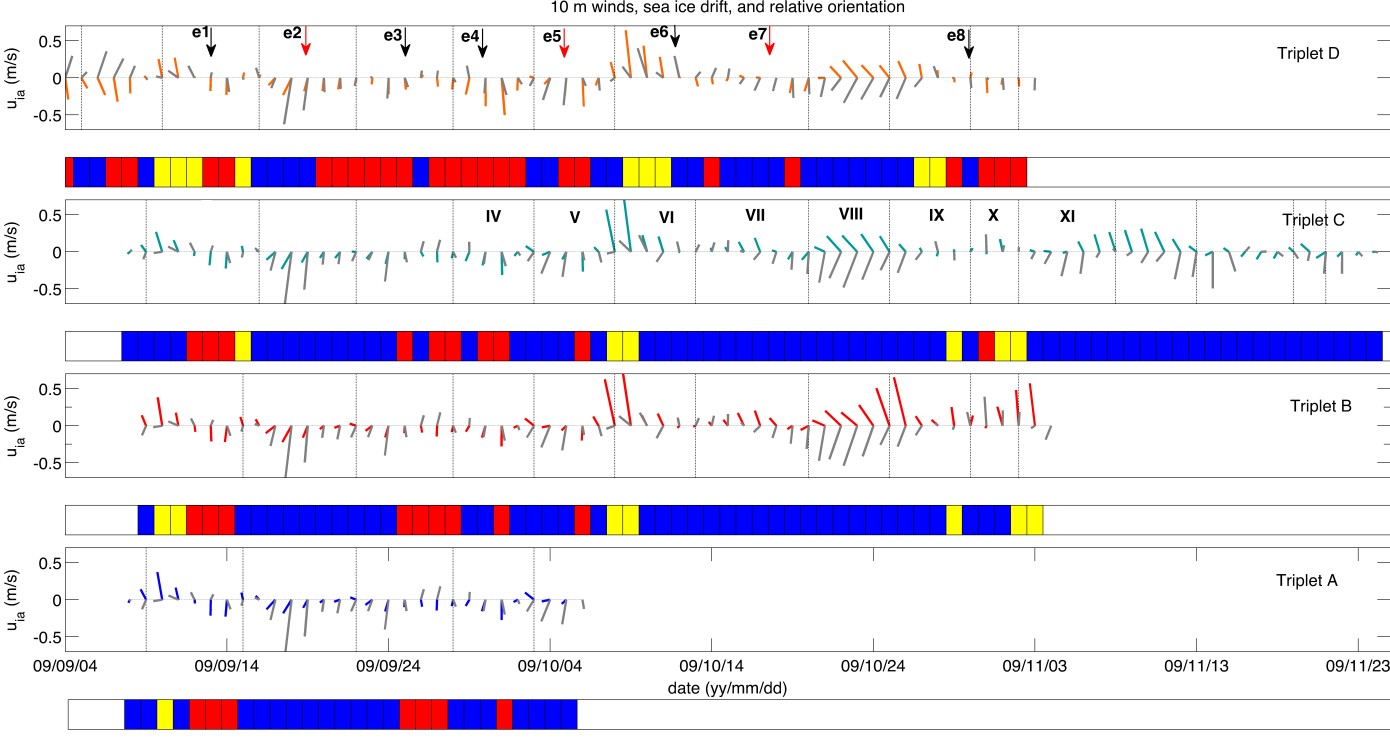

**Figure 6: Winds, sea ice drift and orientation relative to the coastline.** Time series of 10 m North American Regional Reanalysis (NARR) winds in the vicinity of the triplet centroids (grey vectors), sea ice drift for triplets A (lowermost panel) to D (uppermost panel), and offshore (yellow), onshore (red), and alongshore (blue) ice drift orientation. Winds and ice drift show coherence in ice drift for triplets A to C, and variations in triplet D. Orientation highlights increased free drift conditions at higher latitudes associated with triplet D relative to lower latitude triplets. Directional changes *e1*, *e3*, *e4*, *e6* and *e8* occur during reversals in wind and ice drift. Noteworthy also are directional changes in sea ice drift during persistent (~3 – 4 days) northerly winds (events *e2*, *e5* and *e7*). Of particular interest is strong offshore ice drift on 8 October due to easterly winds and strong Ekman convergence during a SLP high within a high ice concentration (> 95%) ice regime. Red arrows associated with directional changes highlight events associated with persistent northerly surface winds.

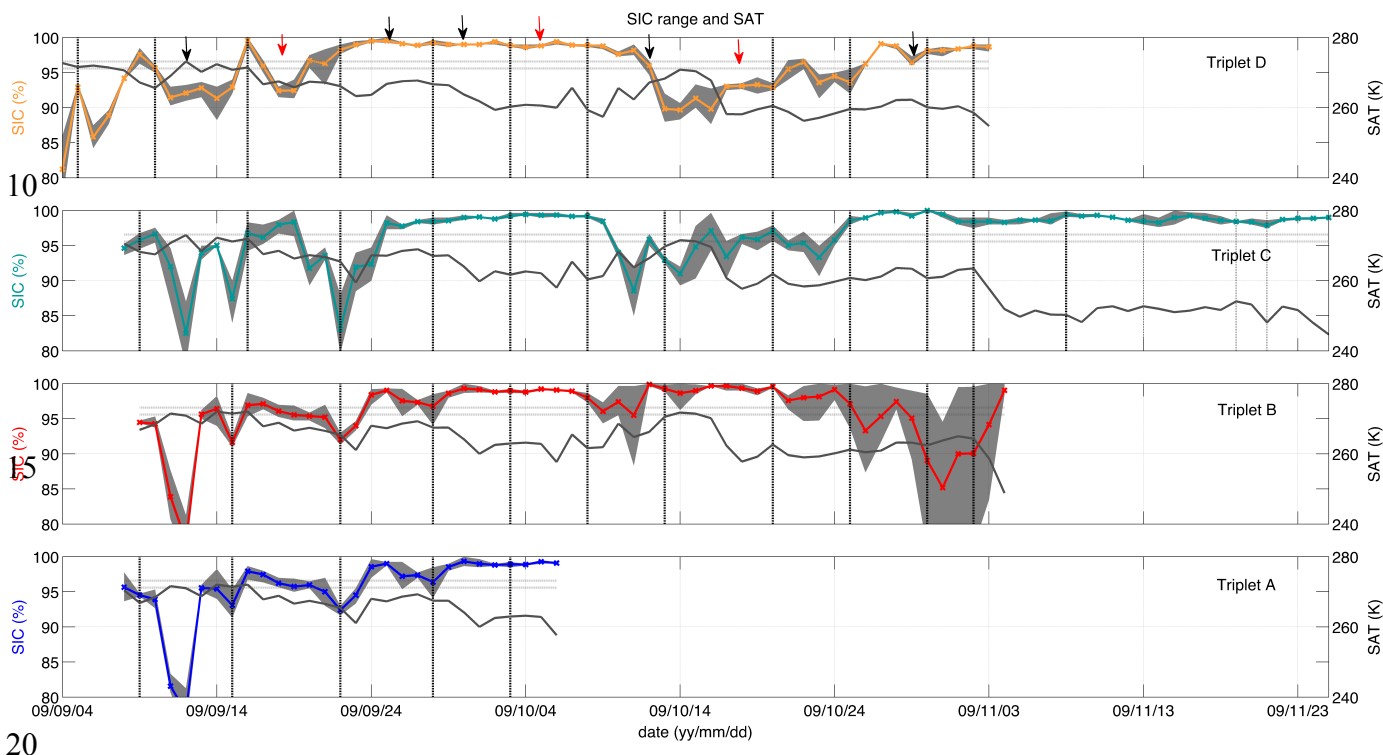

**Figure 7: Sea ice concentration (SIC) range and Surface Air Temperatures (SAT).** Time series of percent sea ice concentration and SAT for triplets A (lowermost panel) to D (uppermost panel). Vertical lines depict dates associated with SLP highs. Grey shading shows the range of ice concentrations encountered by three beacons comprising the triplets. Horizontal lines depict SAT values of 273 K and 275 K. SAT values less than 2 ºC are sustained following 14 September, 2009 for triplets A to D, with an interval of increased SAT near 14 October. SIC varies for triplets A to D, with lower concentrations to 24 September. Low SIC exists during a SLP high for triplet C near 22 September. Lower SICs are observed following the 8 October SLP high indicating deterioration in the sea ice cover.

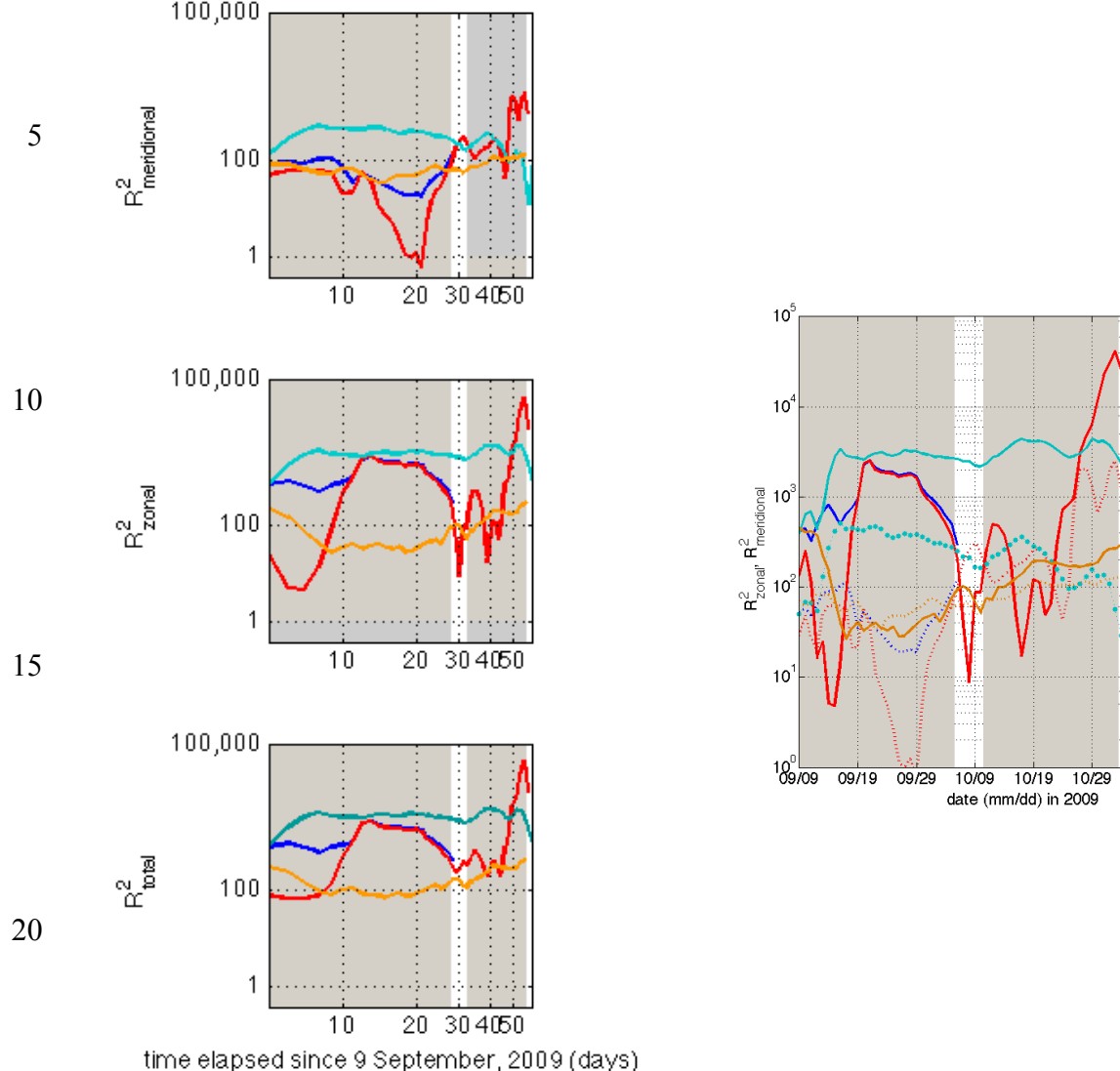

**Figure 8 Relative (two-particle) dispersion showing (left column) meridional (top), zonal (middle), and total (lower panel) relative dispersion as a function of elapsed time since 9 September, 2009, and (right panel) a superposition of zonal (solid line) and meridional (symbol dashed line) relative dispersion. Total dispersion is initially governed by zonal separation for triplets A, B, and C, and by both zonal and meridional separation for triplet D. This distinction may be attributed to predominantly meridional motion (and along-shear transport) of triplet D along the eastern segment of the anticyclonic Beaufort Gyre. A significant decrease in zonal separation is observed near the ice edge in triplet B due to convergence in the ice cover in response to the 8 October SLP high. Inter-beacon distances also increase in the meridional direction and decrease in the zonal direction for triplet B following the SLP high on 8 October. A transition to a more isotropic state in the ice cover for triplets closest to the coastline is evident in comparable zonal and meridional separations in triplet B in particular (right panel) as beacons encounter lower ice concentrations (Figure 7).**

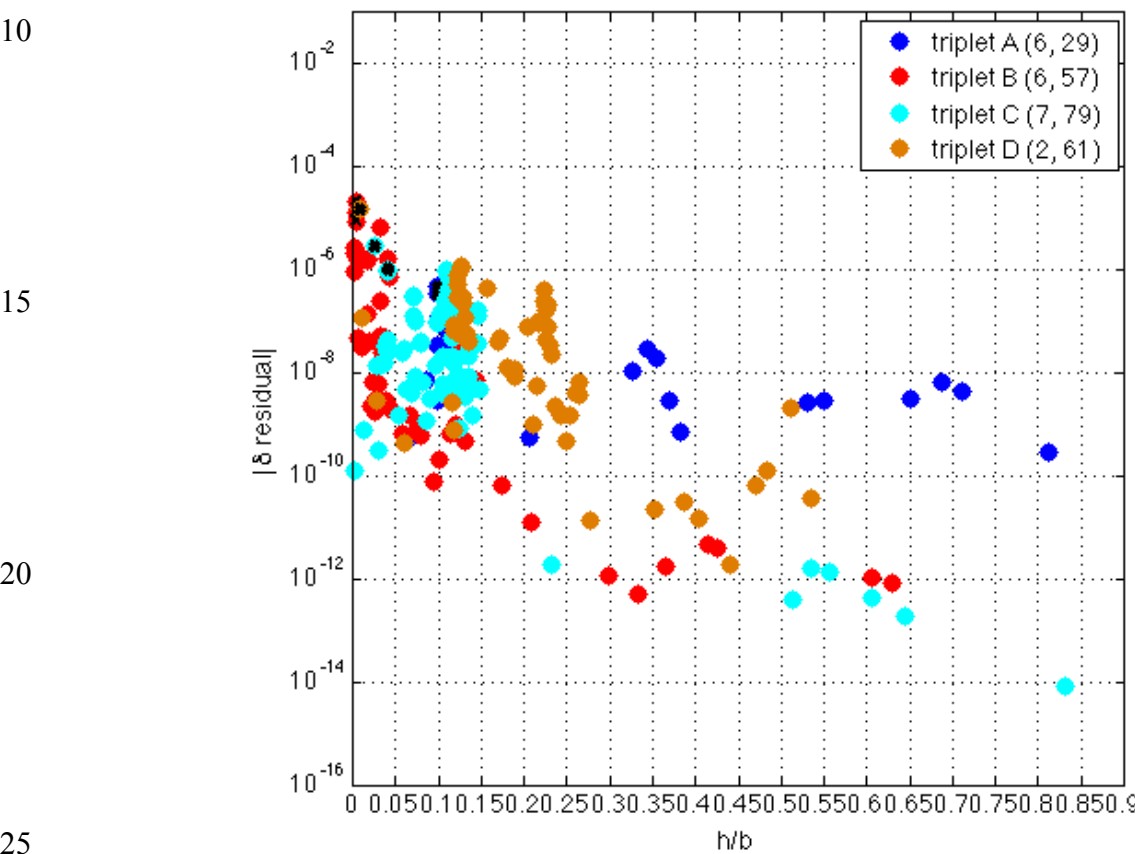

**Figure 9: Three-particle dispersion and DKP validity. Scatterplot of absolute value in divergence residual versus inverse aspect ratio h/b for triplets A to D to determine aspect ratio values for which triplets provide a reasonable estimate of DKPs. An 'x' symbol is placed over residual values exceeding twice the standard deviation. Threshold h/b values for triplets A to D are on the order of ~0.1, 0.0035, 0.025, and 0.008 corresponding to b/h values of 10, 290, 40, and 125, respectively, defined according to minimum aspect ratio values associated with residuals exceeding twice the standard deviation for each triplet. Numbers in brackets in the legend indicate the number of samples excluded from the three-particle and DKP analysis based on this criterion, and the total number of samples (positions) for each triplet trajectory.**

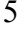

**Figure 10: Triplet area evolution. Time series of triplet area, base, height, base-to-height and perimeter-to-area ratios for triplets A to D. Solid lines in lower panel depict the base-to-height ratio, while the lines with symbols depict the perimeter-to-area ratio. The value for the base-to-height ratio associated with an equilateral configuration (1.155) is also shown. Horizontal bar graphs depict on-, along-, and offshore ice drift as depicted in Figure 6. Intervals excluded from DKP assessment based on the b/h threshold values found in Figure 9 include 14 – 22 September, 2009 for all triplets, 11 – 17 October, 2009 and 19 – 22 October, 2009 for triplet B, and following early November for triplet C. Triplet area evolution demonstrates enhanced variability in triplet B relative to triplets A, C, and D, with an increase in area near the ice edge in late October/early November. Higher base values are observed for triplets A and B relative to triplets C and D, in a manner consistent with local absolute dispersion (Figure 4). The triplet base also provides a measure of two-particle dispersion and separation between a pair of particles/beacons/ice floes.**

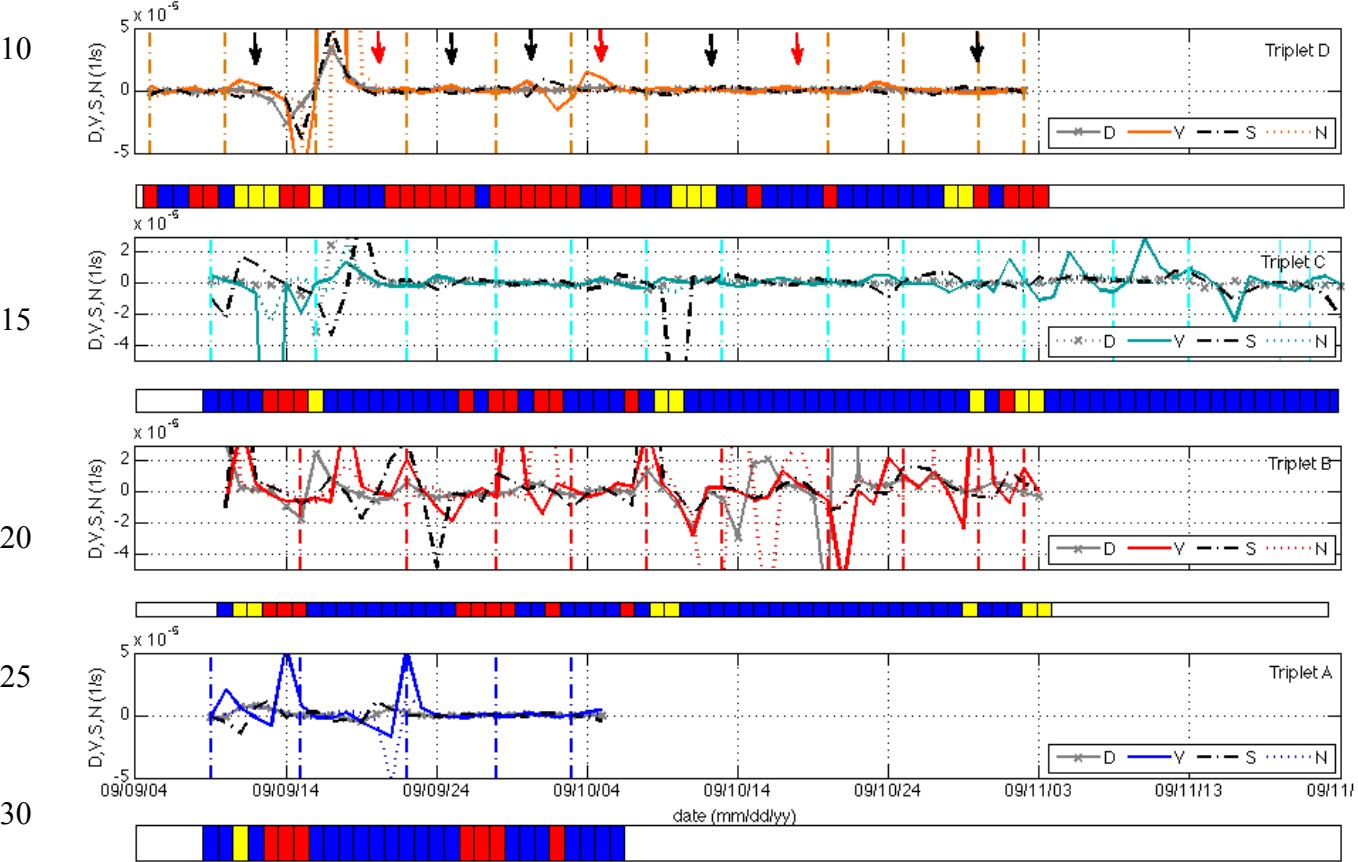

**Figure 11: Sea ice deformation.** Time series of Differential Kinematic Parameters (DKPs) divergence (D), vorticity (V), the
shearing deformation rate (S), and the normal (stretching) deformation rate (N) for triplets A (lowermost panel) to D (uppermost
panel). Vertical lines depict SLP high in the vicinity of each triplet centroid as shown in Figure 3. Horizontal bar graphs depict on-,
along, and offshore ice drift as depicted in Figure 6. For triplet A, located closest to the coastline, sea ice deformation is characterized
predominantly by vorticity and stretching. Triplet B is governed by vorticity for the duration of this triplet evolution, with
contributions from shear until 8-10 October, and divergence and stretching following mid-October. Triplet C, located further from the
coastline is characterized by vorticity in the early and late stages of triplet evolution, with intermittent contributions from shear and to
a lesser extent divergence during low ice concentration regimes. Triplet C also experiences enhanced vorticity following the 8 October
SLP high. A strong shear event is in addition observed for triplet B during the SLP high on 8 October, and for triplet C on 10 October.
Located furthest from the coastline, triplet D is governed by vorticity, stretching, shear, and divergence in the early stages of
development, and by vorticity and to a lesser extent shear for the duration of the triplet evolution.

45

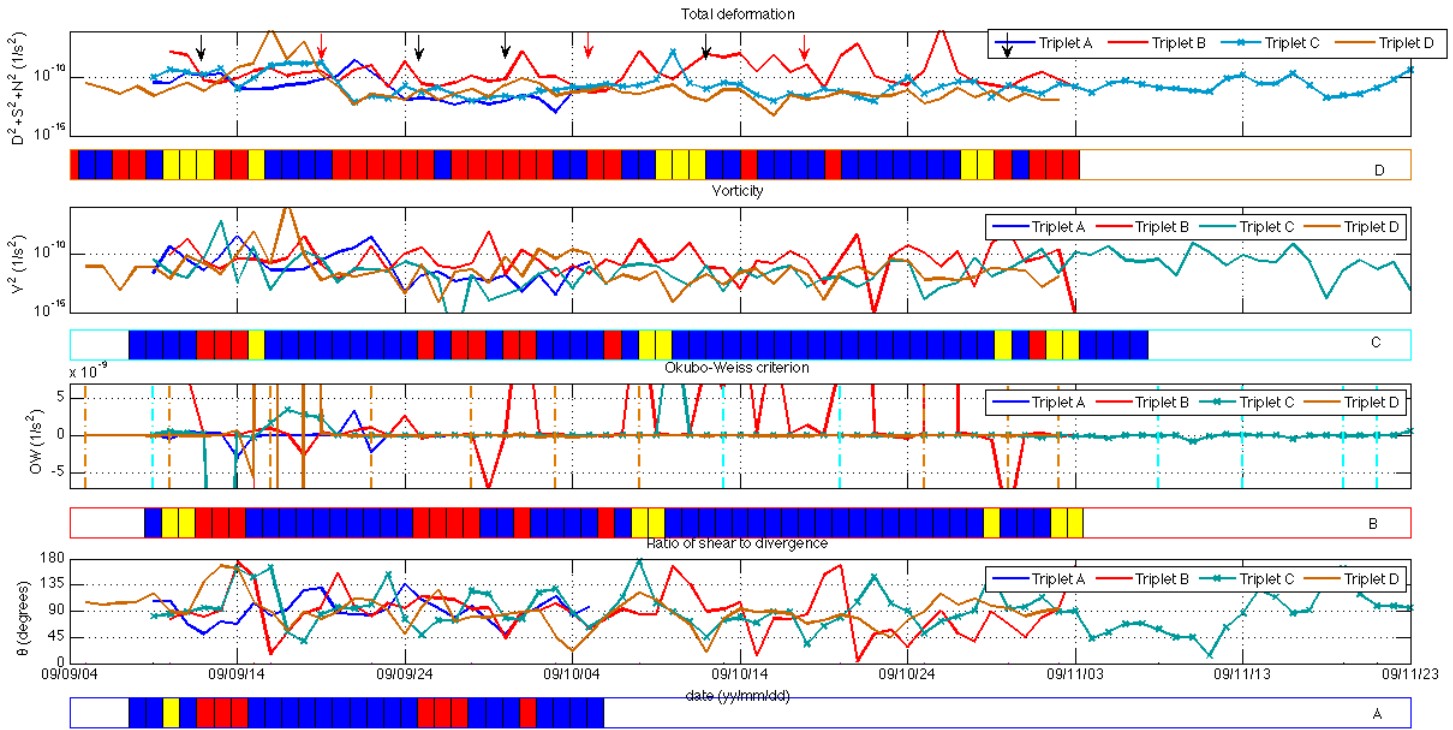

**Figure 12: Relative sea ice deformation.** Time series of relative sea ice deformation for Triplets A to D, including the total deformation
10   ($D^2 + S^2 + N^2$), vorticity ($V^2$), Okubo-Weiss criterion capturing relative contributions of deformation and vorticity with OW > 0
indicating deformation-dominated flow and OW <0 indicating flow dominated by vorticity, and $\theta$, or the arctan of the shear-to-
divergence ratio to further distinguish deformation between strain states as well as shear and divergence-dominated regimes.
Enhanced vorticity is observed during low ice concentration regimes, as demonstrated in comparatively high values for triplet D near
18 September, and for triplet B in late October/early November. Intervals of divergence and deformation-dominated flow prevail for
15   triplet B, with some instances observed for triplet A, C, and D again in the early stages of evolution. The Okubo-Weiss criterion results
in particular show enhanced distortion in the (bounded) ice cover closest to the coastline, especially following the 8 October SLP
maximum. Theta values illustrate changes in the ice strength due to relative contributions from divergence and the total strain rate (S²
+ N²). Noteworthy are the high (~180) values during e2, e7, and following e5, indicating convergence associated with offshore ice drift
due to Ekman convergence and sea ice deformation associated with along- and across-shear transport.

**Table 1: Evolution in triplet A to D trajectories; regional and local dispersion characteristics**

| Event | Total absolute dispersion $\alpha$ | | SLP | SIC (%) | Wind direction | SAT | Type of motion |
|---|---|---|---|---|---|---|---|
| | **Regional** $\alpha_R$ | Local $\alpha_L$ | | | | | |
| *e1* | super-diffusive ($\alpha \sim 2.8$) | super-diffusive ($\alpha > 1$) | L | 80,80,80,90 | S/N | < 2 | Cusp in all centro |
| *e2* | | sub-diffusive ($\alpha < 1$) for all triplets | L | 95-100 for all triplets | N | < 0 | Southwestward advection delayed southward migr |
| *e3* | | | L | | N/S | < 0 | Southward migration for |
| *e4* | | | L | | S/N | < 0 | Onset of south and w |
| *e5* | | | L | | N | < 0 | Onset of south and northw all triplet centroids culminati Triplet D; triplet A stops re |

**8 October SLP high**

| Event | Total absolute dispersion $\alpha$ | | SLP | SIC (%) | Wind direction | SAT | Type of motion |
|---|---|---|---|---|---|---|---|
| *e6* | ballistic ($\alpha \sim 2$) | super-diffusive ($\alpha > 1$) for triplet C; sub-diffusive ($\alpha < 1$) for triplets B and D | H/L | > 95 for triplet B 90 – 95 for triplets C and D | N/S (B, C) S (D) | < 0 | Northwestward migratio cusp in tripl |
| *e7* | | super-diffusive ($\alpha > 1$) for triplet C; sub-diffusive ($\alpha < 1$) for triplets B and D | L/H | 100 – 90 95 – 100 90 - 95 | N | < 0 | Northwestward migratio trajectories and loo |
| *e8* | | super-diffusive ($\alpha > 1$) for triplet B; sub-diffusive ($\alpha < 1$) for triplets C and D | L/H | 80-100 98-100 > 95 | S/N | < 0 | Northwestward migratio and D centroid tr |

**Table 2: Evolution in triplet A to D deformation and DKPs ***

| Event | Total relative dispersion β (algebraic) or exponential | Area | Okubo-Weiss (OW > 0; deformation) (OW < 0 ; vorticity) | Shear-to-divergence ratio 0° = divergence – D 45 ° = extension – E 90° = shear–S 135° = contraction – Ct 180° = convergence – Cv |
|---|---|---|---|---|
| e1 | | max (A); min (B to D) | < 0 (A, B, and D); > 0 (C and D) | A: [E, S, Ct] B: [S, Cv, D, S] C: [S, Cv, E, S] D: [Cv, E, S] |
| e2 | | Inc (A); min (B); const (C and D) | < 0 (A); > 0 (A and B); ~ 0 (C and D) | A: [S, Ct] B: [Ct, S]; C: [S, Ct, E] D: [S, E, S] |
| e3 | β < 1 for all triplets  β < 1 for all triplets | Const. for all triplets | < 0 (B) ~ 0 (A, C, D) | A: [Ct, S, E] B: [Ct, E] C: [E, S, Ct, S] D: [Ct, S] |
| e4 | | Const. | < 0 (D) > 0 (B) ~ 0 (A, C, D) | A: [S] B: [S, E] C: [Ct, E] D: [S, D, E] |
| e5 | | Const. | < 0 (D) ~ 0 (A, B, C) | A: [S] B: [E, S, Cv, S]; C: [E, Cv, E] D: [E, Ct, E] |
| **8 October SLP high** | | | | |
| e6 | β > 1 (~ 3) for triplets C and D | Max (B); min (C); const (D) | > 0 (B, C) ~ 0 (< 0, B; > 0, D) | B: [S, D, S] C: [E, S] D: [E, S] |
| e7 | exponential for triplet B | Min (B); const (C and D) | > 0 (B; C/100) ~ 0 (D) | B: [S, Cv, D, E, S, E] C: [S, Ct, E, S, Cv] D: [S, E, Ct, S] |
| e8 | | Min (B); const (C and d) | > 0 (B and C/100) < 0 (B and C/100) ~0 (D) | B: [E, S, Cv] C: [Cv, S] D: [S] |

**DKPs and impact on triangles**

**Divergence changes the triangle area; vorticity changes the triangle orientation; shear and stretching rates change triangle shape.**

*Events excluded from the three-particle dispersion and DKP analysis due to triangle elongation based on the criteria presented in Figure 9 are depicted as faded text.

**Table 3:** Sea ice drift and deformation: single ($\alpha$) - and two ($\beta$) – particle scaling laws with three particle Okubo-Weiss Criteria (OW) cross referenced

*increasing connectivity* (↓)

| $\beta$ | | | | |
|---|---|---|---|---|
| | >2 | Trapping over a range of spatial scales | Range of spatial scales encountered | Turbulent diffusion ($\beta \sim 3$) and anomalous dispersion ($\alpha > 2$) |
| | ~2 | Homogeneous turbulence | | |
| | ~1 | Trapping due to barriers to transport, uninhibited by spatial gradients in ice drift | $A^2 \sim D^2$ Mixing | Anomalous dispersion (uninfluenced by separations in long-time limit for $D^2$) Advection and long-range correlations; organized structure Ballistic ($\alpha \sim 2$) |
| | <1 | Trapping due to ice-ice and ice-coast interactions governed by spatial scales of separation | Maximum spatial scales of motion attained due to ice-ice and/or ice-coast interactions | Advection (and topological structure) constrained by ice-ice and ice-coast interactions |
| **OW > 0** (deformation) θ: 180 (conv.); 90 (shear); 0 (div) | | < 1 subdiffusive | ~1 diffusive | >1 superdiffusive |
| **OW < 0** (vorticity) | | $\alpha$ | | |

*increasing sea ice concentrations*

(←)

## Appendix A: Methodology to Derive Daily Positions, Triplet Arrays, Centroids, and Ice Drift from Telemetry Data

Geographical positions are recorded from global positioning (GPS) beacons explained in the data section. The observed temporal resolution of the beacon position data is two hours (dt = 2 hours). Daily average
positions are calculated for the analysis and time series. The position data is subsequently used to compute drift components based on triplet centroid daily displacements.

### A.1. Sea ice drift and triplet centroids

A four-step process was used to convert telemetry geographical position records into daily beacon averages, centroid locations, and sea ice drift. First, daily average positions were calculated from the two-
hourly data for each beacon. Second, triplets were organized based on proximity to the coastline, inter-beacon distances, and overlapping time intervals. Triplets A and B, with two shared beacons, were selected to highlight differences in ice drift and deformation on scales comparable to inter-beacon separations.

Third, centroids are calculated using the coordinates of the three beacons comprising each triplet. Fourth, ice drift is computed from centroid displacements. Specifically, the geographical latitude and longitude
decimal degrees are converted to horizontal ortho-linear metric distances, using the north polar azimuthal equal-area map centred on the North Pole. Thus for the Earth's radius $R = 6371.228\ km$, pixel size $C = 25.0675\ km$, latitude $\phi$, longitude $\lambda$, row and column origin $r0 = s0 = 181$ for a $361x361$ grid, and

$$r = 2R \sin \lambda \sin\left(\frac{\pi}{4} - \frac{\phi}{2}\right)$$
$$s = -2R \cos \lambda \sin\left(\frac{\pi}{4} - \frac{\phi}{2}\right),$$

the speeds associated with the metric distances, $u_r = 10^3 \Delta r/(\Delta t)$ and $u_s = 10^3 \Delta s/(\Delta t)$, for $\Delta t = 24 \times 3600(s)$, are transformed to zonal and meridional components such that

$$u = u_r \cos \lambda + v_r \sin \lambda$$
$$v = -u_r \sin \lambda + v_r \cos \lambda.$$