# Peer review of "Method to characterize directional changes in arctic sea ice drift and associated deformation due to synoptic atmospheric variations using Lagrangian dispersion statistics"

_The Cryosphere, 2016_

## Referee Comment (RC1) · Anonymous Referee #1 · 31 Oct 2016

General comments: This paper investigated the deformation processes of sea ice in the southern Beaufort Sea from the analysis of several buoy data with special attention to the sudden changes in sea ice drift and its relevance to atmospheric forcing, ice conditions, and the effect of shore. The goal of this study is placed in developing a framework for understanding sudden changes in ice drift trajectories. For this purpose, firstly the authors set 4 triplet areas composed of three buoys for each, and then traced the temporal evolution of each triple. As a result of analysis during the period from September to November in 2009, they detected eight "sudden change" events and examined the kinetic deformation parameters in relevance to the atmospheric forcing, ice conditions, and the effect of shore. They used triplet area, perimeter-to-area, the Okubo-Weiss criterion as diagnostic parameters. From their analysis, they concluded that sudden change occurred reflecting sea ice deformation, associated with

the transition of atmospheric forcing and the interaction with the coastal line, and so on. I understand the importance of this topic and find that it may be useful to understand the dynamical features of sea ice near the shore on a daily time scale and on a sub-grid spatial scale. And that must be what the author aims at in this paper. I see that the authors attempted to make the most of available several buoy data to reveal them, and their efforts should be appreciated. However, I feel the manuscript is a bit more descriptive and the conclusion does not necessarily seem clear. My opinion is that the paper would be improved much if the author show more clearly what is a new finding of this study in the context of the research history on sea ice dynamics. Thus my evaluation is somewhat reserved at this stage. My major concerns are as follows: 1) Discussion and conclusion seems to me a bit qualitative. I mean that to correlate the sudden change events with the change in atmospheric forcing or the interaction with coastal line, it would be helpful to show how much forcing (e.g. change in wind speed, or shear of sea ice drift) was needed and examine if the result can be explained in the present framework of sea ice dynamics. It might be difficult to draw quantitative conclusion just from the available datasets. Even so, it would be possible to make additional figures which explain the thought of the authors more clearly, such as scatter plots as a function of the distance from the coast or a schematic picture. The figures in the present manuscript are only time series of physical parameters. It is not necessarily easy to understand the essence of this paper just from the time series. 2) Sorry, but I am a bit skeptical about the analytical method of deformation parameters using triplets. Although I agree that this method would be useful if the side lengths of the triplet are of almost similar magnitude, I feel it is questionable if the triplet becomes so distorted that the lengths of base and height have significantly different magnitude, as shown in Figure 6a. This is because the divergence or deformation parameters would take different values, depending on the horizontal scales. In such situation as the side lengths of the triplet are different by more than one-order, I wonder if the obtained values are representative of the region and therefore this method is really applicable. If the authors are convinced about this matter, it would be helpful to add some explanation. 3) About the

terminology. Several expressions about "sudden change" events might be a bit confusing. The authors used "sudden changes", "shear shock events", or "shock-response" and whatever for the similar events. Do they mean the same phenomena? If so, it would make the manuscript more readable to unify the expression after defining them at the beginning. 4) I would like to know more about the motivation of investigating the "sudden change" events. I also consider that this is a very important issue in the sea ice dynamics because it can induce the crack or formation of leads which would affect a large scale dynamics of sea ice area. So it would make the paper more impressive if the authors show some pictures which show how cracks were formed associated with the "sudden change" events. Personally, I think it might be interesting to discuss it from aspect of the yielding mechanics, namely the transition from viscous to plastic behavior in the VP rheology in the numerical sea ice model. To do so, separate the events which appeared in the persistent atmospheric forcing from those that occurred corresponding to the change in the atmospheric forcing. This might be one idea to the manuscript more quantitative.

Specific points: *(P3L24) "(Kwok, 2006)" is missing in the reference lists. *(P9L6) "Figure 2b" seems missing. *(P11L5) "Results show reduced total deformation with increasing distance from the coastal line (figure 8)" For me it is not so clear just from Fig.8. Especially the difference between A and B cannot be explained so well. *(P11L9-10) "Noteworthy is the existence of vorticity-dominated flow. . .." Please explain why this is noteworthy. *(P12L19) "indicating the impact of ice interactions with coastline" Please explain more about the reason. *(P13L28-29) "a continued increase in temperature. . .." I wonder that the reason for this interpretation is not enough because other factors such as the change in synoptic atmospheric circulation might have affected the temperature. Please add some more explanation. *(P14L5) "Noteworthy also is increasing SAT" Please specify the period of this phenomena. *(P16L13) How did you estimate "ice strength". *(P16L19) "vorticity superimposed on shear weakens ice strength" I could not understand this. Please explain more.

Technical corrections: *(Figure 1b) I recommend to have the edge of each circle colored in black because some circles are hard to see. *(Figure 3) It would be helpful if the "sudden change" events are shown by arrows in the figure. (Figure 6a) Please magnify the numbers of latitude and longitude. And please designate which color corresponds to A – D. (Figure 6b, 7, 8) Please magnify the scales of the figures. They are hard to see.

That is all. Faithfully yours.

---

## Referee Comment (RC2) · A. Provenzale (Referee) · 1 Dec 2016

The paper "Characterizing sudden changes in Arctic sea ice drift and deformation on synoptic timescales" discusses the use of Lagrangian triplet dynamics, combined with characterizations such as the Okubo-Weiss parameter, to identify "sudden changes" in sea ice drift in the Arctic.

The material is interesting and it builds upon previous works by the same lead author. However, I find the paper rather difficult to read, and not very clear in its message.

First, most of the figure are simple displays of time series, without too much statistical analysis and/or quantitative interpretation.

The paper would benefit from a more quantitative approach, with results of the statistical analyses, to assess the validity and significance of the conclusions.

I also urge the authors to streamline the paper, making it more palatable and under-standable. In particular, I would like to add a paragraph at the beginning of the Intro-duction explaining some more fact about Arctic sea ice and sea ice drift.

Finally, it is not clear what "sudden changes" in sea ice are, and to what meteorologi-cal/climatic events are related. This point should be further explored and clarified.

---

## Author Comment (AC1) · 31 Jan 2017

Please find below a list of responses to reviewers' comments regarding the submitted manuscript, "Characterizing sudden changes in Arctic sea ice drift and deformation on synoptic timescales", by J.V. Lukovich, C.A. Geiger, and D.G. Barber. General comments are presented, followed by responses to reviewers' specific comments and suggestions. Please note that changes implemented may also be found in the attached supplementary material in response to reviewers' suggestions.

General comments:

The authors would like to thank both reviewers for their constructive comments and suggestions. In consideration of reviewer recommendations, and the initial motivation for the triplet analysis, the authors have in the revised manuscript presented a framework based on Lagrangian dispersion, and single-, two-, and three-particle dispersion statistics in particular to provide a quantitative analysis and more focused narrative of sudden changes in ice drift paths as well as associated changes in sea ice deformation, based on distance from the coastline.

Please find below specific comments to suggestions and recommendations.

Specific comments:

Anonymous Referee #1

General comments: This paper investigated the deformation processes of sea ice in the southern Beaufort Sea from the analysis of several buoy data with special attention to the sudden changes in sea ice drift and its relevance to atmospheric forcing, ice conditions, and the effect of shore. The goal of this study is placed in developing a framework for understanding sudden changes in ice drift trajectories. For this purpose, firstly the authors set 4 triplet areas composed of three buoys for each, and then traced the temporal evolution of each triple. As a result of analysis during the period from September to November in 2009, they detected eight "sudden change" events and examined the kinetic deformation parameters in relevance to the atmospheric forcing, ice conditions, and the effect of shore. They used triplet area, perimeter-to-area, the Okubo-Weiss criterion as diagnostic parameters. From their analysis, they concluded that sudden change occurred reflecting sea ice deformation, associated with the transition of atmospheric forcing and the interaction with the coastal line, and so on. I understand the importance of this topic and find that it may be useful to understand the dynamical features of sea ice near the shore on a daily time scale and on a sub-grid spatial scale. And that must be what the author aims at in this paper. I see that the authors attempted to make the most of available several buoy data to reveal them, and their efforts should be appreciated. However, I feel the manuscript is a bit more descriptive and the conclusion does not necessarily seem clear. My opinion is that the paper would be improved much if the author show more clearly what is a new

finding of this study in the context of the research history on sea ice dynamics. Thus my evaluation is somewhat reserved at this stage. My major concerns are as follows:

1) Discussion and conclusion seems to me a bit qualitative. I mean that to correlate the sudden change events with the change in atmospheric forcing or the interaction with coastal line, it would be helpful to show how much forcing (e.g. change in wind speed, or shear of sea ice drift) was needed and examine if the result can be explained in the present framework of sea ice dynamics. It might be difficult to draw quantitative conclusion just from the available datasets. Even so, it would be possible to make additional figures which explain the thought of the authors more clearly, such as scatter plots as a function of the distance from the coast or a schematic picture. The figures in the present manuscript are only time series of physical parameters. It is not necessarily easy to understand the essence of this paper just from the time series.

Thank you for your comments and suggestions. The authors agree that the initial version of the manuscript was qualitative, and have addressed this through investigation of single-, two-, and three-particle dispersion statistics. Since the initial motivation of this manuscript was to develop a diagnostic that built upon previous Lagrangian dispersion statistical analyses, it was decided that investigation of each using this dataset would provide a comprehensive framework with which to assess the regional-scale changes in sea ice drift (single-particle; Figures 3 and 4 in supplementary material), deformation (two-particle; Figure 9 in supplementary material), and deformation components (three-particle) using the available data, and as motivation for future observational/modeling studies focused on sea ice drift and deformation. In light of the present reviewer's comments, the authors have also included analyses that capture relative contributions of wind and deformation using scatter plots, as suggested, based on distance from coastline, in order to provide additional insight into sea ice dynamics in the Beaufort Sea region based on distance from the coastline (Figure 15 in supplementary material). This work further builds upon oceanographic studies illustrating the benefits of Lagrangian dispersion statistics in capturing structure in the oceanic (flow) field,

through investigation of single-, two-, and three-particle dispersion in the context of changing sea ice and atmospheric conditions.

2) Sorry, but I am a bit skeptical about the analytical method of deformation parameters using triplets. Although I agree that this method would be useful if the side lengths of the triplet are of almost similar magnitude, I feel it is questionable if the triplet becomes so distorted that the lengths of base and height have significantly different magnitude, as shown in Figure 6a. This is because the divergence or deformation parameters would take different values, depending on the horizontal scales. In such situation as the side lengths of the triplet are different by more than one-order, I wonder if the obtained values are representative of the region and therefore this method is really applicable. If the authors are convinced about this matter, it would be helpful to add some explanation.

The triangle base in the present analysis is defined as the triangle side with the longest length, in keeping with past studies that have characterized evolution in the triangle configuration (LaCasce, 2008). In a comparison of the least squares method derived according to (local) spatial derivatives in particle velocities, and the triangle area rate of change approach (Molinari and Kirwan, 1975; LaCasce, 2008), it was shown that both methods yield comparable results for drifters separated by distances of similar magnitude. Both methods were also shown to agree for large spatial gradients in the flow field (Molinari and Kirwan, 1975). As is described in both studies, the least squares approach resembles a Taylor series expansion about a centre of mass for a finite fluid element, and is applicable at local scales, namely when the distances between drifters/beacons are of comparable size. It was also shown that uncertainty associated with the least squares approach is reduced with an increase in the number of particles considered. A description of the triplet area rate of change approach in Saucier (1955) highlights that the scale factor associated with map projections cancels since the triangle area exists in the numerator and denominator of the area-weighted approach for calculating DKPs. Similarly, distortion in the triangle area is captured in the numerator

and denominator so that it is the time rate of change in area that is captured by the DKPs. Diagnostics such as the aspect ratio and Okubo-Weiss criterion are also used to demonstrate relative DKP contributions based on distance from the coastline.

Additional clarification of this approach is provided in the revised manuscript and supplementary material, and methods section in particular, in lines 6 to 11 on page 11, as follows:

"Previous studies of DKPs using the triangle area approach have shown that the role of triplet areas in describing DKPs resides in the evolution in the time rate of change in the triangle area (Saucier, 1955; Molinari and Kirwan, 1975). If the lengths of the base (defined as the longest triangle side) and height (defined as the perpendicular distance and 2A/b) differ by an order of magnitude so that the triangle is significantly distorted, a decrease in area will occur. If in addition the change in area exceeds its uncertainty, the DKP associated with the relevant rotation of coordinates will increase, providing a signature of strong deformation. If, however, little change in triplet area is observed (less than the area uncertainty $\sim$ 0.12 km2), the DKP in question will essentially vanish. In the present study, as is noted below, minimum values for the triplet area amongst all triplets are on the order of 10 km."

3) About the terminology. Several expressions about "sudden change" events might be a bit confusing. The authors used "sudden changes", "shear shock events", or "shock-response" and whatever for the similar events. Do they mean the same phenomena? If so, it would make the manuscript more readable to unify the expression after defining them at the beginning.

The authors agree that multiple terms for the same expression detracted from the narrative in the original version of the manuscript. "Sudden changes" in the initial manuscript referred to minima in the centroid variances as a signature of interruptions to the ice drift path. By contrast, the "shear shock event" referred to a specific event in October characterized by strong shear, following which a loss of coherence

in ocean-sea-ice-atmosphere interactions occurred. Finally, "shock-response" referred to sea ice drift response to atmospheric forcing. The authors have attempted in the revised manuscript to ensure consistency in the terminology referring to shock events in the context of sea ice drift and deformation in response to atmospheric forcing.

In the revised version of the manuscript, we define sudden changes at the beginning of the manuscript as changes in the ice drift path (p. 2, line 20), and quantify sudden changes in terms of minima in ice drift variance and inflection points in single-particle dispersion.. Reference is now made to the 8 October 2008 SLP high that contributed to Ekman convergence in the sea ice cover, rather than the shear-shock event, since the SLP event reflects not only the emergence of shear in triplet B, and in triplet C two days later, but also the shift in sea ice dynamical regimes. In addition the 8 October SLP high depicts a transition in the dynamical state of the ice cover, as captured in single-particle dispersion statistics (Figure 3). Furthermore, the authors have removed reference to "shock-response" mechanisms in order to focus the content and improve clarity of the manuscript.

4) I would like to know more about the motivation of investigating the "sudden change" events. I also consider that this is a very important issue in the sea ice dynamics because it can induce the crack or formation of leads which would affect a large scale dynamics of sea ice area. So it would make the paper more impressive if the authors show some pictures which show how cracks were formed associated with the "sudden change" events. Personally, I think it might be interesting to discuss it from aspect of the yielding mechanics, namely the transition from viscous to plastic behavior in the VP rheology in the numerical sea ice model. To do so, separate the events which appeared in the persistent atmospheric forcing from those that occurred corresponding to the change in the atmospheric forcing. This might be one idea to the manuscript more quantitative.

The investigation of sudden change events is motivated by the development of a Lagrangian framework that can be used in both observational and modeling studies to

characterize sea ice drift and deformation over a range of spatial and temporal scales. Sudden change events provide a means to characterize and quantify changes in the ice drift paths and associated deformation in response to atmospheric forcing and based on distance from the coastline, relevant for ice hazard detection and contaminant transport. In light of the both reviewers' comments and in order to provide a more quantitative analysis, the authors have described sudden change events in the context of single-, two-, and three-particle dispersion. In assessing sea ice drift using single-particle dispersion, a shift in dynamical regimes associated with the SLP high on 8 October was observed, and the poleward retreat in the sea ice edge due to convergence in the ice cover in response to the SLP high is presented in Figure 8 of the revised manuscript showing Canadian Ice Service ice charts prior to, during, and following the 8 October SLP high event.

Thank you also for the suggestion of focusing on yielding mechanics – the authors have attempted to address this through an assessment of the shear-to-divergence ratios as a signature of relative contributions in the yield curve. Although the derivation and assessment of yield curves is beyond the scope of the present study, the authors provide a preliminary assessment of shear to divergence ratios during persistent northeasterly wind events (e2, e5, and e7), where along- and cross-shear transport as captured by two-particle dispersion gives rise to local compression and convergence in the ice cover, evident in S/D values on the order of 180.

Specific points: *(P3L24) "(Kwok, 2006)" is missing in the reference lists.

Thank you for this suggestion. This citation is now included in the reference section.

*(P9L6) "Figure 2b" seems missing.

This is now corrected.

*(P11L5) "Results show reduced total deformation with increasing distance from the coastal line (figure 8)" For me it is not so clear just from Fig.8. Especially the difference

between A and B cannot be explained so well.

This statement has been modified to describe enhanced deformation in triplet B near the ice edge following the SLP high, as captured by the total deformation $\sqrt{(D^2+N^2+S^2)}$.

*(P11L9-10) "Noteworthy is the existence of vorticity-dominated flow. . .." Please explain why this is noteworthy.

This statement has been removed from the revised manuscript as it does not contribute to its content.

*(P12L19) "indicating the impact of ice interactions with coastline" Please explain more about the reason.

This statement was initially intended to demonstrate that coherence in turning angles occurs within a high ice concentration regime. However, the authors agree that high turning angles for all triplets do not indicate the impact of interactions with the coastline, and have thus removed this text from the revised manuscript.

*(P13L28-29) "a continued increase in temperature. . .." I wonder that the reason for this interpretation is not enough because other factors such as the change in synoptic atmospheric circulation might have affected the temperature. Please add some more explanation.

The authors agree with the present reviewer that factors other than a crack in the ice cover may be responsible for increasing surface air temperature in the vicinity of triplets B, C, and D (Figure 7). The local SAT maximum does coincide with a SLP minimum, which could have advected warm air into the region from lower latitudes. Local warming may also be connected to coastal upwelling events associated with the SLP high, Ekman convergence, downwelling in the central basin and corresponding upwelling near the coast. In the absence of a more rigorous analysis explaining the physical mechanisms responsible for this particular feature, which is beyond the scope

of the present study, this statement is also removed in the revised manuscript.

*(P14L5) "Noteworthy also is increasing SAT" Please specify the period of this phenomena.

The statement is now expressed as" Noteworthy also is increasing SAT in the vicinity of triplet B and to a lesser extent C, relative to triplet D in early November."

*(P16L13) How did you estimate "ice strength".

The shear-to-divergence ratio is used in the present study as a measure of ice strength, with theta values of 0 (180) characteristic of divergence (convergence) indicating reduced (increased) ice strength. This sentence is clarified in the revised text such that

"Changes in ice strength monitored by shear-to-divergence ratios are associated with the transition from an on or offshore to along-shore ice drift regime. "

In light of the present reviewer's comments, following the findings of Richter-Menge et al. (2002) showing correspondence between ice deformation and stress measurements, and since ice strength is dependent on ice concentrations and thickness, the phenomenon whereby the ice cover fails due to internal ice stress comparable to ice strength is also examined in the context of Figure 14 and understanding the rheological characteristics of sea ice.

*(P16L19) "vorticity superimposed on shear weakens ice strength" I could not understand this. Please explain more.

The authors agree that this statement did not provide much clarity or information regarding the correspondence between DKPs and the S/D ratio as a measure of ice strength. The initial intent was to describe the role of external forcing captured by vorticity in disrupting shear flow and subsequently reducing ice strength. This sentence has however been removed given the speculative nature of this assertion.

Technical corrections: *(Figure 1b) I recommend to have the edge of each circle colored

in black because some circles are hard to see.

This has been corrected in Figure 1b of the supplementary material.

*(Figure 3) It would be helpful if the "sudden change" events are shown by arrows in the figure.

Thank you for this suggestion. Arrows indicating sudden change events are now included in most figures.

(Figure 6a) Please magnify the numbers of latitude and longitude. And please designate which color corresponds to A – D.

This has also been corrected in Figure 10 of the supplementary material.

(Figure 6b, 7, 8) Please magnify the scales of the figures. They are hard to see.

The scales of the figures have been magnified, and the x- and y-axis labels in particular.

That is all. Faithfully yours.

Thank you once again for helpful comments and suggestions.

Please also note the supplement to this comment:
http://www.the-cryosphere-discuss.net/tc-2016-219/tc-2016-219-AC1-supplement.pdf

[Figure]

**Fig. 1.** Figure 3 (revised version): Absolute (single-particle) dispersion statistics for triplets A to D, depicting zonal (red), meridional (blue), and total (black) dispersion

[Figure]

**Fig. 2.** Figure 4 (revised version): Absolute (single-particle) dispersion statistics depicting meridional (top), zonal (middle), and total (lower) dispersion to characterize local changes in ice drift.

[Figure]

**Fig. 3.** Figure 9 (revised version): Relative (two-particle) dispersion showing meridional (top), zonal (middle), and total (lower panel) relative dispersion as a function of elapsed time.

[Figure]

**Fig. 4.** Figure 15 (revised version): Scatter plots of NARR winds versus DKPs for triplets A to D showing density of values in wind and DKP bins. Symbols depict sudden change events.

**Supplement:**

**Characterizing sudden changes in Arctic sea ice drift and deformation on synoptic timescales**

J.V. Lukovich[1], C.A. Geiger[2], D.G. Barber[1]

[1]Centre for Earth Observation Science, University of Manitoba, Winnipeg, R3T 2N2, CANADA
5 [2]College of Earth, Ocean, and Environment: Geography, University of Delaware, Delaware, 19716, USA

*Correspondence to*: J.V. Lukovich (Jennifer.Lukovich@umanitoba.ca)

**Abstract.** In this study, we develop a framework for the assessment of sudden changes in sea ice drift paths and associated deformation processes in response to atmospheric forcing, based on a Lagrangian statistical analysis of ice buoy triplet centroids and areas. Examined in particular is the spatiotemporal evolution in sea ice floes that are tracked with GPS beacons deployed in triplets in the southern Beaufort Sea at varying distances from the coastline in fall, 2009 – triplets A to D, with A (D) located closest to (furthest from) the coastline. This study illustrates the use of diagnostics to evaluate eight identified sudden changes or shock events on daily timescales. Results from this analysis show that single-particle (absolute) dispersion provides a characterization of regional changes in the sea ice cover in response to atmospheric forcing, while two- (relative) and three-particle dispersion statistics provide a characterization of local changes associated with sea ice deformation. Demonstrated in particular is a change in sea ice dynamics following a SLP high on 8 October, 2009, evident in a transition from absolute dispersion scaling exponents of $\alpha \sim 3$ to 2, and accompanying loss of synchronicity in ice-atmosphere interactions. Results from two- and three-particle dispersion statistics highlight differences in sea ice deformation based on distance from the coastline. The tools developed in this study provide a unique characterization of sea ice dynamical processes in the southern Beaufort Sea based on Lagrangian, and in particular one-, two-, and three-particle dispersion statistics, with implications for quantifying sudden changes relevant for ice hazard assessments and forecasting applications required by oil and gas, marine transportation, and indigenous use of near shore Arctic areas.

**1 Introduction**

Central to our understanding of changes in the Arctic sea ice cover in response to a changing climate and continued anthropogenic forcing is an understanding of sea ice drift and deformation, namely sea ice dynamics. Accelerated ice drift speed over the past several decades reflects a weaker and more mobile ice cover associated with the loss of multiyear ice and changes in atmospheric circulation (Hakkinen et al. 2008; Barber et al., 2009; Rampal et al. 2009b; Spreen et al., 2011; Kwok et al., 2013). Sea ice deformation, or spatial gradients in the ice drift field, associated with opening and closing in the ice cover due to sea ice divergence and convergence, influences moisture and heat exchange between the ocean and atmosphere, ice ridging, sea ice thickness and redistribution (Hutchings et al., 2011, Bouillon et al., 2015) with implications ice hazard detection, and pollutant and contaminant transport. In the Beaufort Sea region, sea ice dynamics is governed by large-scale anticyclonic circulation of the Beaufort Gyre, with reversal to cyclonic circulation throughout the annual cycle (LeDrew et al. 1991; Preller and Posey, 1989; Proshutinsky et al., 2015).

When the ice cover on polar seas changes suddenly, navigation channels are altered as a result of ice-ice and ice-coastline momentum and energy flux exchanges (e.g., The Polar Group, 1980; Hwang, 2005; McPhee, 2012), air-sea heat exchanges increase (e.g., Carmack et al., 2015), and newly opened leads vent high moisture into the atmosphere as a strong mass exchange process (Bourassa et al., 2013). Understanding how these changes develop and relate to the orientation of a coastline is essential when diagnosing response patterns. For clarity, sudden change in this study refers to changes in the ice drift path relative to storm tracks, which have typical duration on the order of days and recurrence rates on the order of several days to weeks.

Arctic air-ice-sea interactions on synoptic timescales (several days to weeks) are governed by a force balance consisting of three interactive components: i) sea ice motion, ii) a confining coastline, and iii) atmospheric forcing. Previous studies have examined sea ice drift and deformation response to atmospheric forcing and coastline geometry on varying timescales (Overland et al., 1995; Richter-Menge et al., 2002; Geiger and Perovich, 2008; Hutchings et al., 2011). In an assessment of springtime sea ice drift in a region to the west of the Antarctic Peninsula, Geiger and Perovich (2008) identified low-frequency motion in response to atmospheric forcing and coastal geometry associated with

regional-scale transport, and higher-frequency near-inertial oscillatory motion associated with mixing. On regional and synoptic scales, Richter-Menge et al. (2002) also distinguish between translational and differential motion associated with shear zones and discontinuities in the ice drift characteristics in the southern Beaufort Sea.

5     The role of forcing (wind stress) and coastline geometry in establishing coherence in lead patterns/fractures in the ice cover captured by sea ice deformation has also been explored in past studies (Overland et al., 1995; Hutchings et al., 2005, 2011). Overland et al. (1995) demonstrated that in the Beaufort Sea for spatial scales: i) exceeding 100 km the sea ice cover moves as an aggregate; ii) less than 100 km the ice cover moves as an aggregate or discrete entity based on whether an elliptic

10  (homogeneous) or hyperbolic (discrete) regime is established relative to the coastline (ice-coast interactions); and iii) on the order of 1 km the ice cover is characterized by floe (ice-ice) interactions. Through analysis of a nested beacon configuration and array with spatial scales ranging from 10 km to 140 km as part of the late winter (April) 2007 Sea Ice Experiment: Dynamic Nature of the Arctic (SEDNA) campaign in the Beaufort Sea, Hutchings et al. (2011) demonstrated coherence between 140

15  km and 20km divergence arrays for time periods of up to 16 days in March. Over shorter (sub-synoptic) timescales from May 2007 onward, nested buoy arrays captured the loss of connectivity in the sea ice cover associated with the winter-to-summer transition during a substantial ice-loss year (Stroeve et al., 2008).

In this study, we extend these analyses to quantify changes in the ice cover and in particular to

20  explore sea ice drift and deformation using Lagrangian dispersion statistics. For simplicity and for the synoptic timescales considered in this study, we ignore lower-frequency ocean current fluctuations. We also ignore higher-frequency fluctuations associated with inertial oscillations, which are explored in a companion paper (Geiger and Lukovich, in preparation). We instead focus here on the spatiotemporal synoptic changes in sea ice drift and deformation that are directly related to storm tracks while they are

25  developing and migrating through a local region, using a novel observational and analytical approach based on one-, two-, and three-particle dispersion analyses and evolution of beacons deployed in a triangular configuration as triplets

Central to developing the tools required to understand sea ice drift and deformation in response to atmospheric forcing and ice-coastline interactions are diagnostics and in the case of drifting buoys, a Lagrangian framework to quantify spatiotemporal changes in the ice cover. Previous studies have used Lagrangian dispersion and ice beacon trajectories to quantify sea ice drift and deformation in the Arctic

5 (Colony and Thorndike, 1984, 1985; Rampal, 2008, 2009a,b; Lukovich et al., 2011, 2014, 2015). Single-particle (absolute) dispersion provides a signature of large-scale circulation and captures linear time-dependence in fluctuating velocity variance characteristic of turbulent diffusion theory (Taylor, 1921; Rampal et al., 2009); i.e., departures in ice fluctuating velocity statistics from turbulent diffusion are attributed to intermittency associated with sea ice deformation and internal ice stress (Rampal et al.,

10 2009).

A two-particle (relative) dispersion analysis monitors sea ice deformation. Through evaluation of buoy pair separations as a proxy of strain-rate (divergence, convergence, and strain) components combined, two-particle dispersion demonstrates heterogeneity and intermittency in the sea ice deformation field associated with space/time coupling inherent in fracturing of the sea ice cover as

15 described by sea ice mechanics (Rampal et al., 2008; Weiss, 2013; Weiss and Dansereau, 2017). Rampal et al. (2008) noted that a triplet or multiple-particle analysis is in addition necessary to illustrate the deformation and related small-scale kinematic features that develop in sea ice.

Three -particle dispersion and triplet areas in particular, such as are explored in this study, enable a distinction between the individual strain-rate tensor components of divergence, convergence, and shear.

20 Specifically, sea ice divergence depicts open water formation and accompanying processes such as new ice growth, brine rejection to the ocean, and heat and moisture exchange; ice convergence depicts ridge and keel formation thus contributing to ice thickness (Stern and Lindsay, 2009; Kwok and Cunningham, 2012), with implications for ice hazard detection, oil spill and contaminant transport and shipping route assessments. Triplet areas also provide a signature of what is referred to by Thorndike (1986) as

25 "nondivergent diffusive mixing" due to compressibility in the ice cover.

Early studies of oceanic circulation have used multiple particles to monitor small-scale deformation and mixing as opposed to larger-scale stirring mechanisms captured by single-particle dispersion analyses. Ice beacon triplet arrays have also been used to monitor sea ice deformation off the Canadian

east coast and in Antarctica (Prinsenberg et al., 1997; Heil et al., 2002; 2008; 2009; 2011). Studies of correspondence between ice stress, convergence and atmospheric forcing off the southern coast of Labrador in March, 1996 showed little change in convergence within an already compact ice cover, in addition to an increase in stress with winds and decrease in stress with temperature as the icepack loses its ability to transmit pressure (Prinsenberg et al., 1997). These results are consistent with studies of derived ice motion fields using synthetic aperture radar data showing sea ice deformation and production 1.5 times higher in the seasonal than in the perennial ice zone throughout the Arctic in late fall and winter due to differences in ice strength and thickness (Kwok, 2006).

Application of Lagrangian dispersion (single- and two-particle) in the seasonal ice zone in the Beaufort Sea region in past studies showed that single-particle dispersion captures the existence of two distinct dynamical regimes characterized by distinctive scaling laws; $t^2$ scaling in the zonal direction characteristic of advection, and $t^{5/4}$ scaling in the meridional direction characteristic of quasi-geostrophic 2D turbulence (Lukovich et al., 2011). Two-particle dispersion studies in this region, based on an assessment of loop and meander reversal events, demonstrated enhanced meridional separation indicative of ice-ice and ice-coast interactions and increased connectivity in the ice cover in winter relative to spring (Lukovich et al., 2014).

In the present study we build upon these previous analyses to provide, using one-, two-, and three-particle dispersion analyses, a prescription for changes in sea ice drift trajectories and deformation in response to atmospheric forcing and coastal interactions for varying distances from the coastline. In particular, we seek to address the following research questions:

i) What is the correspondence between sudden changes in sea ice drift trajectories, atmospheric forcing, and distance relative to the coastline on daily timescales? [How can individual sea ice drift events be characterized?]

ii) How are sudden changes manifested in deformation characteristics? How do these characteristics vary daily based on orientation (near shore, offshore, and along shore) and distance relative to the coast?

The goal of this paper is to develop a framework for understanding sudden changes in ice drift trajectories in the Beaufort Sea on daily timescales using Lagrangian dispersion statistics to quantify

[revised manuscript text omitted]

| Deleted: The term "shocks" is used interchangeably with sudden change from the perspective of a 'shock-response' mechanism associated with sea ice response to atmospheric forcing and ice-coast interactions on daily timescales that forms the focus for the present study. |

| Deleted: Lagrangian dispersion |

[revised manuscript text omitted]
 and ice-coast interactions based on beacons deployed in a triangular configuration as triplets and Lagrangian dispersion analyses. The diagnostics and tools developed provided a unique characterization of sea ice drift and deformation processes in the southern Beaufort Sea, with implications for ice hazard assessments and forecasting applications.

Sudden changes in sea ice drift on daily timescales provide a signature of sea ice response to atmospheric forcing. In response to our first research question, single-particle dispersion statistics were used to identify sudden changes in ice drift paths, captured also in the three-day running means in variances of triplet centroid positions. Absolute dispersion also highlights the existence of two ice dynamical regimes prior to and following the 8 October SLP high responsible for Ekman convergence, poleward retreat in the ice edge, and resulting in the subsequent loss in both ice-atmosphere synchronicity and spatial coherence in triplet events. Demonstrated also are changes in ice concentrations during sudden changes, with lower ice concentrations for all triplets during the early stages of evolution, and for triplet B following the 8 October SLP high. Sudden changes reflect sea ice drift responses to SLP low and high regimes; regional differences are manifested predominantly in turning angles, the local characterizations of which are explored in an assessment of triplet area, deformation (DKPs), and relative deformation.

In response to our second research question, sea ice deformation is characterized in this study by two- and three-particle dispersion, including evolution in triplet area, perimeter-to-area ratio, the Okubo-Weiss criterion to monitor relative contributions from vorticity (due to external forcing) and deformation (associated with distortion in the ice cover), in addition to the shear-to-divergence ratio as a signature of sea ice stress and strength in the ice cover. Triplet areas depict local responses due to ice strength and drift. Triplet area during sudden changes demonstrates an increase in triplet A near the ice edge, and enhanced stretching closest to the shoreline. DKPs further show that sea ice deformation is weakest furthest from the coastline. Relative DKPs highlight enhanced distortion in the (bounded) ice cover encountered by Triplet B, located closest to the coastline, particularly following the SLP high on 8 October that led to a fragmented ice cover and reduced ice concentrations.

Results from this analysis suggest that sudden changes in the southern Beaufort Sea occur either during i) a reversal in winds that induce onshore/offshore ice drift, or ii) sustained north/easterly winds, with response mechanisms governed by ice conditions and interactions with the coastline. Sudden changes associated with wind reversals (minima in drift variance and inflection points in absolute dispersion; *e1*, *e3*, *e4*, *e6*, and *e8*) occur during SLP gradients. Changes in ice stress monitored by shearto-divergence ratios are associated with the transition from an on or offshore to along-shore ice drift regime. Triplet D, located furthest from the coastline, is uninfluenced by ice-coast interactions.

In summary, single-particle dispersion captures i) a shift in ice dynamical regimes following the 8 October SLP high, ii) inflection points and sudden changes in the meridional direction associated with interactions with the coastline, and iii) loss of coherence in ice-atmosphere interactions to provide a regional characterization of sea ice drift. Two- and three-particle dispersion capture i) sea ice deformation induced by northeasterly winds and accompanying along- and cross-shear transport ii) relative DKPs and enhanced deformation with decreasing distance from the coastline following the 8 October event, and iii) local and regional variations in sea ice deformation in the ice cover relevant for rheological characterizations of sea ice based on an assessment of the shear-to-divergence ratios.

Results from this analysis provide a prescription for identifying sudden changes in the ice drift field and accompanying changes in deformation characteristics based on distance from the coastline and proximity to the ice edge (where ice drifts more freely) relevant for prediction and ice-hazard detection on daily timescales. Proposed future work includes the use of the shock-response parameters, and Okubo-Weiss criterion and shear-to-divergence ratio diagnostics in particular for model-data comparison in applications involving ice hazard detection, forecasting and prediction. Numerical experiments testing sea ice response to sudden changes in ice drift during SLP high and low regimes will provide additional insight into physical mechanisms responsible for the observed local kinematic and deformation features in the ice drift field.

Lagrangian dispersion thus provides a framework and prescription for understanding sea ice dynamics that will enable the construction of an integrated observational-modeling framework designed specifically to understand ice-atmosphere interactions in the context of drift and deformation. Furthermore, an understanding of floe size shapes and distributions prior to, during, and following sudden changes in the ice drift cover using these diagnostics, and shear-to-divergence ratio in particular, will improve our understanding of how the sea ice cover responds to atmospheric forcing based on distance from the coastline. Results from this analysis can be applied to develop an observational-modeling framework for Lagrangian dispersion that monitors sea ice drift and deformation at large,

[revised manuscript text omitted]

**Appendix B: Triplet Observation Report**

A description of observations and results determined from shock-response diagnostics is provided in
5  this appendix as supplementary material to the results section. Results are categorized according to i) shock diagnostics (SLP, turning angle, surface winds, ice drift, SIC and SAT as depicted in Figures 3 to 5) used in identifying trajectory changes in ice drift paths, and ii) response diagnostics (triplet Areas, DKPs, relative DKPs and shear-to-divergence ratios as depicted in Figures 6 to 8) used in identifying sea ice deformation responses during sudden changes in ice drift paths.

10  **B.2.1 Identifying trajectory changes in sea ice drift and deformation**

Figure 3. – Atmospheric forcing and sea ice response – spatial variability

In the context of ice shock events, large turning angles are observed for triplets A, B, and C during a
15  SLP low for shock *e1*. Shock events evident during a transition in SLP phase are also observed in turning angles for all triplets (Figures 2b and 3). Specifically, angle differences on the order of 40 degrees are observed during decreasing SLP for shock *e2*. Turning angles increase during a transition to a SLP high for *e3*, whereas for shock *e4* negative turning angles exist for decreasing SLP. A SLP low is observed between successive SLP highs when triplet A stops recording during *e5*. The triplet D
20  centroid loop event during *e6* is manifested in differences in turning angles during a SLP low, in a manner distinct from other triplet turning angles. Slight differences in turning angles are observed during increasing SLP for *e7*. As for *e6*, differences in turning angles highlight spatial (relative to distance from the coastline) differences in sea ice response to external forcing as a SLP high enters the region in the vicinity of the beacon triplets. SLP mean time series in the vicinity of the triplets indicate
25  that the "shock-response" mechanism occurs with SLP gradients and a transition between local SLP

high and low regimes. As will later be explored, turning angles and shock events additionally provide an initial indication of sea ice mechanics and deformation.

Figure 4. – NARR winds, sea ice drift and orientation – spatial variability

A southerly to northerly transition in 10 m winds is accompanied by onshore ice drift for triplets A to C, and a second reversal from northerly to southerly winds accompanied by offshore ice drift for triplet D during *e1 (Figure 4)*. Strong northerly flow is observed in the vicinity of all triplets with alongshore (southwestward) drift in triplet A to C centroids, and onshore drift in triplet D during *e2*. Reversals in 10 m winds are observed with accompanying alongshore/onshore ice drift during *e3*. An interruption to the centroid trajectory is further observed during reversals in 10 m winds during *e4*, with along-shore and predominantly onshore drift for triplet D. By contrast, dominant northerly winds result in along-shore flow for all centroids during *e5*. Reversals in 10 m winds in the vicinity of triplets B and C with accompanying alongshore flow, and predominantly southerly winds in the vicinity of triplet D during a loop event highlight regional and local responses during *e6*. Northeasterly winds and alongshore ice drift for triplets B and C, and onshore drift for triplet D further highlights local responses during *e7*. Differences in winds and drift response are again observed during shock event *e8*.

Figure 5. – SIC range and SAT – spatial variability

In consideration of shock events, SIC values of ~80% with SAT less than zero degrees Celsius are observed for triplets A and B; SIC values of ~80% and ~90% with SAT less than two degrees Celsius are observed for triplets C and D respectively during *e1*. SIC ranges from 90 – 100 % and SAT is less than two degrees Celsius for all triplets during *e2*. Similarly during *e3*, SIC ranges from 95 – 100 % with SAT < 0 for all triplets. SIC approaches 100% concentration with SAT < 0 for shock events *e4* and *e5*. Lower ice concentrations for *e6* followed by an increase in SAT suggest an opening of leads in the vicinity of triplets B and C. A delayed response is observed for triplet D. During *e7* equilibration in SAT is accompanied by SIC values ranging from 90 – 100% for triplets B, C, and D. By contrast,

during *e8* SIC values are less than 80% as SAT increases for triplet B near the ice edge, while triplets C and D approach 100% ice concentration as SAT equilibrates and continues to decrease, respectively.

**B.2.2 Identifying responses in sea ice deformation**

Figure 6 Triplet area evolution

An assessment of evolution in triplet area during each of the sudden changes demonstrates an increase in triplet A area near the ice edge in contrast to decreasing areas for all other triplets during *e1*, with a
10  lagged response in the maxima (Figure 6). The greatest stretching is also observed closest to the coastline. During *e2*, the base for triplets A and B increases during consolidation following which a maximum is achieved on the $21^{st}$ (~ 2-day time lag). By contrast, P/A and b/h are approximately constant for triplets C and D, indicating that the triangle maintains its shape further from the continental coastline. Triplets A, C, and D maintain their shape during *e3*, while triplet B area decreases as the
15  height decreases (P/A and b/h increase). During *e4*, P/A and b/h are constant for triplets A, C, and D with higher values further from the coastline (associated with smaller areas). A decline in P/A and b/h due to restoration in height is observed for triplet B. During *e5*, the area and ratios are approximately constant for all triplets, following which triplet A beacons stop recording. The greatest stretching (high base and ratio values) is observed in triplet B located closest to the shoreline/continental coastline. The
20  area in triplet B decreases during *e6* associated with a decrease in height, resulting in increasing b/h and P/A, while triplets C and D are approximately constant.  During *e7* an increase in area for triplet B is associated with an increase in height so that b/h and P/A decrease, while once again triplet C and D exhibit comparable behavior.   During *e8*, triplet B approaches an equilateral configuration as the beacons approach the ice edge; increased stretching is observed for triplet C relative to triplet D
25  continues. P/A and b/h thus provide a signature of stretching and ice drift response to the coastline, the relative contributions of which are further described by deformation and DKPs.

Figure 7. Sea ice deformation

Sea ice deformation during sudden changes highlights differences in sea ice response based on distance from the continental coastline. In particular, during *e1* triplet A is characterized by dominant contributions from vorticity, shear, and to a lesser extent, divergence, whereas triplet B witnessed a decline in shear and vorticity. Triplet C witnessed a decline in shear and transition to predominantly anticyclonic activity accompanied by comparatively weak stretching, while triplet D experienced a transition to convergence and negative vorticity. During *e2*, Triplet B experiences a transition from vorticity and stretching to shear, while triplet C is governed by shear. DKPs for triplets A and D are comparatively weak. During *e3*, only triplet B exhibits a transition from shear to anticyclonic activity, in contrast to comparatively weak DKPs for all other triplets. It should be noted that a negative stretching deformation rate indicates stretching along the y-axis (meridional) and shrinking along the x-axis (zonal). Triplet B during *e4* is characterized by a transition from cyclonic activity to stretching along the x-axis, while triplet D is distinguished by vorticity and shear compared to triplets A and C, which exhibit weak deformation. Similarly during *e5*, weak DKPs are observed for triplet C, with triplets B and D governed by vorticity. During *e6*, triplet B is governed by meridional stretching, followed by zonal stretching and oscillatory motion in N until *e7* and 20 October, following which convergence and divergence dominate. During *e8*, triplet B is governed by vorticity associated with reversals in drift orientation, while triplet C is governed by weak DKPs, following which alternating cyclonic and anticylonic circulation is observed.

Figure 8. Relative sea ice deformation.

It is interesting to note that extrema in the Okubo-Weiss criterion roughly coincide with sudden changes in ice drift (Figure 8). In particular, during *e1*, triplet A is characterized by vorticity, while triplets C and D are characterized by deformation-dominated flow, and triplet B by shear (Figures 7 and 8). During *e2*, triplet A is governed by deformation (S and N), triplet B by divergence, triplet C by shear (Figures 7 and 8, and shear-to-divergence values ~90), and triplet D by divergence. During *e4*, triplet B is

governed by divergence, while during *e6*, triplet B is governed by deformation (N; Figures 7 and 8), and triplets C and D by divergence.  During *e7*, triplet B is governed by divergence, as are triplets B to D to a lesser extent during *e8*.

---

## Author Comment (AC2) · 31 Jan 2017

Please find below a list of responses to reviewers' comments regarding the submitted manuscript, "Characterizing sudden changes in Arctic sea ice drift and deformation on synoptic timescales", by J.V. Lukovich, C.A. Geiger, and D.G. Barber. General comments are presented, followed by responses to reviewers' specific comments and suggestions. Please note that changes implemented may also be found in the attached supplementary material in response to reviewers' suggestions.

General comments:

The authors would like to thank both reviewers for their constructive comments and suggestions. In consideration of reviewer recommendations, and the initial motivation for the triplet analysis, the authors have in the revised manuscript presented a framework based on Lagrangian dispersion, and single-, two-, and three-particle dispersion statistics in particular to provide a quantitative analysis and more focused narrative of sudden changes in ice drift paths as well as associated changes in sea ice deformation, based on distance from the coastline.

Please find below specific comments to suggestions and recommendations.

Specific comments:

A. Provenzale (Referee) antonello.provenzale@cnr.it

Dr. Provenzale:

Thank you for your comments and suggestions. In light of your comments and those of the first reviewer, the paper has been revised to provide a more quantitative and focused interpretation of Arctic sea ice dynamics based on Lagrangian dispersion statistics. As is noted in the first comment to the first reviewer, emphasis in the revised manuscript is on the development of a Lagrangian framework based on single-, two-, and three-particle dispersion statistics to quantify sudden changes in ice drift and associated deformation in response to atmospheric forcing for varying distances from the coastline.

Please find below more specific responses to your queries.

The paper "Characterizing sudden changes in Arctic sea ice drift and deformation on synoptic timescales" discusses the use of Lagrangian triplet dynamics, combined with characterizations such as the Okubo-Weiss parameter, to identify "sudden changes" in sea ice drift in the Arctic.

The material is interesting and it builds upon previous works by the same lead author. However, I find the paper rather difficult to read, and not very clear in its message. First, most of the figure are simple displays of time series, without too much statistical analysis and/or quantitative interpretation. The paper would benefit from a more

quantitative approach, with results of the statistical analyses, to assess the validity and significance of the conclusions.

The authors agree that the original version of the manuscript was qualitative in nature. In response to both reviewers' comments, the authors have revised the manuscript to provide a framework for examining sea ice dynamics based on Lagrangian dispersion statistics, and one-, two-, and three-particle dispersion in particular. Results from single-particle dispersion (Figure 3) highlight a transition in dynamical regimes evident in a change in slopes following 8 October, 2009 during which a SLP high induced strong Ekman convergence, offshore ice drift, and subsequent deterioration in the ice cover near the ice edge. Two-particle dispersion (Figure 9) also illustrates differences in zonal and meridional separation due to along- and cross-shear transport associated with interruptions to anticyclonic circulation of the Beaufort Gyre. Scatter plots (Figure 15) indicating the frequency of events falling within wind and DKP bins for varying distances from the coastline also illustrate differences in sea ice response to atmospheric forcing and associated deformation processes for varying distances from the coastline.

I also urge the authors to streamline the paper, making it more palatable and understandable.

The paper has been rewritten to provide a more focused interpretation of sudden changes in ice drift paths based on the Lagrangian dispersion approach. Single-particle dispersion is shown to capture sudden changes in sea ice drift, and a transition in dynamical regimes following the incursion of a SLP high into the region that induces Ekman convergence in the ice drift field. Two- and three-particle dispersion are shown to capture deformation associated with ice drift events due to northerly winds, and deterioration in the ice cover following the 8 October SLP high responsible for loss of spatial coherence in the ice cover and synchronicity in ice-atmosphere interactions.

In particular, I would like to add a paragraph at the beginning of the Introduction explaining some more fact about Arctic sea ice and sea ice drift.

Thank you for this suggestion. In consideration of your comments, and to provide additional context for this study, the authors have included an introductory paragraph on Arctic sea ice dynamics and its role in understanding changes in the sea ice cover, as follows:

"Central to our understanding of changes in the Arctic sea ice cover in response to a changing climate and continued anthropogenic forcing is an understanding of sea ice drift and deformation, namely sea ice dynamics. Accelerated ice drift speed over the past several decades reflects a weaker and more mobile ice cover associated with the loss of multiyear ice and changes in atmospheric circulation (Hakkinen et al. 2008; Barber et al., 2009; Rampal et al. 2009b; Spreen et al., 2011; Kwok et al., 2013). Sea ice deformation, or spatial gradients in the ice drift field, associated with opening and closing in the ice cover due to sea ice divergence and convergence, influences moisture and heat exchange between the ocean and atmosphere, ice ridging, sea ice thickness and redistribution (Hutchings et al., 2011, Bouillon et al., 2015) with implications ice hazard detection, and pollutant and contaminant transport. In the Beaufort Sea region, sea ice dynamics is governed by large-scale anticyclonic circulation of the Beaufort Gyre, with reversal to cyclonic circulation throughout the annual cycle (LeDrew et al., 1991; Preller and Posey, 1989; Proshutinsky et al., 2015)."

Finally, it is not clear what "sudden changes" in sea ice are, and to what meteorological/ climatic events are related. This point should be further explored and clarified.

The authors agree with both reviewers' comments that the definition for sudden changes was not clear in the initial version of the manuscript. In the revised version, as is noted in the response to the first reviewer, sudden changes are defined at the beginning of the manuscript as changes in the ice drift path (page 2, line 20), and quantified as minima in the ice drift variance and inflection points in single-particle dispersion. In addition, reference is no longer made to the "shear-shock event". Single-particle dispersion analyses demonstrate a transition in the sea ice dynamical regime on 8 October, 2009, associated with a SLP high and strong Ekman convergence and

offshore ice drift. Distinction is also made between sudden (ice drift) events associated with wind reversals (e1, e3, e4, e6, e8), and those associated with persistent northerly winds (e2, e5, and e7) through evaluation of two- and three-particle dispersion statistics.

Thank you once again for helpful comments and suggestions.

Please also note the supplement to this comment:
http://www.the-cryosphere-discuss.net/tc-2016-219/tc-2016-219-AC2-supplement.pdf

[Figure]

**Fig. 1.** Figure 3 (revised version): Absolute (single-particle) dispersion statistics for triplets A to D, depicting zonal (red), meridional (blue), and total (black) dispersion.

[Figure]

**Fig. 2.** Figure 4 (revised version): Absolute (single-particle) dispersion statistics depicting meridional (top), zonal (middle), and total (lower) dispersion to characterize local changes in ice drift.

[Figure]

**Fig. 3.** Figure 9 (revised version): Relative (two-particle) dispersion showing meridional (top), zonal (middle), and total (lower panel) relative dispersion as a function of elapsed time.

[Figure]

**Fig. 4.** Figure 15 (revised version): Scatter plots of NARR winds versus DKPs for triplets A to D showing density of values in wind and DKP bins. Symbols depict sudden change events.

---

## Author Response (AR2)

**Responses to Editor's Commnets**

5 Dear Jenny:

Thank you for your comments, edits, and suggestions, which highlight shortcomings and provide additional insight on issues requiring additional clarification. In response to your comments, the authors have attempted to focus on main points illustrating the role of single-, two-, and three-particle dispersion in quantifying sea ice drift and deformation. The authors have also included a brief description of the shear-to-divergence ratio in the context of strain rate components using diagnostics from three particle dispersion, and a residual analysis for divergence to determine the conditions for which elongated triplets do not provide an accurate representation of DKPs. Please find below responses and comments to your queries:

15 Editor Decision: Reconsider after major revisions (10 Mar 2017) by Jennifer Hutchings

Comments to the Author: Dear Jennifer, Cathy and David,

20 Overall, this paper is much improved upon the first version submitted. It is much clearer now which physical phenomena you are investigating. Thank you for cleaning up the terminology in the paper referring to "sudden changes".

You have added much to the paper. The addition of the comparison between methods of calculating dispersion and deformation is very interesting, and could actually result in further discussion on later papers. Unfortunately the shear volume of description of results in the paper does distract from what your main points might be.

Your addition of the ratio of divergence to shear rates in response to reviewer 1's comments is potentially interesting. I do have some specific comments, below, regarding this. In particular pay attention to the fact that 30 you are only considering strain-rate. Without information about the forcing or material properties you can not comment on yield stress. I am pretty sure reviewer 1 understands this, I would just make it clear in the text that you have to make assumptions in your interpretation of this.

While the paper is much clearer, and I think I am understanding there are interesting findings, you need to make 35 this much clearer in your text.

Please consider this comment: "I also urge the authors to streamline the paper, making it more palatable and understandable."

- 40 On reading the paper I see that there is still a need to streamline the paper. My suggestion is that there are very interesting phenomena you show, yet this is not clear in your results and discussion sections. Perhaps choose the two most interesting phenomena and focus the writing on those. Or focus the writing on the phenomena that best make the point that your methodology is useful.
- 45 The authors have attempted in the revised manuscript to streamline the paper. The eight events have been retained, with emphasis on the 8 October SLP high and its impact on sea ice dynamical regimes.

I agree with reviewer 1 that the dataset does provide information that could be important. This just needs to be made clear.

50

55

"I understand the importance of this topic and find that it may be useful to understand the dynamical features of sea ice near the shore on a daily time scale and on a sub-grid spatial scale. And that must be what the author aims at in this paper. I see that the authors attempted to make the most of available several buoy data to reveal them, and their efforts should be appreciated. However, I feel the manuscript is a bit more descriptive and the conclusion does not necessarily seem clear. "

I believe you can still hone the paper to write towards your conclusions more concisely. If you can do this it will be much easier to get a positive review.

Effort has been made by the authors to provide a more concise description of findings in the results section based

on single-, two-, and three-particle dispersion in the context of sea ice drift and deformation, with insight provided in the discussion section.

I would like to send this paper out to review in a form that will showcase this unique study's most interesting 5 findings. There are some useful concepts in here that could improve upon previous methodologies investigating large scale ice mechanics, so I do feel there is potential for this paper to have impact.

Some specific suggestions would be to reconsider your motivation for the paper, you need to introduce why your questions are of interest. Consider your conclusions and the new methodology you have developed. Perhaps rethink how you introduce these. So you find dispersion and strain-rate varies with distance from coast. This is not a new finding, however the context you bring of considering the changing dispersion behavior between particular events related to changes in wind forcing is useful.

In the revised manuscript the authors have attempted to clarify the motivation by highlighting the role of single-, two-, and three-particle dispersion in describing sea ice drift and deformation through characterization of directional changes in sea ice drift, and sea ice response to atmospheric forcing in the context of deformation and DKPs.

Specific Comments:

20

Line 10: a conjunction is missing between implications and ice hazard.

This has been corrected. Thank you.

25 page 2 Line 23-24: Can you be a little more precise in your description of the force balance? I assume by sea ice motion you mean the inertial term?

This sentence has been modified to highlight salient features associated with ice-atmosphere interactions, and now reads as follows:

30 "Arctic air-ice-sea interactions on synoptic timescales (several days to weeks) are governed by three interactive components: i) sea ice motion, ii) a confining coastline, and iii) atmospheric forcing."

page 3 line 9-10: This is the first place you have introduced elliptic and hyperbolic in this paper. It would be easier for a reader to follow if you first describe the mechanical properties

35 of the ice pack, such as homogeneous and discrete, and then introduce the mathematical concepts through a one line discussion of the fact that the equations describing each behavior

are elliptic or hyperbolic. I do see that you introduce this concept nicely near the bottom of page 3.

In order to avoid confusion and to preserve the initial organization of the manuscript and 40 Introduction, this text has been removed in the revised version of the manuscript.

Page 7, line 20: consider grammer here. I think there is a missing "they are"

This has been changed. Thank you. 45

Page 12, line 10: you need to more explicitly state what this angle tells you about ice stress. It is only a signature of ice stress if you assume a constant flow rule and yield behavior. It would help a casual reader if you can explain 'rheology'. You actually do not need to invoke any discussion of rheology to make the last sentence in this paragraph.

50

Based on your comments, reference to rheological characterization of sea ice using the shear-to-divergence ratio is removed in the revised manuscript. Instead, the authors note briefly the role of this ratio in describing ice mechanical properties and ice strength type.

55 Rheology discussion on page 21: Strain-rate can not constrain ice stress. What you do show is that in convergence ice failure is followed by opening. Does theta change as well? This is a nice little diagnostic to identify when failure happens for particular modes of failure.

Theta does change for triplets B and C following sea ice convergence associated with the 8

60 October SLP high. In particular, θ values of 180 are followed by values on the order of 0, following the 8 October SLP high for triplet B, with time lags ranging from 4 to 2 days.

page 21 line 8: use greek symbol for theta.

**This has been changed, thank you.**

page 23 Line 2: You need to be precise when you say rheological characterizations. You are not actually assessing strength, but you are assessing the kinematics and hence can identify particular modes of failure. The ratio of shear to divergence gives you a non-conclusive indication of where on a yield curve of arbitrary shape you lie (you can not determine the shape without a lot of data), and you have to assume no dilation in the flow rule or a constant predictable flow rule. I am not saying this is not a useful piece of evidence, it just has to be taken in context of the forcing.

The authors agree that reference to rheological characterizations in the context of the shear-to divergence ratio is misleading, and have there removed this reference throughout the manuscript. The authors have instead focused on this diagnostic as a tool to characterize ice strength type and strain components.

15 I am not convinced you addressed the comment about accuracy of strain-rate reducing as the triangles become more elongated. It would help to show the math in the response to reviewer. Essentially as the triangles become skewed, their area decreases and therefore they do not represent the regional deformation. Reviewer 1's point, which you reiterate is that the smaller deforming triangles have greater deformation rates, so it is difficult to compare to larger triangles and hard to interpret the time series of deformation events.

The authors have sought in the revised manuscript to address concerns regarding the inability of an elongated triangle to provide an accurate characterization of DKPs, following an approach similar to that presented in Ohlmann et al. (2017). In particular, error is estimated based on a scatterplot of divergence residuals versus the aspect ratio h/b, and a minimum threshold for the aspect ratio determined for the residual range [ $\delta_{max}/10,2^*$

25 aspect ratio h/b, and a minimum threshold for the aspect ratio determined for the residual range [ $\delta_{max}/10,2^* \delta_{max}/10$ ]. Intervals during which the aspect ratios exceed threshold values are excluded from the three-particle dispersion and DKP analyses.

Figure 2: identify the bars in the bottom panel.

The bars in the bottom panel are now identified in the Figure caption through the following statement: "The bars in the bottom panel depict the triplet events associated with minima in the total variance."

Figure 4a: labels are too small to read. This is true for several figures. A rule of thumb: make 35 the labels the same scale as the text in the caption.

The font labels have been increased.

30

60

Figure 6: Why are the event numbers coloured black and red? In subsequent figures you place 40 the arrows for event numbers, but no labels.

The event numbers are labeled in black and red to distinguish directional changes due to wind reversals (black), from directional changes due to persistent northerly winds (red).

45 Figure 11: it is figure 6 that shows the same color bars, not 7.

Thank you for pointing this out. This has been corrected in Figures 10 and 11.

Figure 15: This is rather confusing. What are the four rows in the figure? I am not sure I can pull out what you are trying to use this figure for from the paper.

This figure has been removed from the revised version of the manuscript, as it did not contribute to the manuscript objectives or narrative.

55 Some more general comments:

While you have nicely addressed many of the reviewers comments, this paper has become exceptionally long. Do they need all the figures? Can figure 1a and 1b be combined into one panel for example. I am not sure you need Figure 8 for the discussion in the paper. It is acceptable to describe the deployment location and ice conditions in text as you did. I will admit, I got lost in the discussion of particular events in relation to the dispersion and deformation. I need help drawing between this overly detailed discussion and the conclusions.

It is much clearer in this revision why you are considering deformation events. These do present a way to study

response of the ice pack to changes in forcing. It might actually be simpler to introduce your study in this more basic way, as I think this is what reviewer one and others actually see as the strength of splitting the time series into events.

- 5 Here are some highlights of the study in my mind: You have inter-compared three ways of characterizing dynamics of the ice pack. You are presenting a methodology to find changes in forcing in the dynamic response of the ice pack, and how this relates to dispersion characteristics. You have related changes in dispersion as differing response to wind forcing and the confinement on the ice. This is useful because it provides tools with which to identify timing of shock events, such as failure, and the transitions in dispersion related to these.
  10 Looking at figure 12, the large range in DKPs is indicative of arrays that have very small area.
- 10 Looking at figure 12, the large range in DKPs is indicative of arrays that have very small area. The time series are difficult to interpret because they are swamped by large values when the array is highly sheared. It is actually quite hard to determine how the direction change events line up in this time series with the one and two particle dispersion time series.
- 15 My main concern is that it is very hard for the reader to follow your findings, and understand what the particular observations of different ice drift direction changes are telling them. I would urge you to sharpen the manuscript further. Remove unnecessary descriptions and speak clearly to the finding that the enhanced shear near the coast happens during particular deformation events, that you here describe as direction changes. I want to see your work presented in the best light, so I hope you will revise the manuscript before it is sent out for econsideration by the reviewers. I also would urge you to have someone less familiar with the paper proof read

With best regards,

Jenny 25

30

it.

Thank you for your comments and suggestions. The authors agree that the earlier version of the manuscript was too long and distracted from the central narrative, namely the use of single-, two-, and three-particle dispersion to characterize sea ice drift and deformation in response to atmospheric forcing for varying distances from the coastline. Figures 1a and 1b can be combined. In addition, Figure 8 (now Figure 9) has been removed and replaced with the criterion for DKP validity assessment. In addition, the manuscript focuses on the 8 October SLP high event, with reference to events *e1* to *e8* in the context of this event and manuscript objectives.

The authors have sought in the revised version of the manuscript to highlight that three-particle dispersion and DKPs through evaluation of the Okubo-Weiss criterion (deformation versus vorticity-dominated flow), and the shear-to-divergence ratio (which provides additional information on strain components) provide insight into sea ice cover response to external forcing.

In response to the Editor's constructive comments and suggestions, the authors have focused the manuscript and its narrative to highlight key findings resulting from the single-, two-, and three-particle methodology, namely

- 40 the role of single-particle dispersion in capturing a shift in dynamical regimes prior to and following the 8 October SLP high based on a regional characterization, and changes in ice state based on a local characterization; twoand three-particle dispersion in capturing sea ice deformation and response to external forcing, and further distinguishing deformation components based on evaluation of DKPs, and strain components based on the shearto-divergence ratio. These results are summarized in Tables 1 to 3 to provide additional clarity on the use of
- 45 these diagnostics to understand changes in sea ice dynamics and state in response to external forcing for varying distance from the coastline. Implications of this analysis (in terms of Levy walks and failure mechanics) are also noted to highlight the motivation for this manuscript and methodology as a consistent framework which, through deployment of a beacon array, can provide insight into sea ice drift, deformation, and state.

50 Thank you once again for instructive and helpful advice and comments.

Best regards,

55 Jennifer

**Method to characterize directional changes in arctic sea ice drift and associated deformation due to synoptic atmospheric variations using Lagrangian dispersion statistics**

J.V. Lukovich1, C.A. Geiger2, D.G. Barber1

1Centre for Earth Observation Science, University of Manitoba, Winnipeg, R3T 2N2, CANADA 2College of Earth, Ocean, and Environment: Geography, University of Delaware, Delaware, 19716, USA *Correspondence to*: J.V. Lukovich (Jennifer.Lukovich@umanitoba.ca)

Abstract. A framework is developed to assess the directional changes in sea ice drift paths and associated deformation processes in response to atmospheric forcing. The framework is based on Lagrangian

- statistical analyses leveraging particle dispersion theory which tells us whether ice drift is in a subdiffusive, diffusive, ballistic, or superdiffusive dynamical regime using single-particle (absolute) dispersion statistics. In terms of sea ice deformation, the framework uses two- and three-particle dispersion to characterize along and across-shear transport as well as differential kinematic parameters. The approach is tested with GPS beacons deployed in triplets on sea ice in the southern Beaufort Sea at varying distances from the coastline
- 15 in fall of 2009 with eight individual events characterized. One transition in particular follows the SLP high on 8 October in 2009 while the sea ice drift was in a superdiffusive dynamic regime. In this case, the dispersion scaling exponent (which is a slope between single-particle absolute dispersion of sea ice drift and elapsed time) changed from superdiffusive (α ~ 3) to ballistic (α ~ 2) as the SLP was rounding its maximum pressure value. Following this shift between regimes, there was a loss in synchronicity between sea ice drift and atmospheric motion patterns. While this is only one case study, the outcomes suggest similar studies be conducted on more buoy arrays to test momentum transfer linkages between storms and sea ice responses as a function of dispersion regime states using scaling exponents. The tools and framework developed in this study provide a unique characterization technique to evaluate these states with

respect to sea ice processes in general. Application of these techniques can aid ice hazard assessments and weather forecasting in support of marine transportation and indigenous use of near-shore Arctic areas.

| Deleted                                                                                                  |                                                                                                                   |
|----------------------------------------------------------------------------------------------------------|-------------------------------------------------------------------------------------------------------------------|
| veletea:                                                                                                 | in the scaling exponent                                                                                           |
| Formatt                                                                                                  | ed: Not Highlight                                                                                                 |
| Deleted:                                                                                                 |                                                                                                                   |
| Deleted:                                                                                                 | of                                                                                                                |
| Deleted:                                                                                                 | in the southern Beaufort Sea                                                                                      |
| Deleted:                                                                                                 | with implications for                                                                                             |
| Deleted:                                                                                                 | applications required                                                                                             |
| Deleted:                                                                                                 | by                                                                                                                |
| Deleted:                                                                                                 |                                                                                                                   |
| Deleted:                                                                                                 | •[1]                                                                                                              |
| Deleted:                                                                                                 | Central to our understanding                                                                                      |
| Deleted:
response to                                                                                  | of changes in the arctic sea ice cover in
a changing                                                           |
| Deleted:                                                                                                 | and                                                                                                               |
|                                                                                                          |                                                                                                                   |
| Deleted:                                                                                                 | continued                                                                                                         |
| Deleted:
Deleted:                                                                                     | continued is an                                                                                                   |
| Deleted:
is an
, namely                                                                                    |
| Deleted:
is an
, namely
sea ice                                                                         |
| Deleted:
is an
, namely
sea ice
s                                                                    |
| Deleted:
is an
, namely
sea ice
s
Accelerated                                                     |
| Deleted:
is an
, namely
sea ice
s
Accelerated
reflects a                                       |
| Deleted:
is an
, namely
sea ice
s
Accelerated
reflects a
and more mobile                    |
| Deleted:
is an
, namely
sea ice
s
Accelerated
reflects a
and more mobile
associated with |

25

**1** Introduction

30 over the past several decades as a result of a weaker ice cover. The weaker ice cover is due to the loss of Delet thicker - and therefore stronger - multiyear ice which has been exported from the Arctic Ocean through the Delet

Quantifying a transitioning global climate with respect to associated anthropogenic forcing, impacts, and resiliency requires in depth understanding of sea ice drift and deformation; specifically the connections to critical air-sea dynamic exchanges. Current levels of connectivity begin with an accelerated ice drift speed

Fram Strait as a result of changing polar atmospheric circulation patterns (Hakkinen et al. 2008; Barber et al., 2009; Rampal et al. 2009b; Spreen et al., 2011; Kwok et al., 2013). The changing drift patterns have subsequently altered ice-ice interactions which are associated with the opening of leads and closings into pressure ridges and rubble fields. Such features change in location and shape during sea ice deformation processes; specifically divergence and convergence events. These changing deformation patterns in turn impact moisture and heat exchange between the ocean and atmosphere by way of the sea ice thickness and

redistribution (Hutchings et al., 2011, Bouillon et al., 2015). All of these changing patterns impact international shipping, indigenous use of near-shore Arctic areas, and pollutant/contaminant transport.

5

- One area of particular importance to a number of stakeholders is the Beaufort Sea region\_located just\*
   north of Canada and the United States (IPCC, 2015), Here, the large-scale ice-covered oceanic Beaufort
   Gyre circulation pattern is predominantly anticyclonic (clockwise) with reversals to cyclonic (counter
   clockwise) circulation throughout the annual cycle in response to changing surface wind patterns and
   oceanic responses to coastal boundaries (LeDrew et al. 1991; Preller and Posey, 1989; Lukovich et al.,
   2011; Proshutinsky et al., 2015). When circulation patterns change, the ice cover changes abruptly, which
- 15 impacts navigation channels as a result of ice-ice and ice-coastline momentum and energy flux exchanges (e.g., The Polar Group, 1980; Hwang, 2005; McPhee, 2012), air-sea heat exchanges increase (e.g., Carmack et al., 2015), with newly opened leads venting large amounts of moisture into the atmosphere as a strong mass exchange process (Bourassa et al., 2013). Understanding how these changes develop and relate to the orientation of a coastline is essential when diagnosing response patterns. For clarity, directional
- 20 change in this study refers to changes in the ice drift path relative to storm tracks, which have typical duration on the order of days and recurrence rates on the order of several days to weeks.

Arctic air-ice-sea interactions on synoptic timescales (several days to weeks) are governed by three interactive components: i) sea ice motion, ii) a confining coastline, and iii) atmospheric forcing. Previous studies have examined sea ice drift and deformation response to atmospheric forcing and coastline

- 25 geometry on varying timescales (Overland et al., 1995; Richter-Menge et al., 2002; Geiger and Perovich, 2008; Hutchings et al., 2011). In an assessment of springtime sea ice drift in a region to the west of the Antarctic Peninsula, Geiger and Perovich (2008) identified low-frequency motion in response to atmospheric forcing and coastal geometry associated with regional-scale transport, and higher-frequency near-inertial oscillatory motion associated with mixing. On regional and synoptic scales, Richter-Menge et al., 2011.
- al. (2002) also distinguish between translational and differential motion associated with shear zones and
   discontinuities in the ice drift characteristics in the southern Beaufort Sea.

The role of forcing (wind stress) and coastline geometry in establishing coherence in leadpatterns/fractures in the ice cover captured by sea ice deformation has also been explored in past studies

| Deleted: changes in                                                                                                                                                                                                                                                    |
|------------------------------------------------------------------------------------------------------------------------------------------------------------------------------------------------------------------------------------------------------------------------|
| Deleted: Sea ice deformation, or spatial gradients                                                                                                                                                                                                              |
| ni the ice drift field                                                                                                                                                                                                                                                 |
| Deleted: in the ice cover due to                                                                                                                                                                                                                                       |
| Deleted: In the fee cover due to                                                                                                                                                                                                                                       |
| Deleted:                                                                                                                                                                                                                                                               |
| Deleted: ,                                                                                                                                                                                                                                                             |
| Deleted: innucices                                                                                                                                                                                                                                                     |
| Deleted: , ice haging,                                                                                                                                                                                                                                                 |
| Deleted: whit implications for                                                                                                                                                                                                                                         |
| Deleted: and                                                                                                                                                                                                                                                           |
| Deleted:                                                                                                                                                                                                                                                               |
| Formatted: Indent: First line: 0.63 cm                                                                                                                                                                                                                                 |
| Deleted: In                                                                                                                                                                                                                                                            |
| Commont [1]: Succest IDCC former have t                                                                                                                                                                                                                                |
| Comment [1]: Suggest IPCC reference here to
link last paragraph to science mission opening in this
paragraph.                                                                                                                                             |
| REF: IPCC (2015) Climate Change 2014: Synthesis
Report. Contribution of Working Groups I, II and III
to the Fifth Assessment Report of the                                                                                                                       |
| Intergovernmental Panel on Climate Change [Core                                                                                                                                                                                                                        |
| Writing Team,
R.K. Pachauri and L.A. Meyer (eds.)]. IPCC,                                                                                                                                                                                                           |
| Geneva, Switzerland, 151 pp.                                                                                                                                                                                                                                           |
| Deleted: REF).                                                                                                                                                                                                                                                         |
| Formatted: Not Highlight                                                                                                                                                                                                                                               |
| Deleted: sea ice dynamics is governed by large-
scale                                                                                                                                                                                                        |
| Deleted: circulation of the Beaufort Gyre,                                                                                                                                                                                                                             |
| Deleted: to                                                                                                                                                                                                                                                            |
| Deleted:                                                                                                                                                                                                                                                               |
| Deleted: on polar seas                                                                                                                                                                                                                                                 |
| Deleted: ,                                                                                                                                                                                                                                                             |
| Deleted: are altered                                                                                                                                                                                                                                                   |
| Comment [2]: Suggest the following paper referenced here.                                                                                                                                                                                                       |
| http://www.tandfonline.com/doi/full/10.1080/07055
900.2014.890921?src=recsys                                                                                                                                                                                        |
| Deleted: and                                                                                                                                                                                                                                                           |
| Deleted: high                                                                                                                                                                                                                                                          |
| Deleted: (REF)                                                                                                                                                                                                                                                         |
| Formatted: Highlight                                                                                                                                                                                                                                                   |
| Formatted: Highlight                                                                                                                                                                                                                                                   |
| Deleted: a force balance consisting of                                                                                                                                                                                                                                 |
| Deleted: Large-scale (100 km) sea ice circulation patterns are characterized using the eccentricity of an elliptical yield curve where the ratio of divergence to shear is equal to 2 (i.e., divergence is twice as large as shear). However, realistic results |
| are also produced at the 10-km-scale including the resolution of linear kinematic features when one takes the same rheology construct but simply change the eccentricity to 0.7 (Tremblay - personal                                                                   |
| are also produced at the 10-km-scale including the resolution of linear kinematic features when one takes the same rheology construct but simply change the eccentricity to 0.7 (Tremblay - personal communication). Experiments like these oper                       |

(Overland et al., 1995; Hutchings et al., 2005, 2011). Overland et al. (1995) demonstrated that in the Beaufort Sea for spatial scales: i) exceeding 100 km the sea ice cover moves as an aggregate; ii) less than 100 km the ice cover moves as an aggregate or discrete entity; and iii) on the order of 1 km the ice cover is characterized by floe (ice-ice) interactions. Through analysis of a nested beacon configuration and array

- with spatial scales ranging from 10 km to 140 km as part of the late winter (April) 2007 Sea Ice 5 Experiment: Dynamic Nature of the Arctic (SEDNA) campaign in the Beaufort Sea, Hutchings et al. (2011) demonstrated coherence between 140 km and 20km divergence arrays for time periods of up to 16 days in March. Over shorter (sub-synoptic) timescales from May 2007 onward, nested buoy arrays captured the loss of connectivity in the sea ice cover associated with the winter-to-summer transition during a substantial
- 10 ice-loss year (Stroeve et al., 2008). Building upon these earlier findings, an effective next step is to characterize the time-evolving relationships of an ice flow field by leveraging the physics principles in Lagrangian dispersion statistics, which is the focus of this work beginning with a summary of the fundamental principles below.

**1.1 Lagrangian dispersion statistics 15**

Traditionally used to characterize patterns and structure in atmospheric and oceanic dynamical phenomena, Lagrangian dispersion statistics identify topological and dynamical features within a flow field (LaCasce, 2008 and references therein). A number of previous studies already exist using Lagrangian dispersion and ice beacon trajectories to quantify sea ice drift and deformation in the Arctic (Colony and Thorndike, 1984,

1985; Rampal, 2008, 2009a,b, 2016; Lukovich et al., 2011, 2014, 2015). And so, as a review, single-20particle (absolute) dispersion provides a signature of circulation and organized structure in the fluid (or fluid-like) system, and captures linear time-dependence in fluctuating velocity variance characteristic of turbulent diffusion theory (Taylor, 1921; Rampal et al., 2009a), Departures in ice fluctuating velocity statistics from turbulent diffusion are attributed to intermittency associated with sea ice deformation and

**internal ice stress (Rampal et al., 2009). 25**

A two-particle (relative) dispersion analysis monitors sea ice deformation by evaluating time changes in the distances between two ice floe, with each floe typically marked with a GPS beacon. Through evaluation of buoy pair separations as a proxy of strain-rate (divergence, convergence, and shear) components combined, two-particle dispersion provides, by the normal flow rule, an approximation for

30

internal ice stress (Rampal et al., 2009b). Two-particle dispersion also explains heterogeneity and intermittency in the sea ice deformation field associated with space/time coupling inherent in fracturing of the sea ice cover as described by sea ice mechanics (Rampal et al., 2008; Weiss, 2013; Weiss and

| Deleted: based on whether an elliptic
(homogeneous) or hyperbolic (discrete) regime is
established relative to the coastline (ice-coast
interactions)                                                                                                                                                                                                                                                                         |
|----------------------------------------------------------------------------------------------------------------------------------------------------------------------------------------------------------------------------------------------------------------------------------------------------------------------------------------------------------------------------------------------------------------------------------------|
| Deleted: From                                                                                                                                                                                                                                                                                                                                                                                                                          |
| Deleted: we see a need for an approach to                                                                                                                                                                                                                                                                                                                                                                                              |
| Deleted: describe                                                                                                                                                                                                                                                                                                                                                                                                                      |
| Deleted: structure                                                                                                                                                                                                                                                                                                                                                                                                                     |
| Deleted: in                                                                                                                                                                                                                                                                                                                                                                                                                            |
| Deleted: (in this case ice drift) field to quantify changes in the ice cover in the context of sea ice drift and deformation, based on                                                                                                                                                                                                                                                                                          |
| Deleted: and one-, two-, and three-particle dispersions statistics in particular,                                                                                                                                                                                                                                                                                                                                               |
| Deleted: we                                                                                                                                                                                                                                                                                                                                                                                                                            |
| Deleted: on specifically below                                                                                                                                                                                                                                                                                                                                                                                                         |
| Deleted:                                                                                                                                                                                                                                                                                                                                                                                                                               |
| Formatted: Indent: First line: 0 cm                                                                                                                                                                                                                                                                                                                                                                                                    |
| Comment [3]:
Suggest the traditional
Lighthill, M.J. (1978), Waves in fluids,
Cambridge University Press, 504 pp., ISBN 0-
521-29233-6, OCLC 2966533                                                                                                                                                                                                                                                                |
| Deleted: Dispersion statistics is a geophysical
fluid dynamics (GFD) tool tCentral to developing
the tools required to understand sea ice drift and
deformation in response to atmospheric forcing and
ice-coastline interactions are diagnostics and in the
case of drifting buoys, a Lagrangian framework to
quantify spatiotemporal changes in the ice cover.
Traditionally used to characterize dynamic pt[4] |
|                                                                                                                                                                                                                                                                                                                                                                                                                                        |

De De

| Deleted: ,                                         |
|----------------------------------------------------|
| Deleted: thus quantify                             |
| Deleted: moving                                    |
| Formatted: Highlight                               |
| Formatted: Highlight                               |
| Deleted: Previous                                  |
| Deleted: have                                      |
| Deleted: also used                                 |
| Deleted: S                                         |
| Deleted: is a measure of time-dependent     |
| Formatted: Highlight                               |
| Formatted: Highlight                               |
| Formatted: Highlight                               |
| (Deleted: )                                        |
| Deleted: large-scale                               |
| Comment [4]: van Aartrijk, M, Clercx H[7]          |
| Deleted: ; i.e.,                                   |
| Deleted: . departures in ice fluctuating velo      |
| Deleted: . Through evaluation of buoy              |
| Deleted: pair separations as a                     |
| Deleted: linear uniaxial                           |
| Deleted: Uniaxial sea ice strain-rate is a fir [8] |

Dansereau, 2017). Rampal et al. (2008) noted that a triplet or multiple-particle analysis explains deformation and related small-scale kinematic features in sea ice.

Three-particle dispersion, and triplet areas in particular, are explored in this study to distinguish between strain-rate components of divergence/convergence and shear. Specifically, sea ice divergence

5 depicts open water formation and accompanying processes such as new ice growth, brine rejection to the ocean, and heat and moisture exchange. Conversely, ice convergence depicts ridge and rubble formation which contribute to ice thickness redistribution (Stern and Lindsay, 2009; Kwok and Cunningham, 2012) and are further involved in the triplet area phenomenon referred to by Thorndike (1986) as "nondivergent diffusive mixing" due to compressibility in the ice cover. Finally, shear depicts deformation imposed by 10 coastal and ice pack boundaries, such as is found in the Beaufort Sea due to anticyclonic Beaufort Gyre

circulation.

Early studies of oceanic circulation have used multiple particles to monitor small-scale deformation and mixing. Ice beacon triplet arrays have also been used to monitor sea ice deformation off the Canadian east coast and in Antarctica (Prinsenberg et al., 1997; Heil et al., 2002; 2008; 2009; 2011). Studies of

- correspondence between ice stress, convergence and atmospheric forcing off the southern coast of Labrador 15 in March, 1996 showed little change in convergence within an already compact ice cover, in addition to an increase in stress with winds and decrease in stress with temperature as the icepack loses its ability to transmit pressure (Prinsenberg et al., 1997). These results are consistent with studies of derived ice motion fields using synthetic aperture radar data showing sea ice deformation and production 1.5 times higher in
- the seasonal than in the perennial ice zone throughout the Arctic in late fall and winter due to differences in 20ice strength and thickness (Kwok, 2006).

Application of Lagrangian dispersion (single- and two-particle) in the seasonal ice zone in the Beaufort Sea region in past studies showed that single-particle dispersion captures the existence of two distinct dynamical regimes characterized by distinctive scaling laws; t2 scaling in the zonal direction characteristic

of advection, and t5/4 scaling in the meridional direction characteristic of quasi-geostrophic 2D turbulence 25 (Lukovich et al., 2011). Two-particle dispersion studies in this region, based on an assessment of loop and meander reversal events, demonstrated enhanced meridional separation indicative of ice-ice and ice-coast interactions and increased connectivity in the ice cover in winter relative to spring (Lukovich et al., 2014).

In this study we build upon previous analyses to quantify spatiotemporal synoptic changes in sea ice 30 drift and deformation using a novel observational and analytical approach based on one-, two-, and threeparticle dispersion statistics. To address the characterization of sea ice drift and deformation using Lagrangian dispersion statistics, we pose the following research questions:

8

| Deleted: is in addition necessary to illustrate the                                                                                                                                                                                                                                       |
|--------------------------------------------------------------------------------------------------------------------------------------------------------------------------------------------------------------------------------------------------------------------------------------------------|
| Deleted: that develop                                                                                                                                                                                                                                                                            |
| Deleted:                                                                                                                                                                                                                                                                                         |
| Deleted: such as                                                                                                                                                                                                                                                                                 |
| Deleted: ,                                                                                                                                                                                                                                                                                       |
| Deleted: enable a distinction                                                                                                                                                                                                                                                                    |
| Deleted: the individual                                                                                                                                                                                                                                                                          |
| Deleted: tensor                                                                                                                                                                                                                                                                                  |
| Deleted: ,                                                                                                                                                                                                                                                                                       |
| Deleted: ,                                                                                                                                                                                                                                                                                       |
| Deleted: ; i                                                                                                                                                                                                                                                                                     |
| Deleted: keel                                                                                                                                                                                                                                                                                    |
| Deleted: thus                                                                                                                                                                                                                                                                                    |
| Deleted: ing                                                                                                                                                                                                                                                                                     |
| Deleted: ),                                                                                                                                                                                                                                                                                      |
| Comment [5]: removed reference to shipping and
oil spills because that is not the focus here. The
focus here is explaining dispersion theory. Oil spills
and shipping should move to discussion as an
impact. Here you are engaged in character
development.               |
| Formatted: Not Highlight                                                                                                                                                                                                                                                                         |
| Formatted: Not Highlight                                                                                                                                                                                                                                                                         |
| Formatted: Not Highlight                                                                                                                                                                                                                                                                         |
| Deleted: ??? with implications for ice hazard detection, oil spill and contaminant transport and shipping route assessments. Triplet areas also provide a signature of what is referred to by Thorndike (1986) as "nondivergent diffusive mixing" due to compressibility in the ice cover |

**Formatted: Highlight**

i) How can directional changes in sea ice drift trajectories be characterized? What insight is provided by single-particle dispersion statistics?

ii) How can associated/corresponding sea ice deformation processes for varying distances relative to the coastline be characterized? What insight is provided by two- and three-particle dispersion statistics?

5 We address these questions through the development of a framework for understanding sea ice drift and deformation in the Beaufort Sea on daily timescales based on single-, two-, and three-particle dispersion. Diagnostic information resulting from this framework can be used by modelers, satellite image analysts and in field observations to quantify relative contributions (atmospheric, oceanic, internal ice stress) to ice drift, Furthermore, these methods are relevant both from an observational and modelling 10 perspective, with the potential for application to forthcoming model-data comparisons, and an assessment

of dynamical regimes in other regions of the Arctic. The paper proceeds as follows. Data used to identify directional changes in sea ice drift are described in Section 2. In Section 3, methods based on Lagrangian dispersion and the triplet area approach, are presented. Results associated with each of the two objectives are provided in Section 4, followed by the

discussion in Section 5 and conclusions in Section 6 in addition to a short description of future work. 15

**2 Data**

Sea ice position data were obtained from an array of ten ice beacons and one ice mass balance buoy launched from the CCGS Amundsen in the marginal ice zone of the southern Beaufort Sea in September, 2009 (Figure 1). From this array, four triangular configurations were selected, hereinafter referred to as

- triplets A to D, to monitor divergence and convergence of sea ice, with initial inter-beacon distances of 20approximately 11, 11, 11.5, and 7 km for the shortest leg, and 15, 37, 11.5, and 12.5 km for the longest leg, respectively. Triplets A to D were deployed on multi-year ice (MYI) and labeled according to their proximity to the continental coastline: triplet A was located closest to the coastline, while triplet D was located furthest from the coastline. Position coordinates were available for all beacons in: triplet A until
- October 6th; triplet B until November 4th; triplet C until November 25th, and triplet D until November 3rd, 25 
[revised manuscript text omitted]

14

**Comment [6]:** This sentence belongs in the methodology. It is very obvious because it is a sentence by itself – an afterthought – missing in the methodology.

| -( | Deleted: stress                                                        |   |
|----|------------------------------------------------------------------------|---|
|    | Deleted: , with implications for rheological characterizations. |   |
|    | Deleted: and rheological characterizations of the sea ice cover |   |
|    |                                                                        | _ |

**4 Results**

In this section, we identify directional changes in sea ice drift and corresponding sea ice deformation characteristics using Lagrangian dispersion statistics. Two results in particular are emphasized in conjunction with triplet diagnostics relative to the distance from a coastline. First, single-particle dispersion

5 is investigated with respect to ice and external forces contributing to changes in sea ice drift direction, Second, sea ice deformation is examined using two- and three-particle dispersion.

**4.1 Single-particle dispersion identifies directional changes in sea ice drift,**

From September to November, 2009, eight regional-scale drift-direction changes are visually seen in the drift trajectory of centroid positions (Figure 1 and Table 1) from each of the four triplet arrays (A through D). These changes correspond to minima and maxima of the running variance from 3-day running means of centroid positions (Figure 2), For all triplets, these same time periods of directional change also match zonal and meridional inflection points. As a result, we identify these inflection points in zonal and meridional dispersion statistics near 12, 19, 25, and 30 September, and 5, 12, 18, and 29 October, which correspond to drift trajectory changes *e1* to *e8*, respectively (Figure 3).

With respect to external forcing connections, we focus first on one case study to understand the connections and then generalize afterward. We select the most dramatic change which occurs on 8 October by looking at the centroid drift minima variances (Figure 2), inflection points in regional absolute dispersion (Figure 3), and minima in SLP (Figure 5) all of which correspond plus or minus one day (within

- 20 a 3-day running mean) from the beginning of the time series (9 September) until 8 October. At this time, there is an Ekman convergence caused by an SLP high with strong off-shore ice drift and a deterioration of the ice cover (reduced ice area – Figure 7). From 8 October onward the SLP and drift centroid variance maxima are out of phase by approximately two to three days which indicates a loss of synchronicity in iceatmosphere interactions. Furthermore, total (i.e., zonal and meridional components combined in
- 25 quadrature) single-particle dispersion statistics for an ensemble including triplets A to D, to provide a regional characterization of sea ice drift (Figure 3), is governed by zonal dispersion prior to, with increased meridional dispersion contributions following, early October. Recalling that the scaling exponent  $\alpha$  describes sea ice dynamical regimes, total dispersion also shows that  $\alpha \sim 3$  before 8 October, which indicates a superdiffusive regime. After 8 October,  $\alpha \sim 2$ , which indicates a ballistic regime characteristic
- 30 of linear root mean square displacements in particle paths (Figure 3). Turning angles (Figure 5) further highlight loss of synchronicity in ice-atmosphere interactions and the ice cover following the 8 October SLP high event, evident in phase differences of opposite sign between triplets B, C, and D, and more

**Deleted: and discussion**

**Deleted:** and additional...triplet diagnosti**

... [10]

**Formatted: Heading 2**

**Deleted: and ...dentifying**

Trajectories for beacons deployed near 135 °W deployed near 135 °W between 72 °N and 75 °N capture spatiotemporal evolution in ice beacon triplet centroids beginning in September, 2009, with triplet A located closest to, and triplet D located furthest from, the continental coastline (Figure 1). Triplets A and B, deployed near 72 °N, share two of the three beacons and are advected westwards to approximately 144 °W and 158 °W, surviving until October 7th and November 5th, respectively. Triplet C is deployed near 73 °N and is also advected westward to 162 °W, surviving until November 26th. Triplet D is deployed near 74.5 °N and traverses a shorter path southwards and westwards to 145 °W, surviving until November 4th.

| Moved down [1]: Trajectories for beacons
deployed near 135 °W between 72 °N and 75 °N
capture spatiotemporal evolution in ice beacon
triplet centroids beginning in Sentember 200 [11] |
|--------------------------------------------------------------------------------------------------------------------------------------------------------------------------------------------------------|
| Deleted: regional-scale drift-direction ([12]                                                                                                                                                   |
| Formatted [13]                                                                                                                                                                                         |
| Comment [7]: What is an "interruption"?                                                                                                                                                                |
| Deleted: based on changes in the (regional [14]                                                                                                                                                 |
| Formatted: Highlight                                                                                                                                                                                   |
| Formatted: Highlight                                                                                                                                                                                   |
| Moved down [2]: Directional changes in [15]                                                                                                                                                            |
| Deleted: . Directional changesin sea [16]                                                                                                                                                              |
| Comment [8]: Define interruption                                                                                                                                                                       |
| Deleted: which are also known as interru [17]                                                                                                                                                          |
| Formatted[18]                                                                                                                                                                                          |
| Deleted: inflection points in zonal and m [19]                                                                                                                                                         |
| Deleted: ingto drift directional                                                                                                                                                                       |
| Formatted: Indent: First line: 1.27 cm                                                                                                                                                                 |
| Deleted: Comparison ofinima in centro [21]                                                                                                                                                             |
| Deleted:inima variancen SLP high: [22]                                                                                                                                                                 |
| Deleted: demonstratesorrespondence                                                                                                                                                                     |
| Deleted: Date until 8 October. At this [24]                                                                                                                                                            |
| Comment [9]: This is confusing. Don't t[25]                                                                                                                                                            |
| Comment [10]: If you use the word LA ( [27]                                                                                                                                                            |
| Formatted: Highlight                                                                                                                                                                                   |
| Formatted [26]                                                                                                                                                                                         |
| Formatted: Highlight                                                                                                                                                                                   |
| Comment [11]: If you use the word LA ( [28]                                                                                                                                                            |
| Formatted: Not Highlight                                                                                                                                                                               |
| Formatted [29]                                                                                                                                                                                         |
| Comment [12]: You lost me. Cut it back [30]                                                                                                                                                            |
| Deleted: ensemble i.e., in quadrature?                                                                                                                                                                 |
| Deleted: which                                                                                                                                                                                         |
| Deleted: both                                                                                                                                                                                          |
| Formatted[31]                                                                                                                                                                                          |

distinctive differences between all triplet turning angles, Hence, with this case study we see both a loss of synchronicity in ice-atmosphere interactions evident in turning angles, and a transition in sea ice dynamical regimes captured by single-particle dispersion for the ensemble of triplets,

With the single case study above now clarifying regional dispersion regimes from super-diffusive (αR ~
3) to ballistic (αR ~ 2), we examine absolute dispersion for an ensemble of three beacons comprising each triplet to provide a local characterization of sea ice drift (Figure 4), Triplets A and B are characterized by sub-diffusive (αL < 1) behaviour from 19 September, 2009 to 6 October, 2009, and from 17 – 21, October, 2009, and super-diffusive (αL > 2) scaling from 7 – 12 October, 2009, and following 22nd October (Figure 4b). Triplet C, located further from the coastline, experiences similar sub-diffusive behaviour prior to 6
October, 2009, and predominantly super-diffusive behaviour following 12 October, 2009. Triplet D,

- located furthest from the continental coastline experiences the smallest displacements, with sub-diffusive scaling to 5 October, 2009, and diffusive scaling following 19 October, 2009, with no instances of ballistic scaling. As is explored further below, these dynamical regimes correspond to changes in atmospheric and sea ice conditions.
- 15 In terms of external forcing connections, regional sea ice dispersion regimes appear to transitionduring a SLP maximum/high, accompanied by an Ekman convergence, and subsequently an offshore ice drift (Figures 3, 5, and 6). Furthermore, ice concentration plays an important role as follows. When the ice concentration is high (>80%), meridional drift speeds away from the coast are suppressed when high SLP sets up offshore Ekman drift (Figures 5, 6, and 7). However when ice concentration is low (<80%) offshore
- 20 SLP-induced Ekman drift increases due to a reduction in ice-ice frictional interaction (the so-called freedrift state). Enhanced ice drift is in addition observed following the 8 October SLP event in a manner consistent with a regional shift in dynamical regimes captured by the transition from  $\alpha_R \sim 3$  to  $\alpha_R \sim 2$  in regional absolute dispersion results (Figures 3 and 6). Correspondence between local sea ice drift and concentration is further reflected in local total absolute dispersion (Figures 4 and 7). In particular,
- 25 subdiffusive scaling ( $\alpha_{L} \le 1$ ) captures limited beacon displacements in high (> 95%) ice concentration regimes, while superdiffusive scaling ( $\alpha_{L} \ge 1$ ) captures enhanced transport for lower ice concentrations (< 95%), with the exception of triplet D following the 8 October event. In this case, triplet D experiences subdiffusive behaviour indicative of trapping or an obstruction even in lower ice concentrations, during which time local differences in ice drift and winds compared to triplets B and C are also observed.
- 30 Comparison of time series between ice drift and surface winds also show that *e1*, *e3*, *e4*, *e6*, and *e8* are accompanied by meridional reversals in surface winds (i.e. southerly to northerly for *e1*, *e4*, and *e8*, and northerly to southerly for *e3* and *e6*), whereas *e2*, *e5*, and *e7* are accompanied by persistent northerly winds (Figure 6). Results from this analysis show that meridional changes in surface winds associated with

| $\sim$ | Delete de 1                                            |
|--------|--------------------------------------------------------|
| 2      |                                                        |
|        | Formatted: Not Highlight                               |
|        | Deleted: BIG BOTTOM LINE                               |
|        | STATEMENT HERE.                                        |
| >      |                                                        |
|        | Formatted: Subscript                                   |
| >      | Formatted: Indent: First line: 0.7 cm                  |
| >      | Deleted:                                               |
| >      | Formatted: Subscript                                   |
|        | Deleted: to sub diffusive                              |
|        | Comment [13]: Is absolute the same as t( [33]          |
|        | Deleted: continue evaluating the full time [32] |
| >      | Formatted: Highlight                                   |
|        | Deleted: (                                             |
| >      | Deleted: )                                             |
| >      | Deleted:                                               |
|        | Formatted: Subscript                                   |
|        | Formatted: Subscript                                   |
| >      | Deleted: between beacons                               |
| 5      | Deleted: [35]                                          |
| 5      | Moved down [4]:                                        |
|        | Formatted [36]                                         |
| (      | Deleted: An assessment of                              |
|        | Deleted: and atmospheric conditions SLP                |
|        | Deleted: shows that the                                |
|        | Deleted: in dynamical regimes depicted in [37]         |
|        | Deleted: n                                             |
|        | Deleted: accompanying                                  |
|        | Comment [14]: "meridional displaceme [39]              |
|        | Comment [15]: Does this mean slower [40]               |
|        | Comment [16]: Entire sentence is too [41]              |
|        | Deleted: This is further reflected in the po [38]      |
|        | Formatted: Not Highlight                               |
|        | Formatted: Not Highlight                               |
|        | Formatted: Not Highlight                               |
|        | Deleted: Hence, c                                      |
|        | Deleted: T                                             |
| Ĉ      | Deleted: of                                            |
| Ĉ      | Formatted: Font:Italic                                 |
| 2      | Formatted: Font:Italic                                 |
| Ì      | Comment [17]: Reversals from what to what?             |
| Ż      | Deleted: ?? Which direction??                          |
| 2      | Formatted: Font:Italic                                 |
| 2      | Formatted: Font:Italic                                 |
| >      | Formatted: Font:Italic                                 |
| 2      | Formatted: Font:Italic                                 |
|        | Formatted: Font:Italic                                 |
|        | Formatted: Font:Italic                                 |
| 7      | Comment [18]: And so what is the botto                 |

directional changes in ice drift are captured by inflection points in regional total absolute dispersion; the transition in slopes from  $\alpha_R \sim 3$  to  $\alpha_R \sim 2$  documents enhanced meridional transport and deterioration in the ice cover following the 8 October SLP high. Local connections between sea ice drift and concentrations are captured in slopes for local total absolute dispersion, with  $\alpha_L < 1$  subdiffusive (( $\alpha_L > 1$  superdiffusive) regimes indicating reduced (enhanced) local ice dispersion and transport.

5

10

15

**4.2 Two- and three-particle dispersion and sea ice deformation,**

In this section we explore interactions beyond the sea ice drift, by looking at the differential drift or deformation between beacons in each array. In the context of two- and three-particle dispersion, we want to understand the relationships between the sea ice cover and the atmosphere relative to distance from the coastline,

With respect to sea ice response to external forcing as manifested in sea ice deformation, we examine the 8 October case study for two-particle (relative) and three-particle dispersion. Relative (two-particle) dispersion shows that total dispersion is initially (19 September to 5 October) governed by zonal separation for triplets A, B, and C, and by both zonal and meridional separation for triplet D (Figure 9).

- This distinction may be attributed to predominantly meridional motion (and along-shear transport) of triplet D along the eastern segment of the anticyclonic Beaufort Gyre. A significant decrease in zonal separation is observed closest to the coastline in triplet B due to convergence in the ice cover in response to the 8 October SLP high. Triplet B subsequently experiences comparable zonal and meridional separations (right
- 20 panel in Figure 9) until 26 October, following which enhanced zonal separations are observed as beacons encounter lower ice concentrations (Figures 7 and 9). Noteworthy also is a transition in two-particle dispersion scaling law values from  $\beta < 1$  to  $\beta > 1$  prior to and following 8 October, respectively.

We now examine relative dispersion during directional change events (Figure 9). Whereas events *e1, e3, e4, e6, and e8* are associated with reversals in surface winds, changes in along-shear (zonal for

- 25 triplets A, B, and C, meridional for triplet D) and cross-shear (meridional for triplets A, B, and C, zonal for triplet D) separation accompany northerly winds during events *e2*, *e5*, *e7*, with delayed responses following the SLP high on 8 October. During *e2*, an increase in along-shear transport is observed for all triplets, evident in an increase in zonal separations for triplets A, B, and C in the direction of transport along the southern portion of the anticyclonic Beaufort Gyre, and in meridional separations for triplet D in the
- direction of motion along the eastern segment of the BG. During *e5*, an onset in an increase in across-shear transport, namely meridional separation, accompanied by a decrease in zonal separation is observed for triplet B, which is sustained during the SLP high. Following *e7*, a delayed increase in along-shear transport

**Comment [19]:** Bad bottom line. This is better to use as introductory material for the next section. Need a better BOTTOM LINE SENTENCE

What is the overall scaling change and how to connect alpha to specific ice concentrations and specific wind speeds. That is the bottom line that these statements are pointing toward for this time series just as you did very nicely for the SLP paragraph above. Do that same thing here. Connect the dates. If you can connect the quantities – even better.

Quantitative means pulling real numbers out of the plots and establishing magnitude relationships. Showing a quantitative plot without quantitative narrative is a qualitative results. There are plots but can the authors interpret those plots? That is where the science of analysis comes in and is essential for a peer-reviewed paper.

**Deleted:** Local ice drift conditions are reflected in subdiffusive scaling ( $\alpha < 1$ ) in total dispersion indicating limited displacements for triplet B within a high ice concentration regime relative to triplets C and D, where beacons exist within lower ice concentration regimes (Figures 4 and 8) near 19 October.

| Moved down [5]: In summary
section show that single-particle d                                                                                                                                                                                                                                                                                                                                                                                                                                                                                                                                                                                         | , results from this ispersion ( [43])                                        |
|-----------------------------------------------------------------------------------------------------------------------------------------------------------------------------------------------------------------------------------------------------------------------------------------------------------------------------------------------------------------------------------------------------------------------------------------------------------------------------------------------------------------------------------------------------------------------------------------------------------------------------------------------------------|------------------------------------------------------------------------------|
| Formatted: Highlight                                                                                                                                                                                                                                                                                                                                                                                                                                                                                                                                                                                                                                      |                                                                              |
| Formatted: Highlight                                                                                                                                                                                                                                                                                                                                                                                                                                                                                                                                                                                                                                      |                                                                              |
| Deleted:                                                                                                                                                                                                                                                                                                                                                                                                                                                                                                                                                                                                                                                  | [44]                                                                         |
| Formatted: Heading 2                                                                                                                                                                                                                                                                                                                                                                                                                                                                                                                                                                                                                                      |                                                                              |
| Deleted: , and                                                                                                                                                                                                                                                                                                                                                                                                                                                                                                                                                                                                                                            |                                                                              |
| Deleted: in particular,                                                                                                                                                                                                                                                                                                                                                                                                                                                                                                                                                                                                                                   |                                                                              |
| Deleted: in                                                                                                                                                                                                                                                                                                                                                                                                                                                                                                                                                                                                                                               |                                                                              |
| Deleted: based on                                                                                                                                                                                                                                                                                                                                                                                                                                                                                                                                                                                                                                         |                                                                              |
| Deleted: relative to                                                                                                                                                                                                                                                                                                                                                                                                                                                                                                                                                                                                                                      |                                                                              |
| Deleted: , to characterize how the                                                                                                                                                                                                                                                                                                                                                                                                                                                                                                                                                                                                                        | ne sea ice ( [45])                                                           |
| Formatted: Font:(Default)                                                                                                                                                                                                                                                                                                                                                                                                                                                                                                                                                                                                                                 | +Theme                                                                       |
| Formatted                                                                                                                                                                                                                                                                                                                                                                                                                                                                                                                                                                                                                                                 | [46]                                                                         |
| Formatted: Font:(Default)                                                                                                                                                                                                                                                                                                                                                                                                                                                                                                                                                                                                                                 | +Theme                                                                       |
|                                                                                                                                                                                                                                                                                                                                                                                                                                                                                                                                                                                                                                                           |                                                                              |
| Deleted:                                                                                                                                                                                                                                                                                                                                                                                                                                                                                                                                                                                                                                                  | [47]                                                                         |
| Deleted: •
Formatted: Font:(Default)                                                                                                                                                                                                                                                                                                                                                                                                                                                                                                                                                                                                                   | [47]
+Theme Body                                                          |
| Deleted: -
Formatted                                                                                                                                                                                                                                                                                                                                                                                                                                                                                                                                                                                                    | [47]
+Theme Body
[48]                                                  |
| Deleted: -
Formatted
Formatted: Font:(Default) -                                                                                                                                                                                                                                                                                                                                                                                                                                                                                                                                                                     | +Theme Body
( [48]
+Theme Body                                         |
| Deleted: -
Formatted
Moved down [6]: Relative (tw                                                                                                                                                                                                                                                                                                                                                                                                                                                                                                                                     | +Theme Body
+Theme Body
+Theme Body
•o-particle([49]                |
| Deleted: -
Formatted
Moved down [6]: Relative (tw
Formatted: Font:(Default)                                                                                                                                                                                                                                                                                                                                                                                                                                                                                                            | [47]
+Theme Body
[48]
+Theme Body
•o-particle [49]
+Theme     |
| Deleted: -
Formatted
Moved down [6]: Relative (tw
Formatted: Font:(Default)                                                                                                                                                                                                                                                                                                                                                                                                                                                                               | [47]
+Theme Body
[48]
+Theme Body
vo-particle([49]
+Theme     |
| Deleted: -
Formatted
Moved down [6]: Relative (tw
Deleted:                                                                                                                                                                                                                                                                                                                                                                                                                                 | [47]
+Theme Body
[48]
+Theme Body
to-particle( [49])
+Theme   |
| Deleted: -
Formatted
Moved down [6]: Relative (tw
Deleted: a                                                                                                                                                                                                                                                                                                                                                                                                                   | [47]
+Theme Body
[48]
+Theme Body
to-particle([49]
+Theme     |
| Deleted: -
Formatted
Moved down [6]: Relative (tw
Formatted: Font:Italic                                                                                                                                                                                                                                                                                                                                                                                                                                    | [47]
+ Theme Body
[48]
+ Theme Body
• o-particle([49]
+ Theme |
| Deleted: -
Formatted
Moved down [6]: Relative (tw
Formatted: Font:Italic                                                                                                                                                                                                                                                                                                                                                                                                          | [47]
+ Theme Body
Theme Body
Theme Body
Theme
Theme           |
| Deleted:         Formatted:         Formatted         Formatted:         Fornt:         Italic         Deleted:         Deleted:         a         Formatted:         Font:         Italic         Deleted:         italic         Deleted:         with      | [47]
+ Theme Body
[48]
+ Theme Body
• Theme Body
+ Theme      |
| Deleted:         Formatted:         Formatted         Formatted:         Font:         Italic         Deleted:         Deleted:         Formatted:         Font:         Italic         Deleted:         With         Formatted:         Font:         Italic | [47]
+ Theme Body
[48]
+ Theme Body
ro-particle([49]
+ Theme  |
| Deleted: -
Formatted
Moved down [6]: Relative (tw
Deleted: a                                                                                                                                                                                                                                                                                                | [47]
+ Theme Body
[48]
+ Theme Body
ro-particle([49])
+ Theme |

and zonal separation for all triplets is observed, These results highlight contributions from along- and across shear transport to sea ice drift in response to atmospheric forcing.

Evaluation of triangle areas and differential kinematic parameters (DKPs) using three-particle dispersion further highlights changes in the ice cover prior to and following the 8 October SLP high (Figure

- 5 10). As is noted in the Methods section, DKPs are evaluated for time intervals excluding 14 22
   September (all triplets), 11 17 October and 19 22 October (triplet B) based on the analysis presented in Figure 9 monitoring triangle elongation and conditions for which triangles provide a reasonable estimate of DKPs. Regional, differences in triplet area representative of sea ice deformation are observed in the evolution of all triplet triangles (Figure 10). Triplet area evolution demonstrates enhanced variability in
- 10 triplet B relative to triplets A, C, and D, particularly following 8 October. An increase in triangle base (defined as the longest triangle side) is observed with decreasing distance from the coastline, in a manner consistent with local absolute dispersion (Figure 4). Base-to-height and perimeter to area ratios illustrate an equilateral configuration near the ice edge and coastline in the early stages of evolution for triplet A, and late stages of evolution for triplet B, captured in base-to-height ratio values approaching 1.155. Stretching
- 15 is observed closest to the coastline following & October within, a consolidated ice regime (Triplet B), captured in increasing perimeter-to-area and base-to-height ratios.

As for two-particle dispersion, changes in area, height, and P/A and b/h ratios capture deformation associated with events *e2*, *e5*, and *e7*, with delayed responses to northerly winds following the 8 October SLP high (Figure 10). Following *e2*, the base for triplets A and B increases and height decreases during consolidation, resulting in enhanced P/A and b/h. By contrast, P/A and b/h are approximately constant for triplets C and D, indicating that the triangle maintains its shape further from the continental coastline. Following *e5* and the SLP high, the greatest stretching (high base and ratio values) is observed in triplet B located closest to the coastline. Several days following *e7* a decrease in area for triplet B is associated with a decrease in height so that b/h and P/A increase, while once again the area and ratios for triplet C and D remain constant.

Local differences in sea ice deformation are further reflected in differential kinematic parameters (DKPs) defined as the weighted time rate of change in triplet area, as described in the methods section (Figure 11). Results show that sea ice deformation is approximately an order of magnitude smaller for triplets furthest from the coastline ( $\sim 10^{-5}$  for triplet A versus  $\sim 10^{-4}$  for triplet B). A strong shear event is in addition observed for triplet B during the SLP high on 8 October, and for triplet C on 10 October. In consideration of directional changes due to northerly winds e5 and e7, during e5, weak DKPs are observed for triplet C, with triplets B and D governed by vorticity. During e7 convergence and divergence dominate

for triplet B.

| following the 8 October SLP high                                                                                                                                                                                                                                                                                                                                                                                                                                         |                |
|--------------------------------------------------------------------------------------------------------------------------------------------------------------------------------------------------------------------------------------------------------------------------------------------------------------------------------------------------------------------------------------------------------------------------------------------------------------------------|----------------|
| Comment [21]: This is a discussion point – move to discussion.                                                                                                                                                                                                                                                                                                                                                                                                           |                |
| Moved down [7]: Furthermore, $t^3$ scaling
associated with Richardson's scaling law and
attributed to atmospheric dispersion as described i
Rampal et al. (2009), is evident ( $\beta$ ~3.4) for triple
from 10 – 17 October; higher scaling exponents
exist for triplets B and D following the SLP high of
8 October.                                                                                                                                 | n
t C
on |
| Deleted:                                                                                                                                                                                                                                                                                                                                                                                                                                                                 | D              |
| Deleted:                                                                                                                                                                                                                                                                                                                                                                                                                                                                 |                |
| Deleted: Specifically, triplet area                                                                                                                                                                                                                                                                                                                                                                                                                                      |                |
| Deleted: with an increase in area near the ice ed
in late October/early November (Figure 11).                                                                                                                                                                                                                                                                                                                                                                  | ge             |
| Deleted: (i.e. higher base values for triplets A a B relative to triplets C and D),                                                                                                                                                                                                                                                                                                                                                                               | nd             |
| Deleted: The triplet base also provides a measur
of two-particle dispersion and separation between
pair of particles/beacons/ice floes. Triplet area is
governed by height. Base-to-height and perimeter
area ratios illustrate an equilateral configuration ne
the ice edge and coastline in the early stages of
evolution for triplet A, and late stages of evolution
for triplet B, captured in base-to-height ratio value
approaching 1.155. | to
ear
s |
| Deleted: 10                                                                                                                                                                                                                                                                                                                                                                                                                                                              | $ \ge$         |
| Deleted: during                                                                                                                                                                                                                                                                                                                                                                                                                                                          | $ \rightarrow$ |
| Formatted: Font:Italic                                                                                                                                                                                                                                                                                                                                                                                                                                                   | $ \ge$         |
| Formatted: Font:Italic                                                                                                                                                                                                                                                                                                                                                                                                                                                   | $ \ge$         |
| Deleted: 11                                                                                                                                                                                                                                                                                                                                                                                                                                                              | $ \rightarrow$ |
| Formatted: Font:Italic                                                                                                                                                                                                                                                                                                                                                                                                                                                   | $ \rightarrow$ |
| Deleted: n                                                                                                                                                                                                                                                                                                                                                                                                                                                               | _              |
| Comment [22]: This sentence belongs in the methodology. It is very obvious because it is a sentence by itself an afterthought – missing in the methodology.                                                                                                                                                                                                                                                                                                       | _              |
| Deleted:                                                                                                                                                                                                                                                                                                                                                                                                                                                                 | D              |
| Deleted: 2                                                                                                                                                                                                                                                                                                                                                                                                                                                               | $\prec$        |
| Comment [23]: Number not just relative
qualifier                                                                                                                                                                                                                                                                                                                                                                                                                      | _              |
| Formatted: Superscript                                                                                                                                                                                                                                                                                                                                                                                                                                                   | $ \rightarrow$ |
| Formatted: Superscript                                                                                                                                                                                                                                                                                                                                                                                                                                                   | $ \rightarrow$ |
| Deleted: For triplet A, located closest to the
coastline, sea ice deformation is characterized
predominantly by vorticity and stretching (Figure
12). Triplet B is governed by vorticity for the
duration of this triplet evolution, with contribution
from shear until 8-10 October, and divergence and
stretching following mid-October. Triplet Come
53                                                                                          | s
I         |
| Deleted: Located furthest from the coastline,
triplet D is governed by vorticity, stretching                                                                                                                                                                                                                                                                                                                                                                   | Ð              |
| Deleted: e2,                                                                                                                                                                                                                                                                                                                                                                                                                                                             |                |
| Deleted: ,                                                                                                                                                                                                                                                                                                                                                                                                                                                               |                |
| Deleted: during e2 , Triplet B experiences a transition from vorticity and stretching to sh [55]                                                                                                                                                                                                                                                                                                                                                           |                |
| Deleted:                                                                                                                                                                                                                                                                                                                                                                                                                                                                 | D              |

[revised manuscript text omitted]

Relative (two-particle) dispersion provides a signature of changes in along- and cross-shear transport in response to atmospheric and ice conditions (Table 2). Specifically, two-particle dispersion characterizes sea ice deformation, and local changes in the ice cover. Enhanced meridional separation (cross-shear transport) closest to the coastline and near the ice edge (triplet B) following the 8 October SLP high is captured by a transition in meridional, zonal, and total two-particle dispersion. Furthermore,  $t^3$ scaling associated with Richardson's scaling law and attributed to atmospheric dispersion as described in Rampal et al. (2009), is evident ( $\beta \sim 3.4$ ) for triplet C from 10 – 17 October; higher (exponential and

- 15 algebraic) scaling exponents exist for triplets B and D following the SLP high on 8 October. Exponential growth for triplet B indicates the existence of a single-timescale associated with non-local behaviour and unimpeded westward advection due to anticyclonic circulation in lower ice concentrations (reduced frictional effects in lower sea ice concentrations).
- As noted in the Introduction, previous studies of sea ice dynamics in the Beaufort Sea havedemonstrated that the sea ice cover moves as an: aggregate for spatial scales exceeding 100 km; aggregate for an elliptic (homogeneous) regime or discrete entity for a hyperbolic (discrete) regime for scales ranging from several to 100 km; and is governed by ice-ice interactions for scales on the order of 1 km (Overland et al., 1995). Two-particle results from the present investigation show that meridional separations range from ~10 to ~30 km, and zonal separations from ~10 km to spatial scales exceeding 100 km (triplet B) (Figure
- 8). The total (meridional and zonal combined in quadrature) two-particle scaling exponent β indicates that the ice cover in this study experiences both regimes (i.e. moves as an aggregate and discrete entity), evident in a transition in the scaling exponent from β < 1 to β > 1 following the 8 October SLP high, thus providing a measure of connectivity in the sea ice cover. Furthermore, β ≥ 3 for triplets C and D in high ice-concentrations, and β ~ exp(t) for triplet B in low ice concentrations, in late October, which further distinguishes elliptic (local) from hyperbolic (non-local) behaviour.
  - Deformation components, or differential kinematic parameters (DKPs) are computed using threeparticle dispersion and changes in triangle area (Table 2). Results from an evaluation of the Okubo-Weiss criterion, which monitors relative contributions from total deformation (divergence and shear combined in

**Comment [24]:** This is a discussion point – move to discussion.

| Formatted: Font:12 pt                  |   |
|----------------------------------------|---|
|                                        |   |
|                                        |   |
| Formatted: Font:12 pt                  | _ |
| Formatted: Font:12 pt                  |   |
| Formatted: Font:12 pt                  |   |
| Formatted: Font:12 pt, Italic          |   |
| Formatted: Font:12 pt                  |   |
| Formatted: Font:12 pt                  |   |
| Formatted: Normal, Indent: First line: |   |

quadrature) and vorticity, show enhanced deformation (OW > 0) associated with loss of connectivity (captured by a transition in two-particle dispersion scaling exponent from  $\beta < 1$  to  $\beta > 1$  following the 8 October SLP high). Deformation was also found to be 100 times larger near the coast, with vanishing values for triplets C and D indicating weak deformation further from the coastline. Evaluation of individual

- 5 directional change events in the context of three particle dispersion also showed that  $\theta$ , in enabling additional distinction between divergence, convergence, and shear to total deformation strain states in particular, namely uniaxial extension, ( $\theta \sim 45$ ), shear ( $\theta \sim 90$ ), and uniaxial compression ( $\theta \sim 135$ ), can potentially provide a signature of failure in the ice cover, as is described further below.
- Previous studies have demonstrated correspondence between ice deformation and stress
   measurements in the Beaufort Sea (Richter-Menge et al., 2002). Sea ice strength is dependent on ice concentration and thickness. The ice cover fails when the internal ice stress is comparable to ice strength. Previous studies have also shown that uniaxial extension can give rise to ridging for a particular alignment of flows and anisotropic ice cover (Wilchinsky and Feltham, 2006). Failure in the ice cover is in addition attributed to compression within a region of confinement or what is referred to as "trapping" in the present
- 15 study. Results from this analysis suggest that the three-particle dispersion approach, and shear-todivergence ratio in particular can be used in combination with ice concentrations and (in future studies) thickness to assess conditions for which failure in the ice cover occurs (Figures 7 and 12). Convergence ( $\theta \sim 180$ ) experienced by triplet C on 26 September within a high ice concentration regime, and subsequent reduction in SIC encountered by triplet beacons suggests local failure in the ice cover. In addition,
- 20 compression experienced by triplet B during the 8 October SLP high and Ekman convergence coincides with reduced ice concentrations and the strong ice shear event (Figures 7, 11, and 12), also indicating failure in the ice cover for triplets C and D. This deterioration further continues near the ice edge for triplet B to 20 October as shear-to-divergence values on the order of 180 are accompanied by significant reductions in ice concentrations as the triplet encounters the ice edge and free drift conditions. These results
- 25 also suggest that directional change events  $\underline{\rho}_{2,k}e_{5,k}$  and  $\underline{\rho}_{7,k}$  associated with along-shear transport, northerly surface winds and ice deformation anticipate failure in the ice cover closest to the coastline (evident in lower  $\theta$  values following each event for triplet B), although additional study is required to confirm this hypothesis.

Table 3 synthesizes and describes sea ice dynamical regimes characterized by single- and twoand  $\beta$ , respectively. In the context of sea-ice drift, the regional single-particle dispersion scaling exponent  $\alpha_{\rm R}$  describes flow topology and organized structure in the flow (ice drift) field; the local single-particle dispersion scaling exponent  $\alpha_{\rm L}$  describes local changes in the sea ice state. In the context of sea ice deformation,  $\beta$  provides a signature of spatial gradients in the ice drift field. The subFormatted: Font color: Black

**Comment [25]: Discussion Moved (insertion) [5]**

Deleted: In summary, results from this section show that single-particle dispersion (namely inflection points in meridional single-particle dispersion statistics) captures directional changes evident in ice drift paths and minima in centroid drift variance, and provides a regional characterization of sea ice drift. Local differences in ice drift at varying distances from the coastline are manifested in absolute dispersion for beacons associated with each triplet. Moreover, absolute dispersion demonstrates a shift in sea ice dynamics during the SLP high on 8 October responsible for convergence in the ice cover, poleward retreat in the ice edge, and strong offshore ice drift. Formatted: Not Highlight Formatted: Not Highlight Formatted: Normal Formatted: Font color: Auto Formatted: Font color: Auto, Not Highlight Formatted: Font color: Auto Formatted: Not Highlight

| ( | Formatted: Font:Italic, Not Highlight |
|---|---------------------------------------|
| ( | Formatted: Not Highlight              |
| ( | Formatted: Font:Italic                |
| ( | Formatted: Font:Italic                |

diffusive regime ( $\alpha_{L} \le 1$ ) for separations defined by fixed spatial scales over time ( $\beta \le 1$ ) characterizes trapping associated with ice-ice interactions in high ice concentrations and an ice cover with high connectivity that moves as an aggregate. This behaviour is observed for directional change events e3 to e5 with characteristic length scales of 30, 40, and 10 km for triplets A and B, C, and D, respectively.

- 5 Following the 8 October SLP high event, triplet C experiences a super-diffusive regime ( $\alpha_L \ge 1$ ) for increasing separations ( $\beta \sim 3$ ) indicating loss of connectivity in the ice cover, with increases in scale separation from ~ 40 to 70 km in ice concentrations ranging from 90-95% also associated with deterioration in the ice cover. By contrast, triplet D located further from the coastline experiences a subdiffusive regime ( $\alpha_L \le 1$ ) for increasing separations ( $\beta \sim 3.8$ ), with increases in scale separation from ~10 to 15 km,
- 10 indicating limited displacements and sustained trapping for enhanced separations. Triplet B, located closest to the coastline experiences the most rapid increase in separations (from ~ 15 km to 70 km) and limited displacements ( $\alpha_{L}$  <1) within a high ice concentration regime, providing a signature of reduced connectivity, enhanced deformation, and compression associated with ice convergence due to the 8 October SLP high. An exponential scaling law for triplet B indicates sea ice dynamics governed by large-scale
- 15 processes and advection along the southern branch of the anticyclonic Beaufort Gyre. In general terms, the upper right corner of Table 3 depicts sea ice dynamics governed by atmospheric processes ( $\alpha$ ,  $\beta > 1$ ), and the lower left corner depicts sea ice dynamics governed by ice interactions and coastline ( $\alpha$ ,  $\beta < 1$ ). Sea ice deformation monitored by the Okubo-Weiss criterion and shear-to-divergence ratio illustrates regional sea ice response to external forcing based on distance from the coastline, with higher deformation values closer
- 20 to the coastline, and transitions from convergence and shear to divergence providing a local signature of failure in the ice cover. This framework thus provides a prescription for understanding sea ice drift and deformation based on single-, two-, and three-particle dispersion and associated diagnostics.

**6** Conclusions**

- 25 In this study we developed a framework to characterize, using single- and two-particle scaling laws and three particle dispersion diagnostics, directional changes in sea ice drift and associated deformation processes in response to atmospheric forcing based on Lagrangian dispersion statistics. We tested this approach using single-, two-, and three-particle dispersion applied to beacon arrays deployed in a triangular configuration as triplets at varying distances from the coastline in the southern Beaufort Sea.
- 30

In consideration of our first research question, single-particle dispersion characterizes directional changes in sea ice drift trajectories and dynamical changes in the ice cover. Specifically, single-particle dispersion captures i) a shift in ice dynamical regimes following the 8 October SLP high, ii) inflection

22

|     | mattadi | Subcorint |
|-----|---------|-----------|
| гог | maileu. | SUDSCHDL  |

**Formatted: Font:Italic**

Comment [26]: This is a discussion point – move to discussion.

**(Moved (insertion) [7]**

**Deleted:** Furthermore,  $t^3$  scaling associated with Richardson's scaling law and attributed to atmospheric dispersion as described in Rampal et al. (2009), is evident ( $\beta \sim 3.4$ ) for triplet C from 10 – 17 October; higher scaling exponents exist for triplets B and D following the SLP high on 8 October.

**Formatted: Normal**

**Comment [27]:** Need to be able to say here Quantitatively characterize.

But before you can say that, you need to actually do that by beefing up the number in the tables with more real number relationships.

In the end ask yourself, how do I use these results to make a decision about what the ice is doing? That is a great discussion point. If a framework can't be understood, how does one use it? So first and foremost, use the results and discussion sections to provide practical guidance to the reader in terms of how to use this new framework.

That is the point of a methods paper

Be very careful with the word "regime" you use this for many things (dispersion, ice concentration, ...) Choose one regime with dispersion.

In the end, the science is good, it's the need for simpler and more linear flow.

[revised manuscript text omitted]

×

... [58]

---

## Author Response (AR3)

**Responses to Editor's Comments**

Dear Jenny:

Thank you for your comments and suggestions.

Please find below responses to your corrections and suggestions, in red text:

Both reviewers were very happy with your improvements to the paper and recommend publishing. I am in agreement.

The authors would like to thank you and both reviewers for your comments and suggestions, which have helped to focus and improve the manuscript.

Page 3, Line 22: Consider not starting the sentence with 'And so', perhaps just drop this. Not proper English grammar.

Thank you for this suggestion. The authors agree and the phrase "And so" has been removed.

Page 3, Line 32: Do you mean "Two-particle dispersion also explains". Is it more appropriate to say "describes"

It is more appropriate to say "describes" and the text has been changed accordingly.

Page 4, line 2: Ditto for "multi-particle analysis explains".

This too is changed in the revised manuscript.

Page 8, do you mean to have a prime on y in the equation? Number equations.

This was intended to be a comma rather than a prime on y in the equation. In addition, in consideration of your comments, equations are numbered in the revised manuscript.

Page 19, line 20: "shear-to-divergence"

This has been changed. Thank you.

Take this opportunity to make sure your main findings are clearly described in the conclusions.

Thank you, it has been a pleasure working with you.

It has been a pleasure working with you as well. The authors would like to thank you for your insight, contributions, and suggestions, which helped to clarify and focus the manuscript, and contributed to its improvement.

Regards,

Jennifer

Non-public comments to the Author and Editor:

Would you consider thanking the reviewers in your acknowledgments?

The authors thank and acknowledge your contributions and those of the reviewers in the revised version of the manuscript; your insight, patience, and suggestions are appreciated. However, am not sure if it is appropriate to include the names of the reviewers, since the second reviewer indicated a preference for anonymity in the acknowledgements, while the first reviewer indicated that anonymity is not a requirement. Please let me know what you think would be most appropriate.

Thanks once again for your help and suggestions.

Regards,

Jennifer